# The evolution of antibiotic resistance islands occurs within the framework of plasmid lineages

Yiqing Wang [1] & Tal Dagan [1] ✉

Bacterial pathogens carrying multidrug resistance (MDR) plasmids are a major threat to human health. The acquisition of antibiotic resistance genes (ARGs) in plasmids is often facilitated by mobile genetic elements that copy or translocate ARGs between DNA molecules. The agglomeration of mobile elements in plasmids generates resistance islands comprising multiple ARGs. However, whether the emergence of resistance islands is restricted to specific MDR plasmid lineages remains understudied. Here we show that the agglomeration of ARGs in resistance islands is biased towards specific large plasmid lineages. Analyzing 6784 plasmids in 2441 *Escherichia*, *Salmonella*, and *Klebsiella* isolates, we quantify that 84% of the ARGs in MDR plasmids are found in resistance islands. We furthermore observe rapid evolution of ARG combinations in resistance islands. Most regions identified as resistance islands are shared among closely related plasmids but rarely among distantly related plasmids. Our results suggest the presence of barriers for the dissemination of ARGs between plasmid lineages, which are related to plasmid genetic properties, host range and the plasmid evolutionary history. The agglomeration of ARGs in plasmids is attributed to the workings of mobile genetic elements that operate within the framework of existing plasmid lineages.

Plasmids are autonomously replicating genetic elements that reside in prokaryotic organisms. Their ability to persist in the host population often depends on auxiliary functions encoded in their genome that are beneficial for their host, for example, resistance to antibiotics. The ability of plasmids to transfer horizontally between bacterial cells renders them a common driver of antibiotic resistance gene transfer[1]. Indeed, the emergence of antibiotic resistant bacteria in clinical environments is often linked to the acquisition of plasmids encoding antibiotic resistance genes (ARGs; e.g. refs. 2–4). Plasmids that encode multiple ARGs, termed multiple drug resistance (MDR) plasmids, provide their host with a combination of resistances to multiple antibiotics[5]. Pathogens carrying MDR plasmids are one of the major threats to human health. For example, an epidemiological study estimated that *Escherichia coli* strains were responsible for most human

mortality attributed to antibiotic resistance in 2019, followed by *Klebsiella pneumonia* strains[6].

The evolution of antibiotic resistance gene content in MDR bacteria is often facilitated by diverse mobile genetic elements (MGEs) that can mediate gene transfer between DNA molecules via homologous recombination, site-specific recombination, or transposition[7]. These include insertion sequences (ISs) and transposons[8], as well as integrons[9,10], all of which can be associated with adjacent or cargo genes that encode for antibiotic resistance (note that this MGE definition excludes plasmids). Examples are IS*26*, which is a frequent component in pseudo-compound transposons[11], which mobilize diverse ß-lactamase variants, and the sulfonamide resistance gene *sul1* that is associated with class 1 integrons (reviewed in ref. 7). The transposition and recombination of genetic elements in bacterial

[1]Institute of General Microbiology, Kiel University, Kiel, Germany. ✉e-mail: tdagan@ifam.uni-kiel.de

genomes generate diverse ARG combinations[12] that typically cluster in specific genomic loci termed antimicrobial resistance island (REIs)[13] or multi-resistance regions (MRRs)[7]. For example, IS*26*-mediated trans-locatable units (TUs) have been shown to insert most efficiently adjacent to an existing IS*26* copy to form a pseudo-compound transposon[11,14,15]. The assembly of resistance islands in plasmid genomes has been studied in detail for specific plasmid types and bacterial taxa (e.g., refs. 16–18), nonetheless, general quantifications of MGE contribution to the evolution of antibiotic resistance gene content in MDR plasmids are lacking.

Plasmids encoding MDR genes are typically large and their genome size is positively associated with the number of ARGs in the plasmid genome[19]. Small plasmids typically carry only a few ARGs (e.g., ColE1-like plasmids[20]). Notably, plasmid size and plasmid mobility are associated since conjugative plasmids are typically large[21]. A recent study suggested that ISs associated with ARGs are frequently found in conjugative plasmids[22–24]. Disentangling the association of plasmid genome size and plasmid mobility with a high number of ARGs requires us to consider the plasmid evolutionary history. Indeed, recent studies in the field of plasmid biology suggested a classification scheme of plasmids into taxonomic units (PTUs[25,26]). These are groups of putatively closely related plasmids, that are inferred based on genome sequence similarity and specific plasmid backbone genes[27]. Similar to organisms in the same species, plasmids in the same taxonomic unit are expected to have similar plasmid properties and a common evolutionary origin. Integrating the plasmid taxonomic unit in MDR plasmid research enables us to go beyond the plasmid most general properties (i.e., size and mobility) while accounting for the evolutionary history of plasmid lineages.

Here we study the evolution of antibiotic resistance gene content in different lineages of MDR plasmids, with a focus on three genera: *Klebsiella*, *Escherichia* and *Salmonella* (*KES*). Plasmids in *KES* strains share gene content, suggesting low barriers for plasmid transfer among hosts in the three genera as well as common plasmid origins[25]. To examine ARGs in resistance islands, we analyze colinear syntenic regions of cooccurring antibiotic resistance genes and mobile genetic elements. Furthermore, we compare the syntenic regions identified as resistance islands among plasmids in different plasmid taxonomic units.

## Results

### The majority of resistance genes in *KES* plasmids are clustered in compact resistance islands

For the purpose of our study, we clustered 11,995,860 protein-coding genes in chromosomes and plasmids of 2441 *KES* isolates into 32,623 clusters of homologous gene families (Supplementary Data 1). Using the comprehensive antibiotic resistance database (CARD)[28], we identified 114,457 homologs of 397 ARGs in 138 gene families (Fig. 1A). Members of those 138 families not identified by CARD as ARGs were excluded from further analysis. Most of the excluded genes (97%) correspond to chromosomal gene variants not classified as ARGs in CARD. Note that ARGs in our data are clustered into homologs by global protein sequence similarity, hence specific epidemiologically relevant gene variants may be clustered into the same gene family (see Supplementary Data 2). The majority (2013; 78%) of the 2591 ARG-coding plasmids in our set comprise multiple ARGs. Significant cooccurrence of ARG families was tested for all possible paired ARG combinations. About 30% (ca. 260) of the tested ARG pairs were found to significantly cooccur in plasmids, with little variation among the genera (Supplementary Data 3). These ARG pairs are termed here: coARGs. Cooccurring ARGs that are associated with the same MGE are expected to have a similar genomic neighborhood (i.e., similar neighboring genes). As the next step in the analysis workflow, coARGs were retained only if they were found within the same collinear syntenic block (CSB) (Fig. 1A; CSBs are clusters of genes whose order is conserved in

multiple genomes[29]). The majority of coARGs cooccurred in at least one plasmid CSB (e.g., 85% coARG combinations in *Escherichia*) (Supplementary Data 3). The CSBs that contained coARGs were further filtered based on their comparison to sequences of previously recognized transposable elements[30–32]. Only CSBs that match the known transposable elements were retained, which was the majority of CSBs comprising coARGs (e.g., 99.6% in *Escherichia*; Supplementary Data 3; Fig. 1A). The length of the retained CSBs had a median of 8 genes, with the majority (65%) of CSB instances comprising ≤10 genes (Supplementary Fig. S1). The CSBs identified here do not necessarily correspond to intact MGEs or complete resistance islands, which may be found in plasmids in partial forms, e.g., due to genome rearrangements and MGE degradation. The majority of the CSBs we identify here are better described as pieces of genetic elements[12], specifically here, 'pieces of resistance islands.' That being said, some of the CSBs we identified are much longer than MGEs and correspond to conserved MGE genomic neighborhoods (see Supplementary Note SN1). Notably, the CSBs we identified are found in the majority of MDR plasmids (1866; 93%). Taken together, most cooccurring ARGs in *KES* plasmids are organized in CSBs bearing similarity to transposable elements.

To examine which genetic elements are involved in the generation of resistance islands, we identified all the gene families encoding either transposases or site-specific recombinases (collectively referred here as SSRs for simplicity) in our dataset. The majority of SSR gene families encode DDE transposases (257 families) and tyrosine recombinases (161 families). The large number of families in these functional classes is most likely related to frequent gene duplications. Since our clustering approach is aimed to classify mainly orthologs (as in ref. 33, but considering multiple replicons), gene families having frequent (non-identical) duplicates on the same replicon are typically split by our clustering procedure into multiple clusters that correspond to different transposase or recombinase variants (e.g., the paralogs IS*26* and IS*26*-v1[34,35]). In addition, we characterized families encoding DEDD transposases (31 families), HUH endonucleases (25 families), serine recombinases (34 families), as well as three putative Cas1 endonucleases families (Supplementary Data 4). Most of the SSR families (107 out of 116) are characterized by a biased distribution towards plasmids compared to chromosomes (Supplementary Data 4). A total of 76 SSR gene families cooccur with 56 ARG families in plasmids within the CSBs (i.e., resistance island pieces). The most frequent SSR gene families include four DDE transposases (IS*26*, IS*26*-v1, IS*6100*, Tn*3* transposase), two serine recombinases (Tn*3* resolvase), a tyrosine recombinase (class 1 integron integrase), a HUH endonuclease (IS*91* transposase) and a DEDD transposase (IS*110* transposase). Together, these gene families comprise 66% of the SSR genes in antibiotic resistance islands as identified in plasmids using our analysis workflow.

Several of the transposases and site-specific recombinases are expected to coincide in genomes as components of functional transposable elements. In our data, components of Tn*3* elements, including the two gene families of Tn*3* resolvase and the family of Tn*3* transposases, significantly cooccur in plasmids ($P < 0.001$, using Fisher test), which serves as an internal validation for our approach. Different SSR genes may also cooccur due to causal interference between different transposable elements. For example, IS*4321*/IS*5075* (IS*110* family transposases) are known to target 38 bp terminal inverted repeats of Tn*21* subgroup of Tn*3* family[36]. Indeed, in our data IS*110* family transposases significantly cooccur with Tn*3* family transposase in plasmids ($P < 0.001$, using Fisher test), hence their distribution in plasmids is not independent. Similarly, families encoding transposases of IS*110* and Tn*3* cooccur in CSBs (i.e., pieces of resistance islands) significantly more than expected by chance alone ($P < 0.001$, using Fisher test). We note that while integrons are often found in transposons, our approach cannot infer the exact contribution of nested integrons to resistance islands. In our data, tyrosine recombinases (9 gene families) correspond to 16% of the total SSR genes in the plasmid

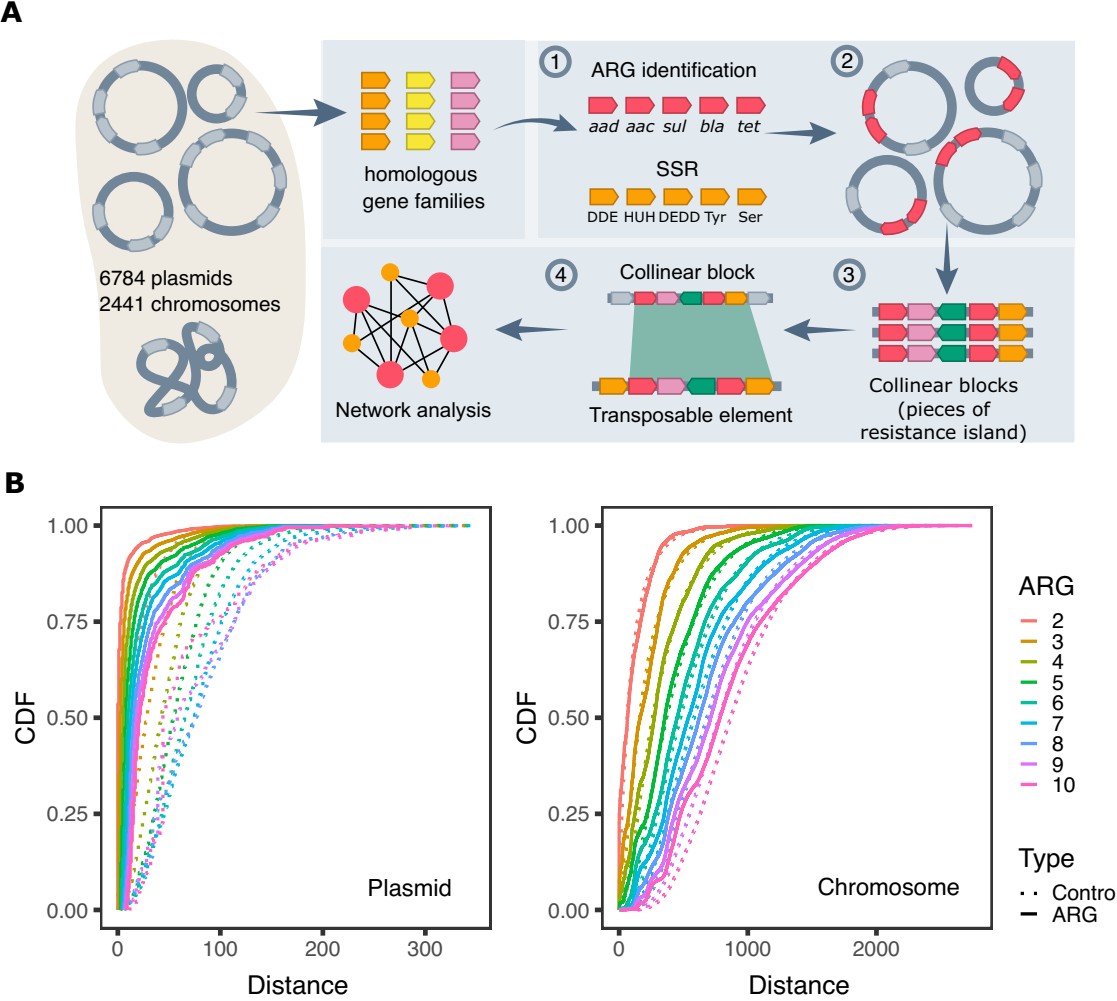

**Fig. 1 | Antibiotic resistance genes cooccur in plasmids in dense resistance islands. A** An illustration of our workflow for detecting resistance islands in plasmid genomes. Arrow boxes correspond to genes. (1) Protein-coding genes in bacterial genomes were clustered into gene families. ARGs and SSR genes were identified. (2) Significant cooccurrence tests for ARG pairs were conducted per replicon type in each genus. (3) Collinear syntenic blocks (CSBs) including coARGs were identified across genomes and (4) further compared to previously recognized transposable elements by sequence similarity (see Methods for additional clarifications). (**B**) Cumulative Distribution Function (CDF) plots show that the distance between ARGs is shorter than the distance between a control group of randomly sampled genes in plasmids (left), but not in chromosomes (right). Intergenic distances were calculated by counting the number of protein-coding genes between two, and up to ten, adjacent ARGs. Source data are provided as a Source Data file.

resistance islands and are present in 1116 (55%) MDR plasmids within resistance islands. Taken together, the resistance islands identified using our workflow comprise diverse combinations of SSRs and antibiotic resistance genes.

To further study the distribution of ARGs in plasmid genomes, we examined the distance between consecutive ARGs by counting the number of protein-coding genes between two, and up to ten ARGs. If the ARGs are randomly distributed in the genome, the distance distribution should fit to a negative binomial distribution. Our results show that the distance between consecutive multiple ARGs deviates significantly from the random expectation (using Goodness-of-fit test, distance distributions were tested separately with α = 0.05). Furthermore, the distance between multiple ARGs was significantly shorter than the distance of the control group of randomly sampled genes (Wilcoxon rank sum test, distance distributions were tested separately, α = 0.05). Notably, the median distance between two adjacent ARGs in plasmids is zero, which means that over half of the ARGs have an ARG as neighbor gene on the same strand. ARGs are thus densely clustered in plasmid genomes. Further evaluation of the ARG loci in plasmids

showed that 84% of ARGs in MDR plasmids are found in the CSBs we identified, i.e., in compact resistance islands.

## ARG and SSR gene combinations in resistance islands are diverse and change little over time

To gain an insight into the composition of ARGs and SSR genes in resistance islands we constructed a network of gene cooccurrence in resistance islands (i.e., the CSBs identified in our workflow). Nodes in the network correspond to either ARG or the main SSR gene families. The gene nodes are connected by edges if they coincide in at least one resistance island with the edge weight corresponding to the number of plasmids where the cooccurrence was observed. The resulting network has a highly condensed structure, that is, the genes (nodes) are highly connected among each other within the plasmid resistance islands (Fig. 2A). Highly connected hub-genes appear at the center of the network: these correspond to two transposase genes IS*26* and IS*26*-v1 and resistance genes *sul1* and *qacE*Δ1. The central location in the network indicates that these genes occur in combinations with most SSR genes and ARGs in plasmid resistance islands.

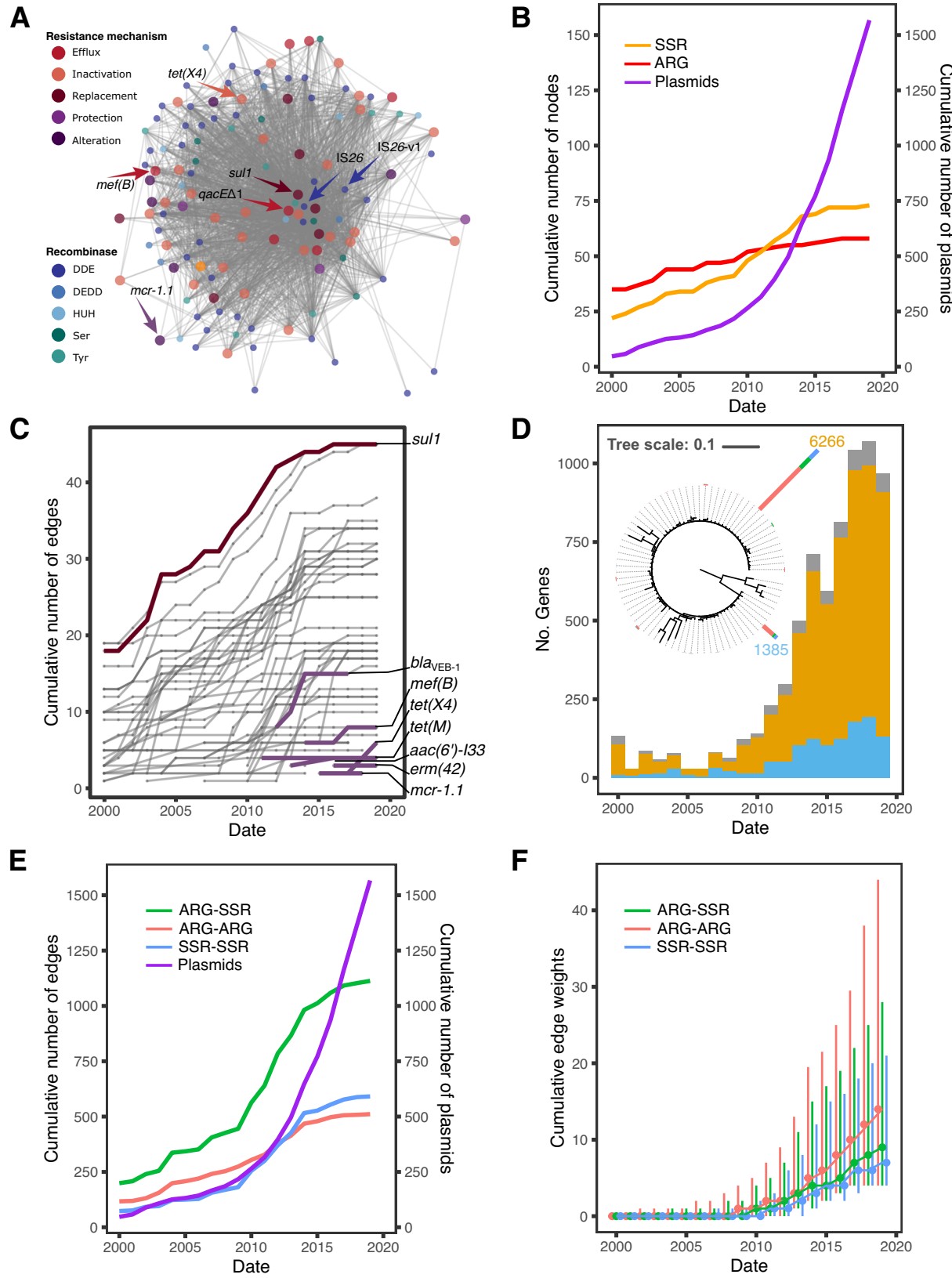

To investigate temporal dynamics in the cooccurrence network we labeled each edge with the earliest report of the genes' cooccurrence in a resistance island. Estimating the age of resistance islands from sequence divergence is challenging due to the high sequence conservation of both ARGs and SSR genes[24,37]. Here we use the isolation date of the respective strain as a proxy for the time of presence of

specific ARG-SSR combinations. Note that in our dataset, time is positively associated with the amount of data used in the analysis (i.e., number of plasmids publicly available). Between the years 2000 and 2010 there is an increase of 24 plasmids per year, on average. The rate of data accumulation increases markedly between 2010-2020 to 145 plasmids per year, on average. Combining the temporal data with the

**Fig. 2 | Temporal network of ARG-SSR cooccurrence in plasmid resistance islands. A** Network of ARG and SSR gene families that coincide in resistance islands. The nodes correspond to gene families and the edges to cooccurrence events. ARG nodes are plotted in a bigger size. **B** Temporal assembly of nodes in the network according to the strain isolation date. The number of plasmids used in the network analysis is shown in a purple line. **C** Temporal assembly of edges connecting to ARG nodes in the network (**D**) Phylogenetic tree of IS*26*. The tree scale bar is in amino acid substitutions per site. Stacked bars at the tree leaves show the number of each variant in the gene family, where two dominant transposases, IS*26* (yellow) and IS*26*-v1 (blue), had 6266 and 1385 members, respectively. The stacked bars at tree tips are colored according to the predicted plasmid mobility class (see legend in Fig. 3). The histogram shows the isolation dates of plasmids carrying the two dominant transposases. Low frequency variants in the data (colored in gray) have not been verified hence we cannot rule out that they stem from sequencing errors. Temporal accumulation of (**E**) edges and (**F**) edge weights in the network (dots represent the weight medians, whiskers represent the weight ranges). Plasmids from 36 isolates sampled before the year 2000 were binned with the year 2000 isolates. Source data are provided as a Source Data file.

network components shows that the network size increases with time and the increase in the number of plasmids at a homogeneous rate (Fig. 2B). The number of new ARG nodes added to the network per year is increasing slowly and steadily over time. At the same time, we note that about half of the ARG nodes (32; 57%) are already present in the network with the early sampled strains. Between 2011 and 2022, seven ARGs are added to the network (Fig. 2C). These include resistance genes to more recently introduced antibiotics. Examples are *tet(X4)* that supplies resistance to tigecycline (first reported in 2019[38]), *mef(B)* (first reported in 2009[39]) and *mcr*−1.1 that was first reported in 2016[3], but may have been present in livestock associated *E. coli* already since the 1980s[40]. We note that *mcr*−1.1 in *E. coli* plasmids was so far reported to be mobilized as a single gene by IS*Apl*1[41,42] (hence, it is not expected to be found as a coARG). Nonetheless, this ARG is connected in our network with other ARGs due to a CSB that is conserved in four plasmids and includes multiple mobile genetic elements (Supplementary Fig. S2).

The number of SSR nodes has a similar trend, with a slight increase after 2010 (i.e., with the stark accumulation in sequenced plasmids; Fig. 2B), where it surpasses the increase in the number of ARGs. The increase in the number of SSRs may be related to diversification of the SSR gene families, and the increase in plasmid data is useful for revealing SSR gene variants. For example, IS*26* and IS*26*-v1[34] transposases comprise 6,266 and 1,385 identical protein sequences in our data, respectively. The two transposases are often present in the same replicon. The protein sequence of IS*26*-v1 transposase has a single G184N replacement, which was reported to increase the cointegration efficiency of IS*26*[35]. The two dominant IS*26* transposases correspond to 92% of the gene family members and the remaining variants have few substitutions. The high frequency of identical IS*26* homologs may correspond to either a very recently emerged variant or a highly conserved variant in the population. To gain further insight into the antiquity of the two main variants, we examined the isolation date of the isolates encoding those variants. The distribution along the isolation timeline as reflected in our data shows that IS*26* is first observed in an isolate from 1954 and IS*26*-v1 in an isolate from 1990. The higher frequency of ARG combinations with IS*26* in comparison to combinations with IS*26*-v1 (Fig. 2A) is in agreement with earlier observations of IS*26* in the three genera *KES*. The positive trend of increasing frequency in time for both IS*26* and IS*26*-v1 suggests that both ISs are highly mobile. Other common SSR gene families in our data (e.g., Tn*3*, IS*91*, and IS*110*) are characterized by a similar pattern of diversification and rapid increase in the frequency of evolved variants (Supplementary Fig. S3).

The accumulation of edges in the network supply hints for temporal aspects in the evolution of gene combinations in resistance islands. The number of edges is initially increasing with time and the number of sampled plasmids (Fig. 2E). From ca. 2014 onwards, the cumulative number of edges plateaus, suggesting that the repertoire of gene combinations in the network is close to saturation, despite a large number of plasmids added to the data. The higher frequency of ARG-SSR edges overall may be attributed to the larger number of possible ARG-SSR combinations in the data. The distribution of edge weight in the network shows that the increase in data from 2010 onwards, contributed mainly to an increase in the median edge weight (Fig. 2F), which corresponds to increasing frequencies of the observed gene cooccurrences in plasmid resistance islands. The distribution of edge weights indicates that the cooccurrence of ARG pairs has a higher abundance in the network compared to ARG-SSR and SSR-SSR pairs (Fig. 2F). Taken together, the network composition suggests that the repertoire of ARG combinations in plasmid resistance islands is diverse but changes only little over time. Based on these results we predict that additional sequencing may reveal additional instances of existing combinations but rarely uncover new cooccurring gene combinations (with the exception of newly emerging resistance genes and SSR gene variants).

## Scarce occurrence of resistance islands in small plasmid types

Which plasmid lineages are typical vehicles of multiple drug-resistance genes? Previous large-scale genomic studies in the literature mainly focus on the plasmid size and mobility class (e.g. refs. 19,22,43), thus neglecting the plasmid evolutionary history. Here we make use of a recently suggested classification scheme of plasmids into taxonomic units[25] and marker genes for the distinction between *KES* small and large plasmids (see Supplementary Note SN2). The small plasmid types are commonly smaller than 19Kb, they are either non-mobilizable (but see ref. 44) or mobilizable and mostly correspond to ColE1-like plasmids. ARGs are rarely found in the small plasmid types; of the total 1961 small plasmid genomes, only 7% harbor ARGs, with 72% of the small plasmids encoding a single ARG. Correspondingly, resistance islands are both small and rare in the small plasmid types (Fig. 3).

To further investigate the evolution of small resistance plasmids we examined the phylogenetic tree of Rop (plasmid primer RNA-binding protein; RefSeq accession: WP_003978814.1) and compared the presence of cooccurring ARGs and SSR genes among closely related plasmids. The Rop sequence is short and highly conserved, hence our ability to reconstruct a reliable phylogeny using all 903 members of that gene family is limited. Consequently, we split the Rop phylogenetic analysis into two plasmid groups: plasmids not carrying resistance genes that reside mostly in *Escherichia* and *Salmonella* (Supplementary Fig. S4), and plasmids residing mostly in *Klebsiella* (Fig. 4). The *Klebsiella* Rop phylogeny shows a clear split into two clades that correspond to Col440I (partially belongs to PTU-E71) and ColRNAI (PTU-E4) plasmids. The ColRNAI clade includes large plasmids from 21 isolates that correspond to cointegrates (of plasmid fusions; Supplementary Data 1) and small resistance plasmids in 27 isolates; in 24 of the isolates, the small plasmids encode one *aac(6')-Ib* in the same Tn*3*-based transposon previously reported as Tn*1331*Δ:IS*26* (Fig. 4)[45,46]. Comparing the plasmid sequences reveals that the plasmid backbones of this sub-clade are almost identical to its sister sub-clade, the only difference between plasmids in the two sub-clades in the presence of Tn*1331*Δ:IS*26* (Supplementary Fig. S5). The branching pattern of the clustered ColRNAI plasmids suggests that a single ancestral transposon insertion event in the ancestor of these plasmids is the most parsimonious scenario for the presence of Tn*1331*Δ:IS*26* in those plasmids. The ARG-carrying ColRNAI plasmids are reported in isolates from human sampled between 1974 and 2019, suggesting that this plasmid has a stable persistence in *Klebsiella*. The *Klebsiella* ColRNAI clade further includes three plasmids reported in *Escherichia* isolates, which likely correspond to plasmid transfer from *Klebsiella* to *Escherichia*. The larger plasmids in the ColRNAI clade are putatively

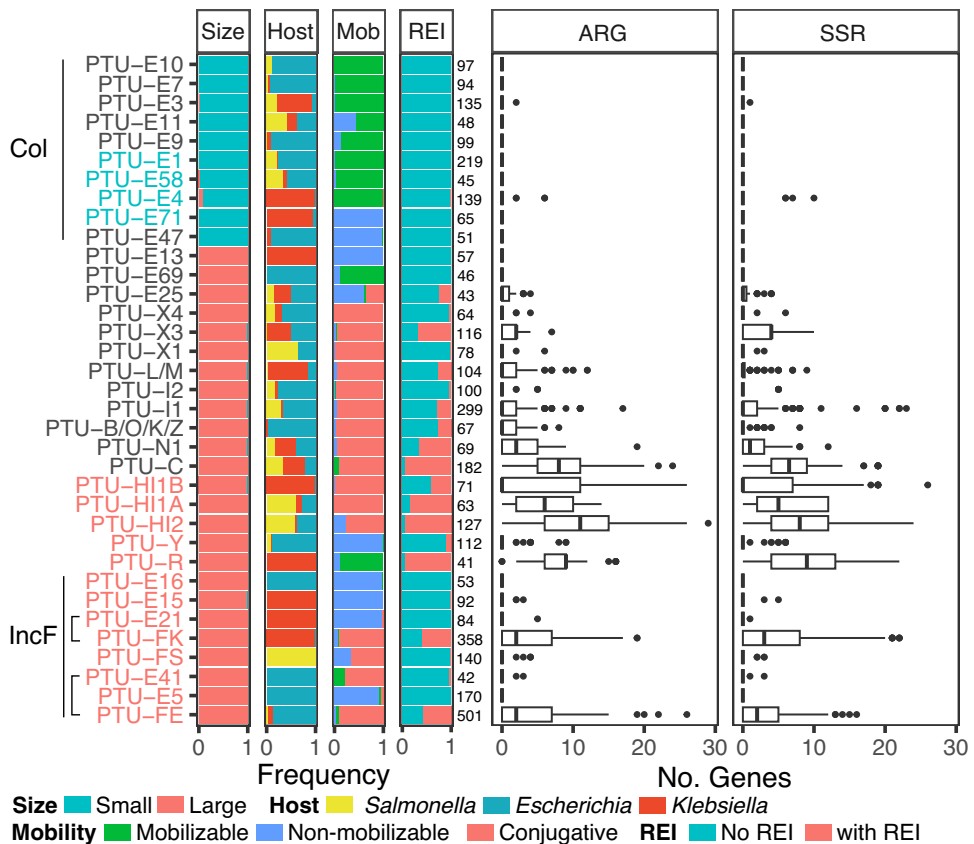

**Fig. 3 | A comparison of resistance islands among PTUs.** Stacked bar plots from left to right show plasmid properties in the corresponding PTU. Size: plasmid size categories (small plasmids here are <19Kb). Host: genus classification of the plasmid host. Mob: predicted plasmid mobility class. REI: plasmids encoding resistance islands (as identified here). The number of plasmids in each PTU is reported right of the stacked bar plots. Boxplots show the number of ARGs and SSR genes within the resistance island pieces (thick center lines represent the medians, box limits represent the upper and lower quantiles, whiskers represent within 1.5x inter-quartile range, points represent the outliers). PTUs belonging to the small-size Col plasmid group exhibit a limited occurrence of resistance islands. PTUs that include plasmids with a Rop protein are written in blue, while those including plasmids with a RepB protein are written in red. Source data are provided as a Source Data file.

mobilizable or conjugative; these plasmids are untypically large for Rop-encoding plasmids (see also Supplementary Data 1) and likely originated from plasmid fusion events. Taken together, resistance islands comprising multiple drug resistance genes are rare in small ColE1-like plasmids with only a few exceptions.

### ARG cooccurrence in resistance islands is restricted to specific large conjugative plasmid taxonomic units

Most of the plasmid ARG homologs (98%) in our data are in large plasmids. Nonetheless, only 38% of the large *KES* plasmids harbor cooccurring ARGs and SSR genes, hence a large plasmid size is not necessarily associated with the presence of resistance islands. Examples are plasmids classified in PTUs E5, E13, E15, E16, and E21, all of which are putatively non-mobilizable (Fig. 3). For example, only three non-mobilizable E15 plasmids harbor resistance islands that comprise multiple ARGs to ß-lactams and aminoglycosides; the plasmids were reported in *Klebsiella* isolates from different clinical studies (e.g. NZ_CP045677.1[47]). Indeed, the number of ARGs in MDR plasmids is associated with plasmid size[19]; however, we note that the proportion of ARGs in such large plasmids is rather low, with a mean of $5.5 \pm 4.7\%$ (±SD). Most of the ARG cooccurrence in resistance islands were observed in conjugative F-plasmids (IncF) that reside in *Escherichia* (PTU-FE) and *Klebsiella* (PTU-FK) hosts but not in *Salmonella* (PTU-FS) hosts. Notwithstanding, several classes of large conjugative plasmids, including PTUs E41, I2, X1, and X4 are rarely associated with resistance islands (Fig. 3). Taken together, ARG cooccurrence in resistance islands is common in specific taxonomic units of large conjugative plasmids.

To study the evolution of large MDR plasmid types we compared the patterns of cooccurring ARGs and SSR genes among closely related plasmids. For that purpose, we inferred the phylogeny of RepB (plasmid replication initiator protein; RefSeq accession: WP_004182030.1), which is the most commonly shared gene family among large conjugative plasmid types. RepB-encoding plasmids are the major carriers of ARGs, where 44% (1045/2392) of the RepB-encoding plasmids carry 68% (10,490/15,534) of the plasmid ARG homologs. The phylogeny reveals several RepB clades that correspond to recognized PTUs (and plasmid incompatibility groups) (Fig. 5A).

The comparison of related PTUs that differ in their mobility may supply hints for the association of resistance islands with specific plasmid characteristics. The RepB phylogeny indicates an evolutionary relatedness between PTUs FE, E5, and E41, which has been previously suggested based on plasmid sequence similarity[48] (note that 17 out of 42 E41 in our set are not included in the RepB phylogeny since they do not encode a RepB). Plasmids in the three PTUs commonly reside in *Escherichia* hosts, with rare occurrence in *Salmonella* or *Klebsiella*. PTU-FE plasmids are mostly conjugative[48] and abundantly encode resistance islands. In contrast, plasmids in the related PTUs E5 and E41 rarely encode ARGs, where PTU-E5 are non-mobilizable plasmids and PTU-E41 plasmids are mostly conjugative. While most of the PTU-FE plasmids are conjugative ($n = 462$), this group includes also rare mobilizable plasmids ($n = 22$) and non-mobilizable plasmids ($n = 17$). The presence of resistance islands among PTU-FE plasmids is significantly different among plasmids depending on the plasmid mobility class ($P = 0.03$, using Fisher test), where the proportion of plasmids

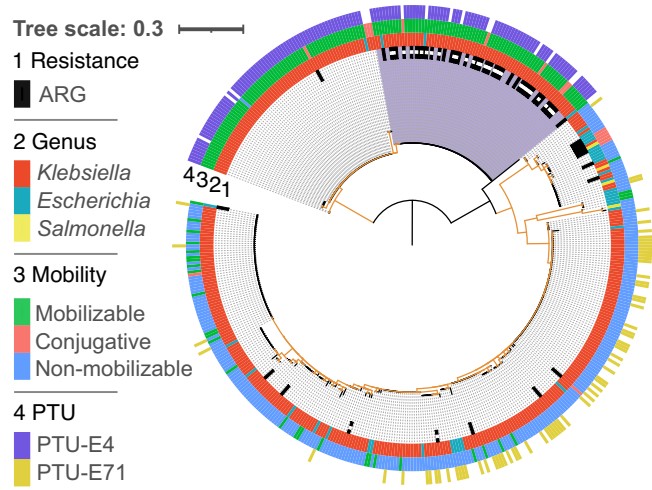

Tree scale: 0.3

**1 Resistance**
ARG

**2 Genus**
*Klebsiella*
*Escherichia*
*Salmonella*

**3 Mobility**
Mobilizable
Conjugative
Non-mobilizable

**4 PTU**
PTU-E4
PTU-E71

**Fig. 4 | Phylogenetic tree of 469 Rop sequences in 443 plasmids.** Rings of the phylogenetic tree show (from inner to the outer ring): the presence of ARGs, host genus, and PTU. The majority of plasmids reside in *Klebsiella* isolates. The Rop phylogeny constitutes two main Rop clades that correspond to non-mobilizable ColRNAI (PTU-E4) plasmids and mobilizable Col440I (PTU-E71) plasmids. A sub-clade of ColRNAI plasmids harboring ARGs is marked in purple, in that clade, small plasmids carrying Tn*1331*Δ:IS*26* are marked by a white square within the ARG ring. Note that while the isolates in our data are not identical, the plasmid variants presented here may correspond to the same plasmid origin, hence the frequency of specific plasmid types reflects the number of isolates where they have been observed. Branches with bootstrap support ≤70 are colored in orange. The tree scale bar is in amino acid substitutions per site. Plasmids lacking a PTU assignment could not be classified using COPLA[27]. Source data are provided in supplementary file.

encoding resistance islands is lower among the conjugative plasmids (57%) compared to mobilizable (77%) and non-mobilizable (76%) plasmids. Furthermore, there is no significant difference in the number of ARGs among the three mobility classes in PTU-FE plasmids encoding resistance islands ($P = 0.43$, using the Kruskal–Wallis test). The comparison among the related PTUs FE, E5, and E41 indicates that the presence of resistance islands in plasmids is PTU-specific and furthermore that conjugative plasmids are not necessarily enriched for resistance islands.

The RepB phylogeny further suggests an evolutionary relatedness between PTU-FK and PTU-E21 plasmids that commonly reside in *Klebsiella* hosts. Plasmids in PTU-FK are mostly conjugative and abundantly encode resistance islands, while PTU-E21 are mostly non-mobilizable and rarely encode ARGs (Fig. 3; Fig. 5A). The RepB phylogeny supports the classification of plasmids in PTU-FS as closely related plasmids (Fig. 5A). Plasmids in that group reside in *Salmonella* hosts and include both conjugative and non-mobilizable plasmids that rarely encode antibiotic resistance genes. There is no significant difference in the occurrence of resistance islands between conjugative and non-mobilizable plasmids in that group ($P = 0.66$, using the Fisher test). The comparison of plasmids in these PTUs shows that differences in the frequency of resistance islands depending on the plasmid mobility class are PTU-specific.

Large conjugative plasmids in PTUs HI1A, HI1B, and HI2 have been suggested to have shared plasmid sequence similarity[25], and the RepB phylogeny indicates that PTU-HI1A and PTU-HI2 are more closely related (Fig. 5A). This inference aligns well with the difference in their host range as reflected in the isolate genomes; HI1B plasmids reside in *Klebsiella*, while HI1A and HI2 plasmids reside in *Escherichia* and *Salmonella* (Figs. 3 and 5A). The split between HI1B and the other two PTUs may thus be linked to plasmid divergence during the evolution of host-specificity. Further comparison among the plasmids depending

on their host genera shows that there is no significant difference in the presence of resistance islands between HI2 plasmids depending on their *Escherichia* or *Salmonella* host ($P = 0.24$, using Fisher test), while the proportion of HI1A plasmids encoding resistance islands is larger in plasmids that reside in *Salmonella* compared to those that reside in *Escherichia* ($P < 0.01$, using Fisher test). Hence the presence of resistance islands in specific PTUs may depend additionally on the plasmid host range.

The association of resistance islands with specific plasmid types (PTUs), and in some cases also the plasmid host range, is in agreement with previous studies showing that closely related plasmids have a shared resistance island content (e.g. ref. 49). To examine this aspect of MDR plasmid evolution we focus on the five prominent MDR plasmid PTUs among the RepB plasmids. The pattern of shared plasmid genes is in accordance with the plasmid PTU classification, and also the expected patterns of their relatedness as inferred by the RepB phylogeny (Fig. 5B). That is, the plasmids are generally clustered by PTU with an overlap between the clusters of HI1A and HI2 plasmids. Plasmids in PTU-FE form a tight cluster that supports their common evolutionary history. In contrary, the PTU-FK plasmids form a rather sparse cloud, indicating a high diversity of plasmids in this PTU, and likely a combination of highly diversified plasmid sub-groups. The proportion of shared genes between the plasmids is weakly positively associated with the proportion of shared CSBs in PTUs HI1A and HI1B ($r_{HI1A} = 0.67$, $r_{HI1B} = 0.68$, $P < 0.01$, using spearman correlation), but less so for the other examined PTUs ($r_{HI2} = 0.2$, $r_{FE} = 0.21$, $r_{FK} = 0.33$, $P < 0.01$, using spearman correlation). Examination of shared CSBs among plasmids in the five PTUs shows that most of the plasmids form a dense cluster, indicating common CSB content across the five PTUs (Fig. 5C). Plasmids in PTUs FE and FK form two additional clusters, indicating that they commonly share CSBs with each other, but not with the other plasmids in the main cluster (Fig. 5C). The pattern of shared CSBs among the five PTUs suggests that resistance islands have a narrow distribution in specific PTUs.

The CSBs that are shared among plasmids in the five PTUs are rather short, with a median CSB length of four genes. Example is a CSB that was inferred in 313 plasmids in the five PTUs and comprises two ARG families: aminoglycoside nucleotidyltransferase (ANT(3″)) and MFS efflux pump (QacE) (see CSB_F_1 in Fig. 6); both of these families in our dataset include multiple epidemiological variants of the ARGs. This short CSB is included in multiple CSBs that correspond to previously recognized MGEs. This includes a CSB similar to a class 1 integron previously reported to be associated with *sul3*[50] (CSB_HI2_3 in Fig. 6) that has a narrow distribution in PTU-HI2 plasmids and a CSB similar to a class 1 integron previously reported in Tn*6450*[51] (CSB_FK_3 in Fig. 6) that has a narrow distribution in PTU-FK plasmids. Another overlapping CSB includes an ARG cassette previously reported in class 1 integron[52] (CSB_FE_3 in Fig. 6) and has a narrow distribution in PTU-FE plasmids. Several CSBs reported in Fig. 6 are shared among plasmids classified into different PTUs. An example is the CSB_HI2_2 (Fig. 6) that we detected in one plasmid classified as PTU-FK. The plasmid pLH94-8 was reported in a *K. pneumonia* isolate sampled from feces of a healthy human[53]. The pLH94-8 plasmid locus of this CSB is almost identical to several other *E. coli* plasmids harboring the same CSB (e.g., pSLK172-1[54]). Another example is CSB_FK_3 (Fig. 6) which we detected in a plasmid classified as PTU-HI1A. The plasmid p24362-1 was reported in a *S. enterica* strain isolated from swine[55]. The locus corresponding to the CSB is identical to other PTU-FK plasmids (e.g., pDA33144-220[56]). The demonstrative example presented here thus suggests that CSBs in our dataset are largely PTU specific.

To test if PTU specificity is a general trend in our dataset, we examined the diversity of plasmids harboring the same CSB using the Gini-Simpson (*GS*) index[57]. This index measures the probability

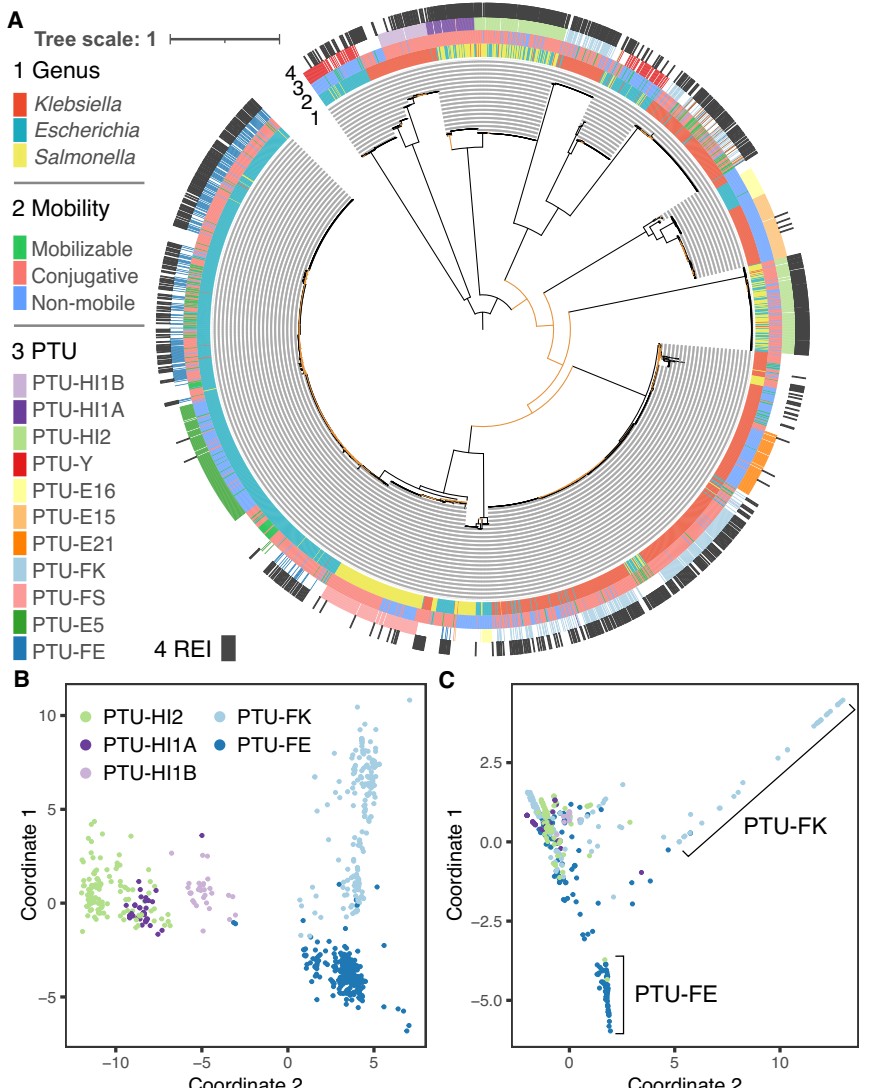

**Fig. 5 | Resistance islands in large plasmids. A** Phylogenetic tree of 2529 RepB sequences in 2354 plasmids. Rings of the phylogenetic tree show (from inner to the outer ring): host genus, predicted plasmid mobility class, PTU and the presence of resistance islands. Branches with bootstrap support ≤70 are colored in orange. The tree scale bar is in amino acid substitutions per site. **B** Multidimensional scaling of shared gene content among plasmids. Plasmids are shown in different colors for each PTU. **C** Multidimensional scaling of shared CSBs among plasmids. Plasmids are shown in the same colors for each PTU. Plasmids lacking a PTU assignment could not be classified using COPLA[27]. Source data are provided in Supplementary file.

that two randomly selected plasmids harboring the same CSB are classified into different PTUs. Note that the CSBs appearing to have a narrow PTU distribution in the demonstrative example are characterized by a low *GS* index (Fig. 6). Our result revealed that the median *GS* index for all CSBs and all PTUs is 0. Furthermore, the majority of CSBs (59%) in our set have a *GS* index ≤0.1. The PTU diversity of CSBs is significantly negatively correlated with the CSB length ($r_s = -0.66$; $P < 0.001$), with longer CSBs having a lower PTU diversity (Supplementary Fig. S6). Taking together, our demonstrative example shows that CSBs matching to previously documented MGEs have a rather narrow distribution in specific PTUs, and furthermore the CBSs we analyzed here are characterized by a low diversity of PTUs. Both lines of evidence indicate that resistance islands are mostly shared among closely related plasmids having similar plasmid properties.

## Discussion

Microbial organisms encounter diverse challenges in their habitat including predation, infection, as well as extreme or toxic

environmental conditions. The response to such stressors is diverse and range from plastic phenotypes to defensive multicellular structures (e.g. ref. 58). Furthermore, adaptive response in prokaryotes can be mediated by the acquisition of new functions via gene transfer, i.e., in response to phage infection[59] or to antibiotics[52]. In both of these cases, the acquired genes are typically organized in dense genomic islands[60,61]. Our results supply a quantitative assessment, according to which, the majority of antibiotic resistance genes in plasmids are located adjacent to mobile genetic elements in resistance islands. The remaining ARGs in MDR plasmids may still be located in resistance islands, however, if these are rarely found in our dataset, the coARG instance would be excluded from the final results. We note that filtering the coARGs for CSBs appearing in ≥4 strains lead to exclusion of 10-15% of the plasmid coARGs (Supplementary Data 3). In agreement with previous studies, our results implicate the interaction between mobile genetic elements and plasmids as the major driving force in the evolution of ARG content in MDR plasmids. Resistance islands are hotspots for the integration of ARG-carrying MGEs due to several reasons, including functional

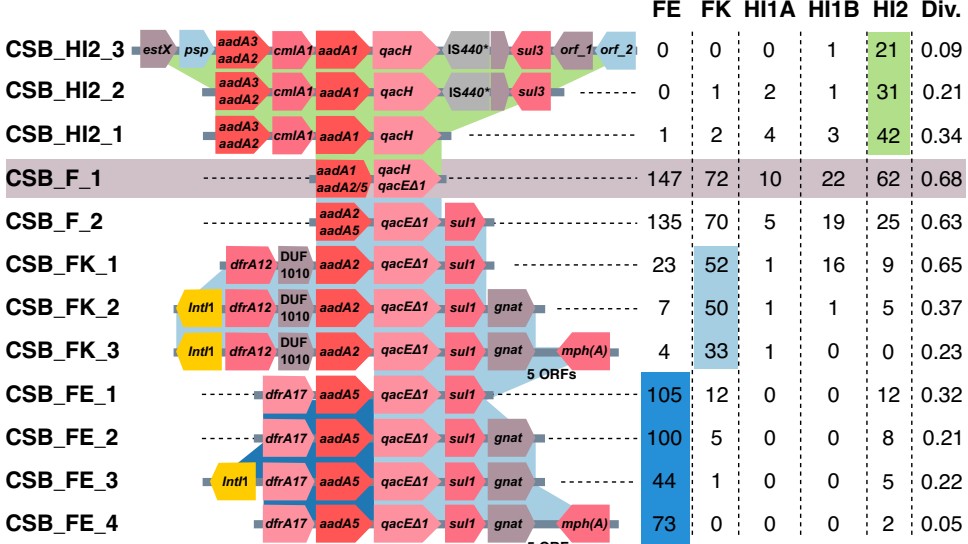

**Fig. 6 | The distribution of CSBs matching mobile genetic elements is biased towards specific PTUs.** A demonstrative example of overlapping CSBs and their frequency in plasmids classified into five closely related PTUs. Protein-coding genes in the CSBs are shown as arrows. ARGs in the same gene family have the same color. Genes with no name correspond to hypothetical proteins. orf_1 corresponds to GrpB family protein. orf_2 corresponds to SDR family oxidoreductase. The adjacent table shows the number of plasmids in each PTU where an instance of the CSB was found. Cells showing the dominant PTU are shaded. The CSB diversity (Div.) was calculated using the Gini-Simpson index. Details of the presented CSBs are listed in Supplementary Data 6. *IS440 is annotated as a pseudogene in the genome versions analyzed here (in more recent versions this locus is annotated as a gene). Source data are provided in the full CSB dataset.

dependencies between specific SSR families (e.g., Tn402 and Tn21 family[62]) and possibly also deleterious fitness effect of MGE integration outside resistance islands on the plasmid functions (e.g., in essential plasmid loci[63]). Plasmids carrying mobile genetic elements are therefore, in themselves, a hotspot for the acquisition of multiple drug resistance genes (e.g., as shown for an IncP-1 plasmid[49]).

The temporal analysis of resistance islands shows that ARG combinations tend to be rapidly saturated. In the last decade, the number of available plasmid sequences in genomic databases increased considerably, yet the frequency of ARG combinations was growing slower (Fig. 2B). More than half of the ARGs are already found in resistance islands in the earliest sampled isolates. The temporal integration of ARG nodes in the network largely fits earlier reports on the emergence of specific resistance genes, providing validation for our approach. An example is the high connectivity of sul1 that supplies resistance to sulfonamides introduced in the 1930's[64]. Additionally, recently reported resistance genes appear in the network periphery. One example is mcr-1.1 that supplies resistance to polymyxin and colistin, and was reported in 2016 in Escherichia isolated from pig feces[3,40] (Fig. 2C). Another example is tet(X4) that was reported in 2019; the tet(X4) gene supplies resistance to tigecycline, a last resort antibiotic that was introduced in 2005. The gene was observed in plasmids of Escherichia isolates sampled already during 2016-2018 from a pig farm as well as human patients[38]. The newly added ARG nodes increase in connectivity rather fast, indicating that the agglomeration of ARGs in resistance islands is a rapid process.

Previous studies on the evolution of MDR plasmids typically compare plasmid ARG content in the context of plasmid size and mobility class, without much distinction of the plasmid origins or properties related to the host range. A recent classification of plasmids into taxonomic units supplies a framework for comparative plasmid studies[25] while considering their distinct characteristics and origins. The results we present here add support for the evolutionary relatedness of plasmids in the same PTU and furthermore expand on the evolution of ARG content in specific plasmid lineages. We further show that the distribution of resistance islands has a narrow PTU range, and their differential presence among plasmids within the same PTU may

furthermore depend on the plasmid mobility class or the host range. Quantifying the diversity of plasmids harboring the same CSBs further revealed a low PTU diversity of CSBs. However, an interpretation of that result in the context of plasmid evolution is challenging, since our approach does not allow for an accurate annotation of resistance island boundaries. Long CSBs in our dataset correspond to conserved neighborhoods of MGE agglomerates, that are unlikely to emerge independently in distantly related plasmids. In contrast, short CSBs in our dataset may correspond either to genuine transposable elements or degraded pieces of resistance islands. Notably, while we are unable to distinguish between these two possibilities, our analysis shows that the plasmid diversity is generally low for all CSB lengths. The pattern of shared resistance islands can be thus explained either by common origins of the islands in ancestral plasmids and evolution by plasmid divergence (i.e., vertical plasmid inheritance), or alternatively, barriers for horizontal transfer of ARGs between plasmids, which are associated with the specific characteristics of plasmids in the same PTUs.

Barriers for the integration of ARG-carrying MGEs into plasmids may be related to the plasmid genetic properties (e.g., mode of replication and mobility), ecology (i.e., host range) and evolutionary history. For example, small plasmid types are almost devoid of ARGs; in our dataset, 62% of the plasmids in small plasmid PTUs are documented in a host where a large plasmid encoding resistance island is found as well. Hence the rarity of coARG containing CSBs in small plasmid types cannot be well explained by plasmid ecology (or host range) only. The narrow distribution of plasmid size among the ColE1-like plasmids in KES hosts (see also[20] and Supplementary Note SN2) may indicate that ColE1-like replicons cannot well support a plasmid size inflation, hence the barrier for ARG acquisition is linked to the plasmid replicon type. For example, it has been suggested that the replication mode of IncQ plasmids, which are typically small similarly to ColE1-like plasmids, may become unstable if the plasmid size surpasses ca. 20 Kb[65]. In contrast, barriers for MGE-mediated integration of ARGs in the large PTU-FS plasmids (where only 5 out of 114 plasmids had resistance islands; Fig. 3), are unlikely to be related to plasmid size. An examination of their hosts showed that only 21 (18%) PTU-FS plasmids reside with other plasmids that encode a resistance island in the

same host, while the remaining FS plasmids are either a sole plasmid in their host (54%) or reside with other plasmids that do not encode resistance islands. The FS taxonomic unit includes homologs of plasmid pSLT, a virulence plasmid in *Salmonella*[66] where ARGs are rather rare (e.g.[67]). The interaction of PTU-FS plasmids with ARG-encoding MGEs is likely rare due to the plasmid host range. Taken together, our study reveals that the interaction between plasmids and MGEs is a major driving force in the evolution of MDR plasmids, where the interaction and its frequency largely depend on the plasmid taxonomic units, that is, the plasmid characteristics and its evolutionary history.

## Methods

### Data collection

Complete genomes of isolates in the genera *Escherichia*, *Salmonella* and *Klebsiella* where plasmids are reported were downloaded from the NCBI RefSeq database (version 01/2021; the same sequenced genomes dataset was used in our recent publication[37]). Isolate metadata was downloaded from BioSample database. The metadata of samples lacking host information was examined manually in detail. A total of 23 strains sequenced within the framework of laboratory experiments were excluded. The surveyed genomes include 1114 strains comprising 3098 plasmids from *Escherichia*, 755 strains comprising 2693 plasmids from *Klebsiella*, and 572 strains comprising 993 plasmids from *Salmonella* (Supplementary Data 1). The dataset does not include redundant genomes of the same strain. Similar and even identical plasmids were retained if they were identified in different strains (see Supplementary Data 1).

### Identification of ARGs

Homologs of antibiotic resistance genes were identified based on the comprehensive antibiotic resistance database[28] (CARD version 3.1.0) using Resistance Gene Identifier (version 5.1.1, with parameter --clean). To reduce the number of mis-assembled or mis-annotated ARGs, we excluded protein-coding genes whose sequence had less than 70% identical amino acids to the CARD reference ARG, or their length was less than 90% or more than 130% of the CARD reference ARG (as in ref.[37]). Of the total 3044 genes in CARD, 416 (14%) ARGs had 114,464 homologs in the examined isolates.

### Construction of homologous gene families

Gene families were inferred based on sequence similarity as previously described in our study of *Escherichia* strains[19]. Briefly, reciprocal best hits (RBHs) of protein sequences between all pairs of 9225 replicons were identified using MMseqs2[68] (v.13.45111, with module easy-rbh applying a threshold of $E$-value $\leq 1 \times 10^{-10}$). RBHs were further compared by global alignment of the protein sequences using parasail-python[69] (v. 1.2.4, with the Needleman-Wunsch algorithm). Sequence pairs with ≥30% identical amino acids in the global alignment were clustered into gene families using a high-performance parallel implementation of the Markov clustering algorithm[70] (HipMCL with parameter --abc -I 2.0). A total of 114,457 ARGs were identified in 138 gene families; the CARD classification was sometime only partially in agreement with the homologous gene family classification (114,457/203,591). Most of the contradictions between CARD and our clusters were chromosomal gene (97%) with only 3% corresponding to plasmid genes. Aiming for a conservative analysis, members not identified as ARGs by CARD were excluded.

### Transposases and site-specific recombinases identification and biased distribution test

All gene families constructed in this study were scanned using InterProScan[71] (version 5.59-91.0) against InterPro's member databases[72]. HMM profiles of transposases and site-specific recombinases were obtained from previous studies[73,74] to search SSRs using hmmsearch (HMMER version 3.3.2)[75]. Three chromosomal gene families were not included in the further analysis: a serine recombinase (flagellar phase variation DNA invertase Hin) and tyrosine recombinases XerC and XerD. The biased distribution of gene families towards plasmids or chromosomes was tested by comparing their frequency in both replicon types with the frequency of remaining family members on both replicon types using Fisher's exact test (as in ref. [37]), implemented with the fisher.test function in R (version 4.0.3, two-sided α = 0.05 and correction for multiple comparisons with false discovery rate (FDR[76])).

### Gene cooccurrence test

Significant cooccurrence of ARG pairs from different gene families was tested using Fisher's exact test with fisher.test function in R (version 4.0.3, two-sided, α = 0.05) and correction for multiple comparisons with false discovery rate (FDR) [76]. The test was performed for plasmids in different genera separately (as in ref. [37]). For each test, the sample size $n$ was equal to the number of ARG-coding plasmids. The significantly cooccurring ARGs are supplied in Supplementary Data 5.

### Inference of collinear syntenic blocks

Collinear syntenic blocks (i.e., conserved gene order) were identified using CSBFinder-S[29] allowing no gene insertion and maximal length of 200 protein-coding sequences (version 0.6.3 with parameter -q 4 -ins 0 -lmax 200 -c -cs -alg MATCH_POINTS). CBSs that were found identical in at least four genomes were retained for further analysis.

### Detection of CSBs matching to mobile genetic elements

Sequences of intact transposable elements were collected from three mobile genetic element (MGE) databases: The Transposon Registry[30] (version 10/2021), TnCentral[31] (version 10/2021), and ISfinder[32] database (version 10/2020). Partial sequences and complete plasmid sequences were excluded. We searched for the known MGEs in all 2441 complete genomes using Blast+[77] (version: 2.9.0+ parameter: $E$-value $\leq 1 \times 10^{-10}$). The blast hits were further compared with the CSBs genomic loci (i.e., the CSB coordinates). CSBs that overlapped with identified MGEs with at least two complete protein-coding genes were retained for further analysis.

### ARG island analysis

The distance between ARGs was counted as the number of protein-coding genes on the same strand from one ARG to the adjacent $n$-th ARG ($n = 2$–10). The stochasticity of ARG distance was tested using chi square goodness-of-fit test with negative binomial distribution as the null hypothesis. The distances between randomly sampled genes was used as a control, where a similar number of gene families was sampled. Each sampled gene family has a similar family size to the corresponding ARG family. The distance between members of the randomly sampled gene families was counted in the same way as for the ARGs, the sampling was repeated 100 times. The distance distribution of ARGs and the control group were compared using Wilcoxon rank-sum test for each step-size separately ($n = 2$–10).

### Network visualization

The network visualization was performed with Cytoscape[78] (version 3.10.1).

### Plasmid typing

We identified the incompatibility (Inc) groups of Rop- and RepB-coding plasmids using PlasmidFinder (version 2.1.6 with parameters: -l 0.6 -t 0.9 -mp blastn). Plasmids that remained of unknow Inc groups were further examined using pMLST[79] (version 2.0.3, with parameter -mp blastn). Plasmid classification into predicted mobility classes (non-mobile, mobile, or conjugative) was done on the basis of the plasmid gene content using MOB-suite[80] (version 3.0.3). Plasmids were assigned into PTUs using COPLA[27] (version 1.0).

## Phylogenetic analysis

Protein sequences were aligned using MAFFT[81] (version v7.475). Owing to the large number of identical protein sequences among transposases, the sequences were filtered into non-redundant amino-acid sequences prior to the alignment. Maximum likelihood trees were reconstructed using IQ-TREE[82] (version 2.1.2) with Le and Gascuel (LG[83]) substitution model and the LG4X model as additional alternative (with parameter -mset LG -madd LG4X -bb 1000). The resulting trees were rooted using the midpoint criterion and visualized using iTOL[84] (version 6.5.8).

## Diversity index

The plasmid diversity of CSBs was calculated according to the Gini-Simpson index[57] with vegan package[85] (v2.6-4) in R. The number of plasmids in each PTU was counted while excluding duplicated CSB instances in the same plasmid.

## Reporting summary

Further information on research design is available in the Nature Portfolio Reporting Summary linked to this article.

## Data availability

Source data are provided with this paper. The genome data is publicly available (https://ftp.ncbi.nlm.nih.gov/genomes/refseq/). Antibiotic resistance genes in comprehensive antibiotic resistance database (CARD v3.1.0; https://card.mcmaster.ca/); intact transposable elements were collected from The Transposon Registry (v10/2021; https://transposon.lstmed.ac.uk/tn-registry), TnCentral (v10/2021; https://tncentral.ncc.unesp.br/), and ISfinder database (v10/2020; https://isfinder.biotoul.fr/). The full CSB dataset and gene family information used in this study is available in Figshare[86]. Phylogenetic data of five transposase gene families, Rop, and RepB are also supplied[86]. Source data are provided with this paper.

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

## Acknowledgements

The authors thank Hinrich Schulenburg, Giddy Landan, Nils Hülter, Devani Picazo, Dustin Hanke, Ishan Bhatt for fruitful discussion and critical comments on the manuscript. We thank Liam Shaw for constructive comments and critical review of the manuscript earlier versions. We are grateful for financial support from the German Science Foundation (RTG 2501 TransEvo), the China Scholarship Council (CSC scholarship to Y.W.), and the European Research Commission (pMolEvol, grant number 101043835). This research was supported in part through high-performance computing resources available at the Kiel University Computing Centre.

## Author contributions

Y.W. and T.D. conceived the study. Y.W. designed and performed the data analysis and visualizations. Y.W. and T.D. interpreted the results and wrote the manuscript.

## Funding

## Competing interests

The authors declare no competing interests.
