## [Peer Review File · Nature Communications]

Reviewers' comments:

Reviewer #1 (Remarks to the Author):

This manuscript describes large-scale analysis of available plasmids in three species of Enterobacteriaceae, looking for co-occurrence of antibiotic resistance genes (ARG) with each other, whether these are associated with plasmid or the chromosome, whether ARG are clustered in “colinear syntenic blocks” (CBS), and associations with transposable elements and small or large plasmids. Any such analysis of large datasets needs to take into account the considerable existing literature/knowledge in this area to design the approach, produce meaningful results and inform conclusions. Many of the findings are already well known in the AMR plasmid field e.g., the title “Transposable elements drive the evolution of multiple drug resistance plasmids” does not state anything new. Lack of detail and problems with wording in places also make it difficult to understand the reason behind some parts of the analysis, exactly what was done, what is being proposed and why.

To very briefly summarize knowledge in this area, plasmid-borne ARG in the species examined here have been captured from the chromosomes of various bacteria and inserted into plasmids by the actions of various mobile genetic elements, including IS and Tn mentioned in this manuscript. Different mobile genetic elements capture and move ARG by different mechanisms. Capture events appear rare, and each AMR gene is generally intimately associated with one particular mobile genetic element, although others (e.g., IS26) can take over to mobilise ARG after the initial capture event (or subsequent movement can occur by homologous recombination).

ARG are known to cluster in regions which have been given various names e.g., antimicrobial resistance islands or multi-resistance regions (MRR), as mentioned on Lines 56-7, but also multi-drug resistance (MDR) regions, complex resistance loci (CRL) etc. This is likely to be at least partly because interruptions in important plasmids genes will be detrimental and an insertion in a place that does not interfere with plasmid functions can act as a ‘founder element’ into which other segments can be safely inserted. In addition, some transposable elements are known to target particular sequences e.g., class 1 integrons are derivatives of ‘res site hunter’ transposons that target transposons in the Tn21 subfamily of the Tn3 transposon family. There are other known associations e.g., certain ARG are found as gene cassettes, which are almost always found inserted in integrons; ISCR1 and associated ARG genes are almost always found in the 3'-conserved segment (3'-CS) of class 1 integrons. There are many publications and reviews on these aspects of mobile ARG, see e.g., Refs. 7 and 11 cited here and the papers that they cite.

Major overall points

a) It is not always clear in this manuscript what is meant by a “transposable element” - a defined element such as a particular IS or a transposon? Or a region containing multiple ARG and IS etc that is proposed to transpose? If the latter is meant, no direct evidence appears to be presented that any “tARGs” are transposable – presence of transposases in a region does not mean that it will be able to transposable as a single entity.

b) Biases in databases

Various places in the text describe analysis relating to dates of samples (lines 219-31, 246-51, 259-66, Fig. 2, Fig. S4, Fig. S5). Line 221 acknowledges that this approach relies on sampling density, but Table S1

shows that only ~300 sequences are available before 2000, with >6,500 from 2000 onwards. This needs to be clearly acknowledged (a hint of this is in the Discussion, Line 464, which refers to only 2000-2020) and likely limits any meaningful conclusions that can be drawn from these parts of the analysis. Biases due to increased sequencing of e.g., plasmids carrying a particular ARG gene, isolates of a particular sequence type (ST) known to carry a particular plasmid are also not considered – see comments below.

Comments on specific sections.

1) Abstract - see also comments above and below.

Lines 23-4 - I would disagree with this. There is a lot of available information on how different mobile genetic elements capture and move ARG and in some cases defined interactions between these elements.

Lines 24-9 – it is not clear how it was determined that these “tARGs” are capable of transposing – no clear, direct evidence of transposition is presented.

Lines 30-2 – this is already well known.

2) ARG coincidence Lines 74-127, Fig. 1B, Table 1

This section describes identifying ARG on the basis of similarity to proteins in the CARD database (which is very inclusive) and looking for patterns of coincidence (“coARGS”) and whether these are more frequently seen on the chromosome or on plasmids.

The ARGs were classified according to mechanism of action and the coARG results are summarised in Fig. 1B according to this classification. It is not clear why ARGs are grouped in this way or what this is expected to show. ARGs with the same mechanism may have been mobilised by different transposable elements and this is one factor that may affect which ARGs co-occur.

Lines 91-3 – it would be better to exclude intrinsic chromosomal ARG from this analysis, as they will behave differently from mobile ARG. The wording here makes the sentence difficult to follow.

Lines 95-6 state that the most frequent coARGS on plasmids are among pairs of genes encoding AMR inactivation mechanisms, but these are also the most common type of plasmid-borne ARG.

Lines 103-4 – it is not clear what is meant here.

The section on lines 112-27 is confusing and does not take into account published information. The E.coli chromosomal ampC gene doesn't seem to have been mobilised onto plasmids, so it is not surprising that it is not found with blaCTX-M-15 on plasmids. It is presumably not found on plasmids with other ARG genes either? Is the point that the “antagonistic” pairs confer resistance to the same antibiotics? If so, this is not clear. Lines 120-2 - the way this is worded make it difficult to understand which pairs of genes are considered “antagonistic”.

3) Results – coARGS and TE, Lines 142-89

This section describes identification of collinear syntenic blocks (CSB) containing coARGS.

Lines 148-9 – this wording is unclear – coARGS in CBS are more frequent on plasmids than the chromosome? Or coARGS in chromosomes are not in CBS?

Line 151-2 – this is as expected and there are many publications about this. Ref 9 refers to a single example of mobilisation. Various reviews cited here would make this point much more strongly.

Lines 153, 195, 165, 184 – this is very unclear – what is meant by a CSB “corresponding” to a transposable element (TE)? That some CBS contain TE? Having one or more transposases in a region does not mean that the region will be transposable as a whole - mobile genetic elements can have

different types of relationships with nearby ARG.

Lines 160-2 – it is not clear what is meant here.

Lines 164-5 – it is already known that ARG in plasmids are often clustered (Lines 56-7, antibiotic resistance islands/MRR).

Lines 180-1 – IS26 is only one particular example. See general comments above.

Lines 186-8 – except for tet(A) this combination of genes, is found in structure called Tn6029 (see e.g., doi:10.1089/mdr.2010.0042, named before recent publications on IS26 pseudocomposite transposons). aph(3'')-Ib is also known by the simpler name strA and aph(6)-Id as strB. These genes are found next to each other, in fact overlapping slightly, in a Tn3-family transposon called Tn5393, fragments of which are common in MRR. Lines 188-9 – again this is already well known, as acknowledged in the Introduction, Lines 56-7.

4) Results - resistance islands in plasmids, Lines 190-231

Lines 191-2 – this now says that coARGs are “in” transposable elements, but again it is not clear what is meant.

Line 192 – it is well known that ARG form such clusters, as acknowledged here on Lines 56-7.

Lines 196-206 - this is not surprising if knowledge on mobile ARG is considered - ARG are known to occur next to one another for various reasons e.g., strAB as mentioned above, gene cassettes in cassette arrays in class 1 integrons, gene cassettes next to qacED1 (probably called ermE here) and sul1 in the 3'-CS of class 1 integrons, ARG associated with CR1, almost always seen in the 3'-CS.

Lines 205-6 – again, this is already well known, as acknowledged in the Introduction, Lines 56-7.

Lines 207-10 – there are not “transposase families” - IS26 and IS6100 are both part of the IS6 family, and IS91 and IS110 are IS families. The IS are not “transposons that comprise ARGs” – the original definition of an IS is that it carries only the gene(s) needed for its own transposition. Although some IS are known to carry passenger genes, as far as I know ARG have not been found as passengers. ARG are usually mobilised by adjacent IS either individually (e.g., IS26, ISCR) or in pairs in conventional composite transposons, although none of the elements mentioned here are known to form these. Tn3-family transposons are the only elements that do contain ARG.

Lines 208-9 – an IS or a Tn (or transposase) found near an ARG may not have had anything to do with movement of this ARG or clustering – it could be a simple insertion.

Line 211 – it is not clear why these particular 24 ARGs are now being analysed.

Line 213 – what is the relationship of the IS26 to the 15 ARGs? Is there evidence that it has mobilized / could mobilize them?

Line 216 – sul1 and ermE/qacED1 are found in the 3'-CS of class 1 integrons not “in transposase families”.

Line 218-9 – if fragments of mobile genetic elements are annotated then it is evident that at least part of the capturing element is often retained – for example, blaTEM genes were captured as part of Tn3 sub-family transposons (Tn1/Tn2/Tn3) and are always still in this context, even if part of the Tn is lost.

Lines 220-31 – see comment (b) above.

Line 223 – which Tn3 element? What is the reference/accession for the date of the first isolate? The Tn3 family includes many different transposons and is divided into the Tn3 and Tn21 subfamilies, which have different organisation and different properties in relation to ARG carriage. Class 1 integrons target members of the Tn21 subfamily and ermE/qacED1 is in the 3'-CS of class 1 integrons.

Line 224 – ISCR elements are related to the IS91 family and sul2 is known to be associated with ISCR2.

Line 228 – if the IS110 family elements being referred to are IS4321 and IS5075 (Fig. S5) then these are known to target the terminal inverted repeats of Tn21 subfamily elements, where they likely prevent further transposition. Tn21-like elements carry class 1 integrons and *dfrA12* is in a gene cassette found in integrons.

Line 229 – *catI* is not the correct name for this gene – it is probably *catA1*, again known to be associated with Tn21 in sequences from back as far as the 1950s (see doi: 10.1128/MMBR.63.3.507-522.1999).

5) Evolution of transposases - Lines 232-280

The part of the manuscript looks at evolution within IS26 and Tn3 family transposases. It's not clear to me why this analysis was carried out or what it would be expected to show.

Line 238 – the most recently assigned name, i.e., IS26-v1 not IS26*, should be used throughout – it is part of a scheme to consistently indicate all IS26 – see Ref 21.

Line 242 – it is not clear from here, or Methods, exactly what has been included in the IS26 transposase family. From Fig. 2C it looks like only minor variants rather than the wider IS6 family, which IS6100 also belongs to. Some of the IS26 variants may be due to errors – many recent plasmid sequences are from hybrid Nanopore/Illumina assemblies where “correction” of long reads by short reads from the wrong copy of a repeat can lead to errors and homopolymer errors are quite common.

Lines 252-3 – the fact that TEs and ARGs are associated with conjugative plasmids was known long before the publication of Ref. 23 (which has two Letters to the Editor commenting on it).

Lines 256-7 – Tn5403 does not carry any ARG and the names TnAs1, TnAs2 and TnAs3 need to be equated to the Tn3 and Tn21 subfamilies. See e.g., TnCentral for more information.

Lines 260, 263, 266 – reference to particular papers or accession nos. is needed for these dates.

Line 271 (also 277) – lateral gene transfer (LGT) is normally used to indicate transfer between cells, not transfer between DNA molecules in the same cell.

Lines 272-80 – I can't find direct evidence of movement of any “tARGi” from a plasmid to the chromosome. This would require specific examples with evidence of transfer e.g., the same CBS containing the same ARG and TE in the same order, identification of “empty” sites before insertion, flanking direct repeats if appropriate. Most MRR would not be transposable as a whole entity because of the way that they are created - subsequent insertion of other elements and ARG often render the original element inactive in terms of further movement, although the original ends may still be present (see e.g., doi: 10.1016/j.plasmid.2020.102528)

Line 277 – is anything known about such barriers? Can any papers be cited?

6) Small vs. large plasmids Lines 301-87

This analysis does not provide new information.

Line 303 – see comments above on Ref 23.

Lines 307-8 – the number of genes in the *tra* operon needed to form the conjugative apparatus and associated genes transfer DNA means that conjugative plasmids will be large.

Lines 311-4, 324 – it is already well known that large and small plasmids are not closely related. They have different replication systems, small, high copy plasmids can rely on random partitioning for vertical inheritance while large plasmids encode mechanisms for partitioning and post-segregational killing etc.

Lines 321-24 – all conjugative and mobilizable plasmids will have a relaxase, but the sequences need not be closely related. The presence of IS26 and Tn3 transposases on both plasmid types is not relevant to relatedness of the plasmid backbones. The fact that these are the only other shared protein families

listed would suggest some transfer of between the two plasmid types, apparently contradicting line 325. Lines 332 – all plasmids need a replication initiation system and it is well known that these differ for large and small plasmids – this is stated on lines 334-5. Many large plasmids have rep genes other than the repB type, which could explain the 49% of plasmids without a repB or rop.

Line 365 – what is meant by “evolution of ARG” here?

Lines 372-3, 383-7 – the text here is partly repeated. Do these large plasmids also have other replication initiation genes? i.e., maybe they should not be considered ropB type plasmids? Do these “fusions” include the whole Col plasmid? (and wording on line 373 should be “or plasmid fusions”?)

Line 374-81 – how closely related are these 24 plasmids? Are they all basically the same? This is one plasmid type known to be associated with *K. pneumoniae* carrying blaKPC genes. Many isolates of limited ST carrying blaKPC have been sequenced, which could result in biases. This is not considered.

Line 383 – are these plasmids likely mobilizable?

Line 386 – “depleted” is not the right word here – it suggests that small plasmids have lost ARG, while they may have never had them.

Lines 404-6 – again, the purpose of looking at RepB phylogeny is unclear.

Lines 414-5 – it is well known that highly related plasmids can have different MRR with different ARGs and associated mobile genetic elements.

Lines 425-31 – I am not sure if I follow these arguments - non-conjugative plasmids related to conjugative plasmid may have lost conjugation genes and/or relaxase (e.g., due to deletions mediated by mobile genetic elements), rather than being ancestral. This would need to be looked at, as well as whether any plasmids might have been mis-classified in terms of their mobility.

7) Discussion

Many of the comments above also apply to conclusions made in the Discussion.

Lines 457-9 - a reference is needed for this statement.

Lines 459-60 – wording does not make sense.

Lines 474-6 – this is already well known.

Lines 487 – small plasmids likely have constraints on their size. Conjugative plasmids are large because they need to encode conjugation machinery, which requires many genes.

Lines 490-493 – have any previous studies tried to do this?

Lines 498-99 – a plasmid with more genes will always be larger than one with less genes? How much does the work reported in Ref 12 overlap with this manuscript?

Lines 504-5 – small plasmids may lose TE after ARG have been inserted – see e.g.,

<https://doi.org/10.1016/>

[j.plasmid.2011.10.001](https://doi.org/10.1016/j.plasmid.2011.10.001) in reference to small mobilizable InnQ-type plasmids

Lines 511-2 – movement from the chromosome to a plasmid in the same cell, mediated by a mobile genetic element, explains how mobile ARGs are captured and spread, e.g., capture of blaSHV genes from *K. pneumoniae* by IS26 – see e.g., Ref. 11 for other examples. This does seem to be a rare event, as there is usually only evidence of each AGR type (i.e., closely related variants) having been captured by one particular mobile element, occasionally two.

Line 514 – most of the TE types considered here are IS and do not themselves carry ARG (except Tn3 family).

Lines 515-6 – why would the evolution of “tARGs” expected to be the same as plasmids that carry them? Mobile elements are, by definition, mobile and MRR evolve due to the actions of these and by

recombination.

8) Methods

Line 531 - how was "plasmid carrying" defined/identified?

Lines 530-45 – much of the wording here is identical/near identical to the start of the Methods section in Ref. 16, which has authors in common. While such reuse of text in Methods may not be a problem, and can in fact be useful, the previous paper should at least be cited here to show that this approach/dataset is not specific to this manuscript. Are the differences in numbers due to inclusion of an extra 78 complete genomes that were not from RefSeq in Ref. 16?

Line 534 – it is not clear what is meant by "Samples lacking host information were examined in detail". What were they examined for and how was this done?

Line 543 – "was less than 90% shorter" is unclear. More than 10% shorter?

Lines 547-52 – again wording/methods are similar to Ref 16.

Lines 565-7 – problems with wording.

Lines 568-9 – what are these eight families?

Lines 572-4 – it's not clear what is meant here? Kept for what? It is not clear in the Results that analysis was restricted to a particular set of CSBs.

Lines 580-2 – it's not clear what is meant here or why this was done. Kept for what? Is this to find composite transposons mobilised by a pair of bounding IS? The TE discussed in the Results are not known to form conventional composite transposons. Again, it is not clear in the Results that not all CBS were examined. What is meant by "TE-related CBSs" – those that carry one or more transposases?

Line 584 – why these combinations? Exactly what was in the sequences extracted?

Line 587 – why were >80% coverage and >90% identity chosen? Direct transfer is only really evident if regions present in two or more locations are virtually identical, with explanations for minor differences, backed up things such as finding an "empty" target sequence or flanking direct repeats for mobile genetic elements that generate them.

Line 605 – the pMLST information doesn't seem to be reported anywhere?

Line 606 – how accurate is MOB suite in predicting mobility type?

Lines 615 – the supplementary information does not provide enough detail to follow some parts of the manuscript.

9) References

The references cited are not always the most appropriate and the information contained in many of those cited doesn't seem to have been taken into account when designing the analysis or interpreting results. The reference list also needs to be formatted correctly (journal name abbreviations, IS numbers should be in italics, use of Title Case Like This Needs Correcting) and some are missing page numbers or equivalents (e.g., Refs. 7, 16, 19, 32).

10) Table 1

Title – CBS could be used instead of "syntenic blocks" for consistency.

This table is difficult to follow and the point of it is not clear. The difference between part A and part B is not clear – from the text it seems that part A is how many different coARGs were seen and part B the total number of occurrences of all coARGs?

The wording "their proportion of/from" is not clear.

TE and fcTE – what is the definition of a TE here? What is meant by a CBS fully matching with a TE? Does this include only the CBS as defined on lines 572-4 and/or 581-82 in Methods?

11) Figures

Fig. 1 – it's is not clear what is meant by a TE here. See also comments on text.

Fig. 2 - intrinsic chromosomal ARG would be expected to behave differently from mobile ARG that have moved to the chromosome. If only the latter were examined then maybe the story would look more like plasmids, only with smaller numbers? Part B - Line 288 – why are only these 24 ARG examined? This is not clear. In Part D, labels need to refer to well established Tn3 subfamilies – see TnCentral. In Parts C, D - these are trees of transposase proteins (as implied in the text and methods), not of the whole Tn, as implied in the figure?

Fig. 3 – Part A – I don't think that this is useful – it is well known that small and large plasmids are quite different – see comments on text.

Fig. 4, Fig. 5 - it is not clear what these figures are aimed to illustrate.

Fig. 5 What is the scale for the outer “golden” ring showing plasmid size? I can't see pink or purple arrows. Part B, see comments on Table 1. Many large conjugative AMR plasmids have different replicon types to those listed.

12) Supplementary

Table S1 lists all 9,255 replicons included in the analysis and is fairly self-explanatory. There are typos in the Comments column (“intergration”, “Co-intergate”). The sequence noted a “10 whole plasmid duplication” is probably due to PacBio assembly errors.

Table S2 - title needs rewording and teh content of the table needs more explanation.

Table S3 lists 935 genomes without plasmids used in searching for “plasmid CSBs containing ARG-TnpA combinations”. This seems to include multiple versions of some strains, ATCC strains and lab stains like DH5alpha, K12 substrains. It may not be appropriate to include all of these.

Table S4 - title needs rewording and the relevance is unclear.

Fig. S1 – see comments on Fig. 1B. What is meant by “ARGS encoding multiple mechanisms”?

Fig. S2 – see comments on Fig. 1B. Again, what is meant by “ARGS encoding multiple mechanisms”?

Fig. S3 – not clear what is meant by “CBS that match TE” - see comments on text. aph(3'')-Ib and aph(6)-Id = strAB, overlapping genes on Tn5953 that always occur together (unless one is deleted), found with blaTEM-1b in Tn6029, blaOXA-1-aac(6')-Ib-cr-catB3delta (probably catB11 here?), sometimes with arr-2, is a common cassette array, often seen on plasmids with blaCTX-M-15. See comments on text on relevance of grouping genes by mechanism of action.

Fig. S4 – see comments on text on biases of plasmid sequences in databases over time. Numbers of plasmids go up dramatically in the early 2000s – sequencing entire plasmids before this time was very difficult and relatively few isolates are sequenced retrospectively.

Fig. S5 - see comments on Fig. 2C and D. IS6100 belongs to the IS6 family – it is not a separate family.

Fig. S6 – legend – LGT is between cells, not between plasmids and chromosomes.

Fig. S7 – it's not clear why this is needed – this information could be included in Fig. 3.

Fig. S8 – the title says that this figure is for Escherichia and Salmonella only, Klebsiella included in the key and there seem to be the odd pink block in the innermost circle.

Fig. S9 – the reason for including this figure is not clear. Part A is already largely described in the text and part B doesn't seem to be referred to in the main text.

13) Minor/formatting/examples of English/wording problems etc

Throughout – it should be “antibiotic resistance genes”, not “antibiotics resistance genes”

Lines 43-4 – problems with word order.

Line 48-9 – wording does not make sense.

Line 51 – what is meant by mutations here?

Line 53 – incomplete sentence and which vancomycin resistance gene?

Lines 79-80 - wording problems.

Line 94 – “encoding for” is like saying “coding for for”. Either “encoding” or “coding for” should be used.

Lines 103-4 – wording does not make sense.

Lines 234-5 – this is not evolution of the TE, this is evolution of things carrying the TE?

Line 579 – “searched for”?

Lines 594-6 – wording problems “distances between... genes...as for ARG... repeated 100...”

Figs 4, 5, S8 - keys on diagram have “Conjugative”.

Reviewer #2 (Remarks to the Author):

See attached for my comments, uploaded them as a word document.

Reviewer #3 (Remarks to the Author):

The authors analyzed the distribution pattern of ARGs and other genes on plasmids and chromosomes, then tried to identify rules in the contents of collinear syntenic blocks containing ARGs distributed on different types of plasmids found in *Escherichia*, *Salmonella*, and *Klebsiella*. The authors found a trend that CSB-carrying ARGs were not shared between small plasmids and large plasmids.

Most readers expect to observe particular IS groups, such as IS26, and Tn3-related elements, as drivers of ARG transmission among replicons in bacterial populations; however, the observation frequency for each mobile DNA, and their ‘favorite replicons’ has not yet been presented in a quantitative manner in previous pieces of literature based on a large-scale genomic data. Therefore, I think that this type of research is valuable and benefits the scientific community.

However, I had difficulty in finding the research question. I assume that the title “Transposable elements drive the evolution of multiple drug resistance plasmids” is because 100% of CSB equaled known transposons registered in the database (Table 1). This means that known mechanisms fully explain how ARG clusters accumulate in KES replicons. This conflicts with the statement introduced at abstract line 22 in the abstract: “However, the genetic mechanisms involved in the accumulation of diverse resistance genes in plasmid genomes remain elusive”. Genetic mechanisms other than TEs/site-specific recombination, that can transfer ARGs between replicons are only illegitimate recombination and homologous recombination. The authors did not examine these two mechanisms. Moreover, the authors did not conduct either gene gain/loss event estimation on the tree (relevant to the statement in line 429)

or TE insertion location search at the base position level to address how many independent CSB acquisitions occurred in the plasmid evolutionary history. I think that the authors' research is not designed to address if TEs or other alternative factors drive the evolution of plasmids (=gene content change of plasmids in this manuscript).

I think that this research needs to address more specific research questions, such as "Which protein families are the major players in clustering and transmitting coARGs among plasmids in KES", "Whether specific TE protein families favor specific replicon types", etc.

It was not clear whether the authors ranked the strength of the association of protein families (MCL cluster) of all types of "DNA strand nicking/exchange enzymes (such as tyrosine recombinase of integron)" other than representative transposases with ARGs, before jumping to specific transposases at line 207 (Fig. 2B). Observation frequency of each MCL cluster is not presented as supplementary data (please provide with annotation). MCL clusters other than the five focused transposase may also be interesting.

More specific comments:

(1) Assigning "DNA strand nicking/exchange enzyme types" to protein families (MCL clusters) may help interpret the data trend of CSBs in the paragraph starting at line 142. Prokaryotic DNA strand nicking/exchange includes (i) DDE transposase (ii) HUH endonuclease (replication initiator, relaxase, IS transposase), (iii) DDED transposase (ruvC type), (iv) tyrosine recombinase (integron integrase, phage/ICE integrase, integrases from other types of integrative elements), and (v) serine recombinase (resolvase, invertase, and some phage integrases). Another class (vi) is Cas homologs; however, Cas may be rare in KES.

(2) Integron is frequently nested in Tn3 family transposons. I think that the authors' co-occurrence analysis cannot distinguish whether integron nested in a transposon brought ARGs into the transposon already located on a plasmid or transposition of the transposon already carrying integron with ARGs introduced ARGs into plasmids. This issue should be stated somewhere. The discovery rate of the tyrosine recombinase gene in CSB may help the discussion.

(3) The types of major TEs discovered: IS6100, DDE; IS91/ISCR, HUH; Tn3, DDE; IS26/IS26*, DDE; IS110, DEDD. Except for IS110 with an unknown transposition mechanism, all move by a replicative mode (copy-paste, also referred to as copy-in). Replicative transposons seem to be major players driving the CSB transfer among large plasmids. Again, please show the ranks of all protein families as supplementary data.

(4) Line 508. Important new finding of location bias of specific transposase genes: I suggest that authors show the location bias of other types of transposase genes as contrasting examples. Is this characteristic of specific replicative transposon groups? How about tnsB of Tn7 (cut-paste) and tnpA of IS200 (peal-paste)? Addressing these questions would reinforce the new finding of the association between specific TE and replicon types.

(5) IS110 family targets 38 bp terminal inverted repeats of Tn21 subgroup of Tn3 family (<https://pubmed.ncbi.nlm.nih.gov/14563872/>). IS110 family members may not be the driver of ARGs transfer, but just a hitchhiker. Frequent observation of IS110 family transposase genes in CSB might be associated with co-occurring Tn3 family transposons hitchhiked by the ISs. This possibility should be stated. I suggest the authors closely look at the insertion location of the IS110 family elements.

(6) Line 518. There is at least one paper wherein CRISPR spacers are deduced to be derived from IS elements. <https://pubmed.ncbi.nlm.nih.gov/23661565/>

(7) Line 519-521. These statements are incorrect. TnpB (cas homolog) of IS200 family is likely used by the ISs to introduce strand breaks at the target location or donor location, but not a part of host's immune system. IS200 family cannot function as acquired immunity as they do not have "spacer"-equivalent.

(8) Rop and repB: Please show NCBI Refseq protein ID of the representative plasmids in the main text. Headers in the fasta file submitted as Supplementary material are already converted numbers by software. They are not useful for readers to interpret the results.

Reviewer:

Thanks to the authors for an interesting paper on an important topic. I enjoyed reading it and did my best to understand the analysis used. I am afraid I have major concerns. I know myself how it can be frustrating to receive this kind of feedback when one has worked hard and in good faith on a paper – I've invested time trying to make these comments in a constructive spirit out of a concern for the quality of the science. I hope they are useful.

Major concerns:

1. The authors want to examine the co-occurrence of ARGs. Many important mobile ARGs produce similar protein products but are epidemiologically distinct, mobilised by different mobile elements, and spreading on short evolutionary timescales (years to decades). This is key to understanding their co-occurrence patterns. However, if I understand the analysis methods correctly, these distinct genes are merged into a single 'gene' because the authors cluster based on protein similarity using an MCL approach as if they were doing a standard pangenome analysis. They then discard members of the resulting protein families that are not identified by CARD as ARGs, but this still means that different genes in CARD end up within the same protein family. This is misleading: looking at co-occurrence of protein families that contain the products of ARGs as if they were one 'thing' is simply not the same as co-occurrence of ARGs themselves. The pangenome analysis approach is not right for this problem, and so all the downstream results seem flawed.

To give a concrete example, I will take the section on antagonistic ARG families (L118 and following)

The authors write that *ampC* and *blaCTX-M-15* rarely co-occur on *Escherichia* plasmids. But *ampC* is a gene that is rarely (if ever?) found on *Escherichia* plasmids (see minor comments below). The main text talks about individual genes, yet when the reader goes to Table S2 they find multiple genes in their a/b comparison. It is not made clear in the main text that the protein families group together differently named genes. *ampC* (a specific gene) is grouped with two other genes (from table: 'ampC, CMY-2, DHA-1'), which I take to be because *CMY-2* and *DHA-1* produce AmpC beta-lactamases and their protein products have been clustered, although this is not made clear in the text. The co-occurrence is computed not against *blaCTX-M-15* alone, but against 'CTX-M-15, KPC-2, CTX-M-14, CTX-M-65, CTX-M-55, KPC-3'. Yes, these are all extended-spectrum beta-lactamases. However, they are different proteins: KPC-2 and CTX-M-15 have ~64% protein similarity (and ~46% identity). This makes me concerned about the clustering of protein families the authors used if such different proteins are grouped together. Certainly nobody working in the epidemiology of ARGs should treat their genes as the same gene, and talking about 'co-occurrence' of genes from such data is wrong. *blaKPC-2* has a totally different mobilization history (involving the composite Tn4401) to *blaCTX-M-15* (involving ISEcp1). Reaching any conclusions about ARG movement or barriers to gene transfer from 'co-occurrence' of a and b when a and b each contain such disparate genes is impossible.

2. The authors' title reflects something that I would say is already believed in the literature. There is a lot of previous knowledge about mobile elements in AMR and there are a vast array of mobile elements that move ARGs (see e.g. TNpedia). In that context, I would suggest that defining a new general term (tARGis) is not a good idea. Furthermore, I tried to find where the authors define what a tARGi is, yet I couldn't find any definition. In the methods, they talked about 'TE-related CSBs' based on a colinear syntenic block that

contains two protein hits to a single TE sequence – this could be what they use as tARGis? If so, they should be aware that the presence of a hit to a TE within a colinear block does not necessarily mean that it is transposable nor that other mechanisms of transfer cannot be involved. Doing this sort of bioinformatic analysis means being open about these limitations, but the authors write as if they have confidently identified ‘transposable ARG islands’ – I don’t think they can claim this from their analysis.

3. The work on plasmid types unfortunately suggests a lack of engagement with existing work on plasmid types. Plasmids come in many different types which are quite distinct. The small/large distinction is only one broad distinction, and plasmids cluster very tightly into different clusters – see previous work clustering plasmid groups (e.g. doi 10.1038/s41467-020-17278-2 or 10.1038/s41467-020-16282-w). This is where to start to look at gene overlap, not overlapping genes between small and large. Plasmid gene turnover is very fast: most plasmids that have 95% identical relaxases exhibit less than 50% similarity in their gene content (PMID: 35639760) so gene overlap between any two random plasmids will be extremely low. The authors talk about an ‘objective approach’ to classify plasmids as small or large: I think this is a bit of a strawman (isn’t a size cutoff fine if we’re talking about size?) I agree with the authors that in general large plasmids carry ARGs and small plasmids don’t – but this is well-known. Also, conjugative plasmids clearly have a minimum size – they need the conjugative transfer apparatus, which sets a minimum size. The initial requirement for conjugation is the Mating pair formation (Mpf) machinery. For example, look at the F plasmid NC_002483.1 in proksee.ca and one can see that the *tra* genes required stretch around the plasmid from ~60kb to ~99kb (I think the minimum size of conjugative plasmid is about 20kb). The authors seem to pin a lot on their small vs. large genes *rop* and *repB* but these are known about – as they then go on to acknowledge. For example, *repB* (aka *mobA*) is part of mobility-typing, one of the oldest methods for grouping plasmids, which the authors are using by using MOBsuite. I found the inferences weren’t supported by the data: for example, the authors find an absence of gene sharing between small and large plasmids but they don’t give comparable figures for gene sharing *within* small and large plasmids, so there is no way of contextualising whether this is more or less than one would expect given their sizes etc.

Minor comments:

1. The dataset appears extremely similar to that analysed already by the authors in reference 16 (<https://royalsocietypublishing.org/doi/10.1098/rstb.2020.0467>) with only very slight discrepancies in numbers (in general, marginally fewer genomes in this later study e.g. (16) has 3,286 Escherichia plasmids, this study has 3,098; I’m not sure why). It’s fine to reanalyse datasets in a new way which is what the authors are doing here, but I think it’s good practice to be open and honest where analysis of the same (or highly similar) dataset has been previously presented by the same authors.

2. L118 ‘*ampC* is encoded universally in Escherichia chromosomes’

I highlight this example because I was at first glance not sure what the authors meant by this. *ampC* is a gene, AmpC is the protein product. It seems trivial but actually if what is meant by ‘*ampC*’ is not strictly defined, any interpretation of the results is difficult. I could not check any supplementary data of AMR gene presence in the genomes (sorry if I missed it, but I don’t think this is provided). I had a look in their previous paper using ~the same data, and I can see from Table 2 that *ampC* was seen in 1119/1169 Escherichia chromosomes and 1 plasmid. But now in this paper, assuming I read Table S2 correctly the authors find *ampC* or one of the related-but-different genes that make an AmpC beta-lactamase (*CMY-2*, *DHA-1*)

on 84 Escherichia plasmids. So there's a discrepancy already that confuses me and suggests that the authors are talking not about *ampC* the gene but 'AmpC beta-lactamases' in general (AmpC is the protein name, so maybe AmpC protein family), This is not currently clear in the text.

There are two plausible readings of the authors' intended meaning:

(1) Every Escherichia chromosome carries *ampC*

(2) Every time *ampC* is seen, it is on an Escherichia chromosome

The authors can't possibly mean (2) because they have detected '*ampC*' on plasmids here.

But some readers reading less carefully might think that is the case, because that is what CARD will tell them for *ampC*: if one looks on CARD's own prevalence data then only ~12% of *E. coli* have it, because CARD prevalence uses a perfect matching criterion (see <https://card.mcmaster.ca/ontology/41454>). The only non-zero entries in the CARD prevalence table are for Escherichia and Shigella (i.e. Escherichia), excepting one plasmid for Klebsiella oxytoca. So this is consistent with (2) with a strict definition of what '*ampC*' is. In fact, I suspect the authors detected *CMY-2* on plasmids, which they have clustered with *ampC* because the AmpC product is a similar protein – but these are distinct genes, which is why CARD separates them.

Furthermore, even (1) may not be strictly true. If one blasts or clusters proteins using default settings, then at first glance not every *E. coli* has *ampC* e.g. on panX, 482/500 of genomes in the dataset used have the *ampC* protein: https://pangenome.org/Escherichia_coli When I double-checked this by blasting against these genomes, I also find 482 have a copy with >95% identity. Of course, the remaining 18 may well have a more diverged copy of *ampC* and probably do have an AmpC beta-lactamase, but this suggests 'near-universal' would be better. (I haven't reproduced the authors' method for clustering proteins, but I would be surprised if they identify *ampC* on every Escherichia chromosome in this dataset given that previously they found only 1119/1169 had it – if they used more permissive protein clustering, that may be why they grouped together different ARGs into the same protein family) As a side note, I suspect the authors are probably aware, but they do not state that chromosomal *ampC* is not really a useful clinical resistance gene without further mutations. For example, from a review by Tamma et al. 2019 PMC6763639: 'although *ampC* is chromosomally encoded in *E. coli*, it lacks the necessary inducible mechanisms for expressing this β -lactamase at a high-enough level to be clinically significant in the absence of rare promoter and/or attenuator mutations'. It's a strange example to pick to suggest 'barriers to gene transfer'.

3. The 2x2 table for *Escherichia* for the *ampC/CTX-M-15* example discussed looks like 'CTX-M-15 etc.'

'ampC etc.'	absent	present
absent	699	286
present	81	3

This adds up to just n=1,069 plasmids - yet the authors say that their dataset includes 3,098 plasmids from Escherichia (L536). I don't understand this discrepancy and read the methods a few times without finding the explanation (sorry if I missed it).

Line-by-line comments:

Abstract L30 Why does a distinct evolutionary history implicate TEs as the 'main drivers' of AMR gene transfer?

Introduction: check synonyms and if unnecessary, consider just using the same word. e.g. auxiliary/accessory, microbial/prokaryotic.

L35 suggest delete ‘between microbial populations’ in this sentence

L37 I wouldn’t use genome for a plasmid sequence (in normal usage, a genome is chromosome+plasmids; but a phage genome is allowed – not sure why!)

L45 ‘antibiotics [sic] resistance’ should be ‘antibiotic resistance’ (and elsewhere)

L49 Define what a transposable element is

L51 ‘induce novel mutations’ – unclear to me what is meant here

L58 up to 30 ARGs – I couldn’t find this in the paper cited.

L99 ‘frequently observed’ also in chromosomes – how frequently? Using ‘frequent’ for different things in quick succession (L100) – suggest give the numbers

L117 ‘drug class of cephalosporin and penicillin’ this is slightly unclear, because I think the authors are talking about extended-spectrum beta-lactams for *blaCTX-M-15* not just ‘penicillin’.

L120 ‘Four additional ARG pairs’ of what sort?

L123 ‘antagonistic at the isolate level’ – I don’t understand.

L154 ‘significantly more transposases’. The results make perfect sense, although did the authors control for size of plasmid (using length in bp or number of genes) in these comparisons? It seems to me that a slight discrepancy in length would bias the observation of 8 vs. 6 for ABR vs. non-ABR.

L187 Fig. S3 – I’d appreciate more explanation in legend. Antibiotic inactivation vs. other categories is shown, but the x-axis order differs in each of them and some are missing (zero entries excluded?)

L190 compact resistance islands in plasmids – I liked this part of the analysis.

L251 Doesn’t increasing frequency of IS26 over time in terms of plasmids just suggest that we are sequencing more bacterial genomes, rather than any trend or inference about IS26 itself?

L255 Be aware that many non-mobilizable plasmids are probably mobilizable, as per PMID:36442505

Couldn’t the authors actually test their claims about temporal trends with some statistics? I don’t personally believe I see any trend in transposon types in Fig 2c,d

L238 there is previous bioinformatic work on co-residence of small/large plasmids (that the authors cite), but I don’t understand why this has relevance – because gene flow is possible due to co-residence, but doesn’t occur? To really address this, I think one would have to look at gene flow between plasmid taxonomic units categorised by whether they have ever co-occurred. Which would be an interesting analysis, but different to what has been done here. The evolution and turnover of plasmid mobility is an interesting question – the authors cite Coluzzi 2022 in their discussion (doi: [10.1093/molbev/msac115](https://doi.org/10.1093/molbev/msac115)) but don’t seem to have integrated its insights here.

L303 The authors present their work as an improvement on something by different authors (ref 23). I should declare that I wrote a letter in response to that study, which I thought had major issues – I highlighted only one in my letter, the problem of length. Another group of authors independently wrote a letter highlighting other problems (doi: [10.1073/pnas.2104685118](https://doi.org/10.1073/pnas.2104685118)); the same criticisms about integrating previous biological knowledge on ARG movement apply here.

I liked the analysis that showed that the frequency of tARGs was independent of plasmid mobility. This is a neat thing to do.

L426 I don’t understand L426 onwards ‘the evolution of plasmid mobility’ bit. What do the authors mean?

L460 The periodic extinction model was confusing to me

L467 'A constant process of transposase diversification' – I don't understand. The authors write diversification but then say that there is purifying selection (which I agree with). Can they explain this more clearly please?

L488 I don't understand what the authors mean by finding a small ancestor of a large plasmid.

L516 do the reasons remain elusive? The evolutionary histories of TEs vs. plasmids are different because they are different elements. They are not 'incompatible' (L528). Isn't this like asking why a gene can have different evolutionary history to other genes?

L553-555 I was surprised by the large contradiction between CARD and the authors' own protein family classification. The contradictions being only 3% in plasmids seems to fit my mental model of ~3% of gene content being plasmid-borne in *E. coli*, suggesting this is reflective of some sort of misassignment of general genes. But I don't know...though excluding them to be conservative is sensible, it suggests that the protein families are far too permissive.

L555 why 114,457 here where it was 114,464 elsewhere? What does 'excluding members not identified as ARGs by CARD' mean in detail?

Miscellaneous

Phylogenetic trees – what do the scale bars correspond to? This isn't stated. A scale bar of 1 as seen in some of the trees suggests extreme differences. How confident are the authors in their alignments and trees? Do the branches have bootstrap support? Did they test for recombination?

Fig. 4 'Conjungative [sic]' should be conjugative in legend and in Fig. 5 also.

Reply to referee comments on “Transposable elements drive the evolution of multiple drug resistance plasmids” by Yiqing Wang and Tal Dagan (note that the manuscript title has been modified to “Plasmid antibiotic resistance gene content is shaped by the interaction with mobile genetic elements”).

Reviewers' comments:

Reviewer #1 (Remarks to the Author):

This manuscript describes large-scale analysis of available plasmids in three species of Enterobacteriaceae, looking for co-occurrence of antibiotic resistance genes (ARG) with each other, whether these are associated with plasmid or the chromosome, whether ARG are clustered in “colinear syntenic blocks” (CBS), and associations with transposable elements and small or large plasmids. Any such analysis of large datasets needs to take into account the considerable existing literature/knowledge in this area to design the approach, produce meaningful results and inform conclusions. Many of the findings are already well known in the AMR plasmid field e.g., the title “Transposable elements drive the evolution of multiple drug resistance plasmids” does not state anything new. Lack of detail and problems with wording in places also make it difficult to understand the reason behind some parts of the analysis, exactly what was done, what is being proposed and why.

To very briefly summarize knowledge in this area, plasmid-borne ARG in the species examined here have been captured from the chromosomes of various bacteria and inserted into plasmids by the actions of various mobile genetic elements, including IS and Tn mentioned in this manuscript. Different mobile genetic elements capture and move ARG by different mechanisms. Capture events appear rare, and each AMR gene is generally intimately associated with one particular mobile genetic element, although others (e.g., IS26) can take over to mobilise ARG after the initial capture event (or subsequent movement can occur by homologous recombination).

ARG are known to cluster in regions which have been given various names e.g., antimicrobial resistance islands or multi-resistance regions (MRR), as mentioned on Lines 56-7, but also multi-drug resistance (MDR) regions, complex resistance loci (CRL) etc. This is likely to be at least partly because interruptions in important plasmid genes will be detrimental and an insertion in a place that does not interfere with plasmid functions can act as a ‘founder element’ into which other segments can be safely inserted. In addition, some transposable elements are known to target particular sequences e.g., class 1 integrons are derivatives of ‘res site hunter’ transposons that target transposons in the Tn21 subfamily of the Tn3 transposon family. There are other known associations e.g., certain ARG are found as gene cassettes, which are almost always found inserted in integrons; ISCR1 and associated ARG genes are almost always found in the 3'-conserved segment (3'-CS) of class 1 integrons. There are many publications and reviews on these aspects of mobile ARG, see e.g., Refs. 7 and 11 cited here and the papers that they cite.

Response: We thank the referee for the critical comments and the many suggestions and references. We acknowledge our error in approaching the topic almost solely from the perspective of antibiotic resistance gene cooccurrence and are grateful for the comments and suggestions made by the referees, which led us to perform new analyses and revise the manuscript entirely. In the revised version, we now integrate the role of mobile genetic elements in a proper manner (we believe). Since the modifications in the manuscript are quite profound, we list here the major changes done in revised version of the manuscript:

- The abstract has been mostly replaced, except for our main result, which remained as before.
- The introduction has been rewritten. Lines 47-72 have been replaced by an introduction of resistance islands and transposable elements.
- The results sections between lines 74-281 have been largely deleted from the manuscript, including the results reported in Table 1, Figure 1B and Figure 2B. Instead, we now describe the pipeline shortly in the first results section. Figure 1A was modified to reflect the new analysis components that correspond to the identification of protein families coding for site-specific recombinases and a network analysis of gene content in resistance islands.

- A new section on highly frequent recombinases in resistance islands has been added.
- A new section on the analysis of gene content in resistance islands (using a temporal network analysis) was added (including a new Figure 3).
- The section on distinction between small and large plasmids in lines 301-361, including Figure 3 has been moved to the supplementary file.
- A new section on the distribution of resistance islands in plasmid taxonomic units (PTUs) was added.
- The section on resistance islands in large plasmids (lines 398-431) has been replaced by a section on the distribution of resistance islands in large plasmid types (PTUs). Figure 5 has been modified accordingly and supplemented with additional elements on gene and resistance island sharing among plasmids.
- The discussion (lines 454-528) has been largely deleted and replaced by a discussion of the new results.
- The methods section was modified according to the new analyses performed.
- Supplementary Figure 1-4,6 & 9 were deleted from the manuscript.

Major overall points

a) It is not always clear in this manuscript what is meant by a “transposable element” - a defined element such as a particular IS or a transposon? Or a region containing multiple ARG and IS etc that is proposed to transpose? If the latter is meant, no direct evidence appears to be presented that any “tARGis” are transposable – presence of transposases in a region does not mean that it will be able to transposable as a single entity.

Response: We thank the referee for this important comment. We revised the manuscript to use the same terms used in earlier reviews on the topic (e.g., Partridge 2011 FEMS Microbiol Rev) and rewrote the introduction to include a definition that terminology. Throughout the manuscript we now mainly use the term ‘site specific recombinases’ (SSRs) and ‘resistance islands.’ Drawing inspiration from Baquero (2004 Nat Rev Microbiol) we also use the term resistance island ‘pieces’ or ‘blocks.’

b) Biases in databases

Various places in the text describe analysis relating to dates of samples (lines 219-31, 246-51, 259-66, Fig. 2, Fig. S4, Fig. S5). Line 221 acknowledges that this approach relies on sampling density, but Table S1 shows that only ~300 sequences are available before 2000, with >6,500 from 2000 onwards. This needs to be clearly acknowledged (a hint of this is in the Discussion, Line 464, which refers to only 2000-2020) and likely limits any meaningful conclusions that can be drawn from these parts of the analysis. Biases due to increased sequencing of e.g., plasmids carrying a particular ARG gene, isolates of a particular sequence type (ST) known to carry a particular plasmid are also not considered – see comments below.

Response: this is a fair comment. While we cannot solve the issue of bias in sequence databases, we opted for a greater transparency in the report of our results. In the new analysis presented in Fig. 2 we show the data accumulation over time and also refer to that property of the data in the legend of Fig. 2 and our interpretation of the results.

Comments on specific sections.

1) Abstract - see also comments above and below.

Lines 23-4 - I would disagree with this. There is a lot of available information on how different mobile genetic elements capture and move ARG and in some cases defined interactions between these elements.

Lines 24-9 – it is not clear how it was determined that these “tARGis” are capable of transposing – no clear, direct evidence of transposition is presented.

Lines 30-2 – this is already well known.

Response: In the revised manuscript we rewrote the abstract entirely in order to reflect the existing literature and better emphasize the novel results of our study.

2) ARG coincidence Lines 74-127, Fig. 1B, Table 1

This section describes identifying ARG on the basis of similarity to proteins in the CARD database (which is very inclusive) and looking for patterns of coincidence (“coARGS”) and whether these are more frequently seen on the chromosome or on plasmids.

The ARGs were classified according to mechanism of action and the coARG results are summarised in Fig. 1B according to this classification. It is not clear why ARGs are grouped in this way or what this is expected to show. ARGs with the same mechanism may have been mobilised by different transposable elements and this is one factor that may affect which ARGs co-occur.

Response: Indeed, our methodology for clustering of protein sequences into clusters of homologous proteins is insensitive for specific epidemiological variants. This feature of the analysis is now clearly stated in the first results part. We think that our objectives (and interpretation) do not require (or depend) on specific epidemiological variants. That being said, we now excluded most of the results on antibiotics resistance gene co-occurrence from the manuscript (i.e., Fig. 1b in the previous version) and focus mostly on the combinations observed in resistance islands (see the new Fig. 2). CARD is arguably the best and most updated database on antibiotic resistance genes. We used resistance gene identifier (a tool supplied by CARD), which detects functional ARGs based on both homology and SNP models (that explained in detail in the original CARD publication DOI: 10.1093/nar/gkw1004). To keep a strict set of ARGs, we excluded the members which are not identified as ARGs by CARD from our protein families. Our usage of CARD is slightly more conservative compare to others in the field (eg. DOI: 10.1073/pnas.2001240117 and DOI: 10.1128/mbio.01851-22).

Lines 91-3 – it would be better to exclude intrinsic chromosomal ARG from this analysis, as they will behave differently from mobile ARG. The wording here makes the sentence difficult to follow.

Response: We agree on that and indeed now focus only on the evolution of plasmid-encoded resistance genes.

Lines 95-6 state that the most frequent coARGS on plasmids are among pairs of genes encoding AMR inactivation mechanisms, but these are also the most common type of plasmid-borne ARG.

Response: this section has been now thoroughly rewritten and shortened. It is nonetheless important for us to comment that our use of Fisher test to test for significant co-occurrence of genes is done exactly for that reason. In other words, the test considers also the frequency of each gene on the relevant replicon.

Lines 103-4 – it is not clear what is meant here.

Response: A biased distribution towards plasmids means that most of the members in the relevant protein family are encoded in plasmids, i.e., they are plasmid specific (see also Rodríguez-Beltrán et al. 2020 DOI: 10.1073/pnas.2001240117 and Wang et al. 2022 DOI: 10.1098/rstb.2020.0467). We added this explanation in the manuscript where required.

The section on lines 112-27 is confusing and does not take into account published information. The *E. coli* chromosomal ampC gene doesn't seem to have been mobilised onto plasmids, so it is not surprising that it is not found with blaCTX-M-15 on plasmids. It is presumably not found on plasmids with other ARG genes either? Is the point that the “antagonistic” pairs confer resistance to the same antibiotics? If so, this is not clear. Lines 120-2 - the way this is worded make it difficult to understand which pairs of genes are considered “antagonistic”.

Response: This section has been excluded from the manuscript. But just to explain how it came to be: in that section we wrongly reported ampC as the annotation of part of the protein family in plasmids. The pair of genes that co-occur less frequently than expected by chance in plasmids is blaCTX-M-15 and bla-CMY-2. Our method clustered ampC (a chromosomal gene, which is the majority in this protein family) and blaCMY-2 (mainly found on plasmids) into one protein family, the two proteins have 76% identity and 99% coverage using local alignment. We reported here the test result of the co-occurrence only in plasmids, it was indeed inappropriate to use ampC here. According to our data, ampC has rarely been transferred onto plasmids. CARD identified only one ampC in a plasmid from *E. coli* (accession: NZ_CP048309.1) in our dataset.

3) Results – coARGS and TE, Lines 142-89, this section describes identification of collinear syntenic blocks (CSB) containing coARGS.

Lines 148-9 – this wording is unclear – coARGS in CBS are more frequent on plasmids than the chromosome? Or coARGS in chromosomes are not in CBS?

Response: this section was revised considerably. To make clear: although we analyzed CSBs in chromosomes we now report only CSBs in plasmids. We think that this is now well clarified in the manuscript.

Line 151-2 – this is as expected and there are many publications about this. Ref 9 refers to a single example of mobilisation. Various reviews cited here would make this point much more strongly.

Response: This is a fair comment (on previous knowledge that TEs mobilize ARGs). This section was considerably revised and this specific sentence deleted. We cite multiple reviews now in the introduction.

Lines 153, 195, 165, 184 – this is very unclear – what is meant by a CSB “corresponding” to a transposable element (TE)? That some CBS contain TE? Having one or more transposases in a region does not mean that the region will be transposable as a whole - mobile genetic elements can have different types of relationships with nearby ARG.

Response: We realize that this comment is a major one. First, as for the technical details, which are now better clarified in the manuscript (first results section, the corresponding methodology and Fig. 1A): the analysis we performed here is comparing the sequence of CSBs containing coARGS to the sequence of previously detected mobile genetic elements. Second: as for the interpretation of such a finding: we realized that our wording and presentation of the results in the previous manuscript version implied a link between specific ARGs and specific transposable elements (or SSRs), which is not always the case. Since our method cannot be used in order to delineate such specific relations, we revised the results section considerably and also the interpretation of the results. The main change being the analysis of resistance island content using a network approach, which enables us to treat the data of gene presence in resistance islands as many-to-many relations, without assuming a specific interaction between ARGs and SSRs. In our results and discussion, we focus on the content of whole resistance islands rather than such specific relations.

Lines 160-2 – it is not clear what is meant here.

Response: these lines in the previous manuscript version relate to the same issue as above, which we now revised considerably.

Lines 164-5 – it is already known that ARG in plasmids are often clustered (Lines 56-7, antibiotic resistance islands/MRR).

Response: Correct. Our focus is the quantification of such clustered ARGs in plasmid genomes and we think that our results on that are novel.

Lines 180-1 – IS26 is only one particular example. See general comments above.

Lines 186-8 – except for tet(A) this combination of genes, is found in structure called Tn6029 (see e.g., doi:10.1089/mdr.2010.0042, named before recent publications on IS26 pseudocomposite transposons). aph(3'')-Ib is also known by the simpler name strA and aph(6)-Id as strB. These genes are found next to each other, in fact overlapping slightly, in a Tn3-family transposon called Tn5393, fragments of which are common in MRR. Lines 188-9 – again this is already well known, as acknowledged in the Introduction, Lines 56-7.

4) Results - resistance islands in plasmids, Lines 190-231

Lines 191-2 – this now says that coARGS are “in” transposable elements, but again it is not clear what is meant.

Line 192 – it is well known that ARG form such clusters, as acknowledged here on Lines 56-7.

Response: These comments relate to our reports on transposable elements in resistance islands in the previous version of the manuscript. That section was now replaced by a new section on the frequency of SSR families in the resistance islands.

Lines 196-206 - this is not surprising if knowledge on mobile ARG is considered - ARG are known to occur next to one another for various reasons e.g., strAB as mentioned above, gene cassettes in cassette arrays in class 1 integrons, gene cassettes next to qacED1 (probably called ermE here) and sul1 in the 3'-CS of class 1 integrons, ARG associated with CR1, almost always seen in the 3'-CS.

Lines 205-6 – again, this is already well known, as acknowledged in the Introduction, Lines 56-7.

Lines 207-10 – there are not “transposase families” - IS26 and IS6100 are both part of the IS6 family, and IS91 and IS110 are IS families. The IS are not “transposons that comprise ARGs” – the original definition of an IS is that it carries only the gene(s) needed for its own transposition. Although some IS are known to carry passenger genes, as far as I know ARG have not been found as passengers. ARG are usually mobilised by adjacent IS either individually (e.g., IS26, ISCR) or in pairs in conventional composite transposons, although none of the elements mentioned here are known to form these. Tn3-family transposons are the only elements that do contain ARG.

Response: We thank the referees for these comments. This section has been now replaced by a section on the frequency SSR protein families in resistance islands.

Lines 208-9 – an IS or a Tn (or transposase) found near an ARG may not have had anything to do with movement of this ARG or clustering – it could be a simple insertion.

Response: Point well taken. We now refrain from such inference.

Line 211 – it is not clear why these particular 24 ARGs are now being analysed.

Line 213 – what is the relationship of the IS26 to the 15 ARGs? Is there evidence that it has mobilized / could mobilize them?

Line 216 – sul1 and ermE/qacED1 are found in the 3'-CS of class 1 integrons not “in transposase families”.

Line 218-9 – if fragments of mobile genetic elements are annotated then it is evident that at least part of the capturing element is often retained – for example, blaTEM genes were captured as part of Tn3 sub-family transposons (Tn1/Tn2/Tn3) and are always still in this context, even if part of the Tn is lost.

Lines 220-31 – see comment (b) above.

Response: this section was deleted from the manuscript and replaced with a new analysis where we looked for the frequency of specific SSR protein families and the network-based analysis of resistance island gene content.

Line 223 – which Tn3 element? What is the reference/accession for the date of the first isolate? The Tn3 family includes many different transposons and is divided into the Tn3 and Tn21 subfamilies, which have different organisation and different properties in relation to ARG carriage. Class 1 integrons target members of the Tn21 subfamily and ermE/qacED1 is in the 3'-CS of class 1 integrons.

Line 224 – ISCR elements are related to the IS91 family and sul2 is known to be associated with ISCR2.

Line 228 – if the IS110 family elements being referred to are IS4321 and IS5075 (Fig. S5) then these are known to target the terminal inverted repeats of Tn21 subfamily elements, where they likely prevent further transposition. Tn21-like elements carry class 1 integrons and dfrA12 is in a gene cassette found in integrons.

Response: this section was deleted. In the revised version we refrain from interpreting the co-occurrence of MGEs and ARGs in the same island as evidence for co-transfer.

Line 229 – catI is not the correct name for this gene – it is probably catA1, again known to be associated with Tn21 in sequences from back as far as the 1950s (see doi:

10.1128/MMBR.63.3.507-522.1999).

Response: here we adopted the annotation from CARD, which formerly used *catI*. We note that the suggested reference also uses *catI* as a synonym.

5) Evolution of transposases - Lines 232-280

The part of the manuscript looks at evolution within IS26 and Tn3 family transposases. Its not clear to me why this analysis was carried out or what it would be expected to show.

Response: This section was deleted. The analysis we performed is now used to explain why we may be seeing an increased cumulative number of recombinase nodes in the gene content network (new section, Figure 3).

Line 238 – the most recently assigned name, i.e., IS26-v1 not IS26*, should be used throughout – it is part of a scheme to consistently indicate all IS26 – see Ref 21.

Response: thank you for this comment. We modified the terminology accordingly.

Line 242 – it is not clear from here, or Methods, exactly what has been included in the IS26 transposase family. From Fig. 2C it looks like only minor variants rather than the wider IS6 family, which IS6100 also belongs to. Some of the IS26 variants may be due to errors – many recent plasmid sequences are from hybrid Nanopore/Illumina assemblies where “correction” of long reads by short reads from the wrong copy of a repeat can lead to errors and homopolymer errors are quite common.

Response: We agree on that point, yet, we don't think that it will be correct to just exclude those sequences. Following this comment, we added a note in the legend of Figure 3D where this analysis is now presented.

Lines 252-3 – the fact that TEs and ARGs are associated with conjugative plasmids was known long before the publication of Ref. 23 (which has two Letters to the Editor commenting on it).

Lines 256-7 – Tn5403 does not carry any ARG and the names TnAs1, TnAs2 and TnAs3 need to be equated to the Tn3 and Tn21 subfamilies. See e.g., TnCentral for more information.

Lines 260, 263, 266 – reference to particular papers or accession nos. is needed for these dates.

Line 271 (also 277) – lateral gene transfer (LGT) is normally used to indicate transfer between cells, not transfer between DNA molecules in the same cell.

Lines 272-80 – I can't find direct evidence of movement of any “tARGi” from a plasmid to the chromosome. This would require specific examples with evidence of transfer e.g., the same CBS containing the same ARG and TE in the same order, identification of “empty” sites before insertion, flanking direct repeats if appropriate. Most MRR would not be transposable as whole entity because of the way that they are created - subsequent insertion of other elements and ARG often render the original element inactive in terms of further movement, although the original ends may still be present (see e.g., doi: 10.1016/j.plasmid.2020.102528)

Line 277 – is anything known about such barriers? Can any papers be cited?

Response: this section was deleted. Part of the data is shown in supplementary figures. With regards to the comments on Tn5403, please note that the supp figures S1 shows the phylogeny of transposase protein family regardless of their co-occurrence with ARGs.

6) Small vs. large plasmids Lines 301-87 This analysis does not provide new information.

Line 303 – see comments above on Ref 23.

Lines 307-8 – the number of genes in the *tra* operon needed to form the conjugative apparatus and associated genes transfer DNA means that conjugative plasmids will be large.

Lines 311-4, 324 – it is already well known that large and small plasmids are not closely related. They have different replication systems, small, high copy plasmids can rely on random partitioning for vertical inheritance while large plasmids encode mechanisms for partitioning and post-segregational killing etc.

Lines 321-24 – all conjugative and mobilizable plasmids will have a relaxase, but the sequences need not be closely related. The presence of IS26 and Tn3 transposases on both plasmid types is

not relevant to relatedness of the plasmid backbones. The fact that these are the only other shared protein families listed would suggest some transfer of between the two plasmid types, apparently contradicting line 325.

Lines 332 – all plasmids need a replication initiation system and it is well known that these differ for large and small plasmids – this is stated on lines 334-5. Many large plasmids have rep genes other than the repB type, which could explain the 49% of plasmids without a repB or rop.

Response: this section was modified, also taking into account the comments by the referee, and moved to the supplementary material. We agree with the referee on most statements here on differences between small and large plasmids, and thought that it would be good to show that also by the simple measures of common genes (for the benefit of scientists that are not aware of such differences). In the revised manuscript we now use the plasmid taxonomic units (PTUs) to examine the distribution of resistance islands in plasmids. We chose to retain the distinction between small and large plasmids with our marker genes, as the *KES* PTUs were so far not described as size-specific (at least as far as we know).

Line 365 – what is meant by “evolution of ARG” here?

Response: we meant here ‘ARG content’. This sentence was deleted during the modification of that section.

Lines 372-3, 383-7 – the text here is partly repeated. Do these large plasmids also have other replication initiation genes? i.e., maybe they should not be considered ropB type plasmids? Do these “fusions” include the whole Col plasmid? (and wording on line 373 should be “or plasmid fusions”?)

Response: the section was revised and the additional information is provided in the supplementary.

Line 374-81 – how closely related are these 24 plasmids? Are they all basically the same? This is one plasmid type known to be associated with *K. pneumoniae* carrying blaKPC genes. Many isolates of limited ST carrying blaKPC have been sequenced, which could result in biases. This is not considered.

Response: this is a fair point. Indeed, these plasmids may be identical, however, we tested that the strains in our dataset are not. To make it accurate, we now refer to the number of isolates where these plasmid genomes were observed. Additionally, we added a note in the legend of Figure 4 to explain that we are counting isolates where the plasmid was documented rather than different specific plasmid types.

Line 383 – are these plasmids likely mobilizable?

Response: Yes, they are mobilizable and we mark that plasmid property in the Figure 4.

Line 386 – “depleted” is not the right word here – it suggests that small plasmids have lost ARG, while they may have never had them.

Response: Indeed. We now use the term ‘rare’ instead (end of section on small plasmids).

Lines 404-6 – again, the purpose of looking at RepB phylogeny is unclear.

Lines 414-5 – it is well known that highly related plasmids can have different MRR with different ARGs and associated mobile genetic elements.

Lines 425-31 – I am not sure if I follow these arguments - non-conjugative plasmids related to conjugative plasmid may have lost conjugation genes and/or relaxase (e.g., due to deletions mediated by mobile genetic elements), rather than being ancestral. This would need to be looked at, as well as whether any plasmids might have been mis-classified in terms of their mobility.

Response: this section was deleted and replaced with results on PTUs including large plasmids. Regarding the last comment here – we agree. We think that those plasmids are related rather than misclassified as our classification largely corresponds to previous publications that used the PTU

scheme. At any rate, we hope that the new analysis using the PTUs classification scheme supplies a better depiction of the differences between different plasmid types.

7) Discussion

Many of the comments above also apply to conclusions made in the Discussion. Lines 457-9 - a reference is needed for this statement.

Reply they come later in line 462

Lines 459-60 – wording does not make sense.

Lines 474-6 – this is already well known.

Lines 487 – small plasmids likely have constraints on their size. Conjugative plasmids are large because they need to encode conjugation machinery, which requires many genes.

Lines 490-493 – have any previous studies tried to do this?

Lines 498-99 – a plasmid with more genes will always be larger than one with less genes? How much does the work reported in Ref 12 overlap with this manuscript?

Lines 504-5 – small plasmids may lose TE after ARG have been inserted – see e.g., <https://doi.org/10.1016/j.plasmid.2011.10.001> in reference to small mobilizable IncQ-type plasmids

Lines 511-2 – movement from the chromosome to a plasmid in the same cell, mediated by a mobile genetic element, explains how mobile ARGs are captured and spread, e.g., capture of blaSHV genes from *K. pneumoniae* by IS26 – see e.g., Ref. 11 for other examples. This does seem to be a rare event, as there is usually only evidence of each AGR type (i.e., closely related variants) having been captured by one particular mobile element, occasionally two.

Line 514 – most of the TE types considered here are IS and do not themselves carry ARG (except Tn3 family).

Lines 515-6 – why would the evolution of “tARGis” expected to be the same as plasmids that carry them? Mobile elements are, by definition, mobile and MRR evolve due to the actions of these and by recombination.

Response: the discussion was newly written with a focus on the new results. The topics that raised the critical comments on that section have been deleted.

8) Methods

Line 531 - how was “plasmid carrying” defined/identified?

Response: in each of the complete genome assemblies there is at least one plasmid.

Lines 530-45 – much of the wording here is identical/near identical to the start of the Methods section in Ref. 16, which has authors in common. While such reuse of text in Methods may not be a problem, and can in fact be useful, the previous paper should at least be cited here to show that this approach/dataset is not specific to this manuscript. Are the differences in numbers due to inclusion of an extra 78 complete genomes that were not from RefSeq in Ref. 16?

Response: point well taken. We now cite our previous manuscript. The source of differences in numbers is indeed as the referee described.

Line 534 – it is not clear what is meant by “Samples lacking host information were examined in detail”. What were they examined for and how was this done?

Response: this means that we looked up the background of the strains manually in BioSample, GenBank database and literatures, we looked for any info indicating that the sample has been engineered or evolved in laboratory. We added the word ‘manually’.

Line 543 – “was less than 90% shorter” is unclear. More than 10% shorter?

Response: We now simplified that description.

Lines 547-52 – again wording/methods are similar to Ref 16.

Response: we now acknowledge our previous work here.

Lines 565-7 – problems with wording.

Response: the paragraph was modified to include tests of the gene coincidence in plasmids only and that sentence was modified.

Lines 568-9 – what are these eight families?

Response: these eight protein families encode more than one resistance mechanisms in CARD, eg., *Escherichia coli acrR* with mutation (ARO: 3003807) encodes efflux and antibiotic target alteration. This part was excluded in the revised manuscript.

Lines 572-4 – it's not clear what is meant here? Kept for what? It is not clear in the Results that analysis was restricted to a particular set of CSBs.

Response: indeed. These CSBs found in at least four genomes were retained for further analysis (the excluded CSBs were detected in less than 4 plasmids). We modified this sentence. The results section does not include information about this threshold. We think that having that in the methods section is sufficient.

Lines 580-2 – it's not clear what is meant here or why this was done. Kept for what? Is this to find composite transposons mobilised by a pair of bounding IS? The TE discussed in the Results are not known to form conventional composite transposons. Again, it is not clear in the Results that not all CBS were examined. What is meant by "TE-related CBSs" – those that carry one or more transposases?

Response: we rewrote this section to better explain our pipeline. Indeed, this pipeline cannot be used to identify composite transposons. We therefor took extra care in the interpretation of our results (in the results section).

Line 584 – why these combinations? Exactly what was in the sequences extracted?

Line 587 – why were >80% coverage and >90% identity chosen? Direct transfer is only really evident if regions present in two or more locations are virtually identical, with explanations for minor differences, backed up things such as finding an "empty" target sequence or flanking direct repeats for mobile genetic elements that generate them.

Response: this section was deleted from the manuscript.

Line 605 – the pMLST information doesn't seem to be reported anywhere?

Response: we used both PlasmidFinder and pMLST to perform plasmid typing, in the previous version these results were reported in the RepB tree (Fig. 5). Now we use this info mainly to match PTUs to other known classifications (shown, e.g., in Figure 3).

Line 606 – how accurate is MOB suite in predicting mobility type?

Response: the original publication of MOB-suite reported impressive specificity and sensitivity of >95% and this classification method is standard in the field. That being said, as far as we know there was no large-scale experimental validation of those predictions. It is quite possible that the method may misclassify plasmids as non-mobile, e.g., due to the presence of yet unknown oriT regions (recent work from Eduardo Rocha suggests that). We are quite certain that self-transmissible plasmids are well classified (with some mis-classification, e.g., due to gene loss).

Lines 615 – the supplementary information does not provide enough detail to follow some parts of the manuscript.

Response: the supplementary material was modified and shortened considerably.

9) References

The references cited are not always the most appropriate and the information contained in many of those cited doesn't seem to have been taken into account when designing the analysis or interpreting results. The reference list also needs to be formatted correctly (journal name abbreviations, IS numbers should be in italics, use of Title Case Like This Needs Correcting) and some are missing page numbers or equivalents (e.g., Refs. 7, 16, 19, 32).

Response: thank you for this comment. In the revised version we sought to better explain the limitation of our approach comparing to the type of analyses that are described in those references (which we would assume is mostly done manually and in much detail). The references have been checked and corrected.

10) Table 1

Title – CBS could be used instead of “syntenic blocks” for consistency. This table is difficult to follow and the point of it is not clear. The difference between part A and part B is not clear – from the text it seems that part A is how many different coARGs were seen and part B the total number of occurrences of all coARGs? The wording “their proportion of/from” is not clear. TE and fcTE – what is the definition of a TE here? What is meant by a CBS fully matching with a TE? Does this include only the CBS as defined on lines 572-4 and/or 581-82 in Methods?

Response: the table was deleted from the manuscript.

11) Figures

Fig. 1 – it's is not clear what is meant by a TE here. See also comments on text.

Response: the figure was modified.

Fig. 2 - intrinsic chromosomal ARG would be expected to behave differently from mobile ARG that have moved to the chromosome. If only the latter were examined then maybe the story would look more like plasmids, only with smaller numbers?

Response: thank you for this interesting observation. In the revised version of the manuscript, we chose to focus on plasmids only. This is mainly because the distinction of intrinsic chromosomal ARGs would require us to subjectively select specific ARG families and we prefer to avoid that analysis.

Part B - Line 288 – why are only these 24 ARG examined? This is not clear. In Part D, labels need to refer to well established Tn3 subfamilies – see TnCentral. In Parts C, D - these are trees of transposase proteins (as implied in the text and methods), not of the whole Tn, as implied in the figure?

Response: the section in Fig. 2B was deleted. Indeed, the trees are reconstructed using only the transposase proteins. We moved Fig. 2D to supplementary and referred the labels to Tn21 clade in the legend.

Fig. 3 – Part A – I don't think that this is useful – it is well known that small and large plasmids are quite different – see comments on text.

Response: we agree. A shorter version of that is now in the supplementary material (see also above).

Fig. 4, Fig. 5 - it is not clear what these figures are aimed to illustrate.

Response: these figures were modified and the sections describing those results modified.

Fig. 5 What is the scale for the outer “golden” ring showing plasmid size? I can't see pink or purple arrows. Part B, see comments on Table 1. Many large conjugative AMR plasmids have different replicon types to those listed.

Response: this figure was modified; results are modified accordingly.

12) Supplementary

Table S1 lists all 9,255 replicons included in the analysis and is fairly self-explanatory. There are typos in the Comments column (“intergration”, “Co-intergate”). The sequence noted a “10 whole plasmid duplication” is probably due to PacBio assembly errors.

Response: the typos were fixed. The whole plasmid duplications might be indeed erroneous but this is the data and we report it as is.

Table S2 - title needs rewording and the content of the table needs more explanation.

Response: we rewrote the table titles and add annotations for the columns at the end of each supplementary tables.

Table S3 lists 935 genomes without plasmids used in searching for “plasmid CSBs containing ARG-TnpA combinations”. This seems to include multiple versions of some strains, ATCC strains and lab stains like DH5alpha, K12 substrains. It may not be appropriate to include all of these.

Response: Many thanks for this important comment. We now deleted this part of analysis (we double checked that lab strains are not included in our dataset).

Table S4 - title needs rewording and the relevance is unclear.

Fig. S1 – see comments on Fig. 1B. What is meant by “ARGs encoding multiple mechanisms”?
Fig. S2 – see comments on Fig. 1B. Again, what is meant by “ARGs encoding multiple mechanisms”?

Fig. S3 – not clear what is meant by “CBS that match TE” - see comments on text. aph(3’)-Ib and aph(6)-Id = strAB, overlapping genes on Tn5953 that always occur together (unless one is deleted), found with blaTEM-1b in Tn6029, blaOXA-1-aac(6’)-Ib-cr-catB3delta (probably catB11 here?), sometimes with arr-2, is a common cassette array, often seen on plasmids with blaCTX-M-15. See comments on text on relevance of grouping genes by mechanism of action.

Response: we rewrote the titles for supplementary tables. These figures were deleted from the manuscript.

Fig. S4 – see comments on text on biases of plasmid sequences in databases over time. Numbers of plasmids go up dramatically in the early 2000s – sequencing entire plasmids before this time was very difficult and relatively few isolates are sequenced retrospectively.

Response: indeed. We are aware of this property of the data and try to be transparent about it, especially in figure 2.

Fig. S5 - see comments on Fig. 2C and D. IS6100 belongs to the IS6 family – it is not a separate family.

Response: indeed. In our set IS6100 sequences have been clustered into a single protein family hence the phylogenetic reconstruction corresponds to that cluster.

Fig. S6 – legend – LGT is between cells, not between plasmids and chromosomes.

Response: this figure was deleted from the manuscript.

Fig. S7 – it’s not clear why this is needed – this information could be included in Fig. 3.

Response: this is the theoretical distribution of plasmid size. The whole section is now joined in the supplementary material.

Fig. S8 – the title says that this figure is for *Escherichia* and *Salmonella* only, *Klebsiella* included in the key and there seem to be the odd pink block in the innermost circle.

Response: We deleted *Escherichia* and *Salmonella* from the title. This part of Rop Phylogeny in the figure shows mainly the Rop proteins in *Escherichia* and *Salmonella*, indeed it includes a few Rop proteins in *Klebsiella*. We split the Rop phylogeny according to the sequence similarity, but not strictly according to the host genus.

Fig. S9 – the reason for including this figure is not clear. Part A is already largely described in the text and part B doesn't seem to be referred to in the main text.

Response: figure B was deleted.

13) Minor/formatting/examples of English/wording problems etc
Throughout – it should be “antibiotic resistance genes”, not “antibiotics resistance genes”

Response: thanks. We corrected this.

Lines 43-4 – problems with word order.

Line 48-9 – wording does not make sense.

Line 51 – what is meant by mutations here?

Line 53 – incomplete sentence and which vancomycin resistance gene?

Response: this section was replaced.

Lines 79-80 - wording problems.

Response: unfortunately, we cannot see the problem here.

Line 94 – “encoding for” is like saying “coding for for”. Either “encoding” or “coding for” should be used.

Response: thanks. We corrected this.

Lines 103-4 – wording does not make sense.

Lines 234-5 – this is not evolution of the TE, this is evolution of things carrying the TE?

Line 579 – “searched for”?

Response: this section was replaced.

Lines 594-6 – wording problems “distances between... genes...as for ARG... repeated 100...”

Response: thank you. We corrected this paragraph.

Figs 4, 5, S8 - keys on diagram have “Conjunctive”.

Response: that was a typo. Thanks!

Reviewer #2 (Remarks to the Author):

Reviewer:

Thanks to the authors for an interesting paper on an important topic. I enjoyed reading it and did my best to understand the analysis used. I am afraid I have major concerns. I know myself how it can be frustrating to receive this kind of feedback when one has worked hard and in good faith on a paper – I've invested time trying to make these comments in a constructive spirit out of a concern for the quality of the science. I hope they are useful.

Response: we thank the referee for the constructive comments, all of which helped to improve the manuscript (we believe). The manuscript has been considerably revised with some sections entirely excluded. Please see the complete list of modifications in first response to R1.

Major concerns:

1. The authors want to examine the co-occurrence of ARGs. Many important mobile ARGs produce similar protein products but are epidemiologically distinct, mobilised by different mobile elements, and spreading on short evolutionary timescales (years to decades). This is key to understanding their co-occurrence patterns. However, if I understand the analysis methods correctly, these distinct genes are merged into a single 'gene' because the authors cluster based on protein similarity using an MCL approach as if they were doing a standard pangenome analysis. They then discard members of the resulting protein families that are not identified by CARD as ARGs, but this still means that different genes in CARD end up within the same protein family. This is misleading: looking at co-occurrence of protein families that contain the products of ARGs as if they were one 'thing' is simply not the same as co-occurrence of ARGs themselves. The pangenome analysis approach is not right for this problem, and so all the downstream results seem flawed. To give a concrete example, I will take the section on antagonistic ARG families (L118 and following). The authors write that *ampC* and *blaCTX-M-15* rarely co-occur on *Escherichia* plasmids. But *ampC* is a gene that is rarely (if ever?) found on *Escherichia* plasmids (see minor comments below). The main text talks about individual genes, yet when the reader goes to Table S2 they find multiple genes in their *a/b* comparison. It is not made clear in the main text that the protein families group together differently named genes. *ampC* (a specific gene) is grouped with two other genes (from table: 'ampC, CMY-2, DHA-1'), which I take to be because CMY-2 and DHA-1 produce AmpC beta-lactamases and their protein products have been clustered, although this is not made clear in the text. The co-occurrence is computed not against *blaCTX-M-15* alone, but against 'CTX-M-15, KPC-2, CTX-M-14, CTX-M-65, CTX-M-55, KPC-3'. Yes, these are all extended-spectrum beta-lactamases. However, they are different proteins: KPC-2 and CTX-M-15 have ~64% protein similarity (and ~46% identity). This makes me concerned about the clustering of protein families the authors used if such different proteins are grouped together. Certainly nobody working in the epidemiology of ARGs should treat their genes as the same gene, and talking about 'co-occurrence' of genes from such data is wrong. *blaKPC-2* has a totally different mobilization history (involving the composite Tn4401) to *blaCTX-M-15* (involving ISEcp1). Reaching any conclusions about ARG movement or barriers to gene transfer from 'co-occurrence' of *a* and *b* when *a* and *b* each contain such disparate genes is impossible.

Response: we thank the referee for the kind words. Indeed, our pipeline does not allow the distinction between epidemiological variants and we make is clear in the revised version. Analysis where this distinction is important were excluded from the manuscript (such as the one mentioned here).

2. The authors' title reflects something that I would say is already believed in the literature. There is a lot of previous knowledge about mobile elements in AMR and there are a vast array of mobile elements that move ARGs (see e.g. TNpedia). In that context, I would suggest that defining a new general term (tARGis) is not a good idea. Furthermore, I tried to find where the authors define what a tARGi is, yet I couldn't find any definition. In the methods, they talked about 'TE-related CSBs' based on a colinear syntenic block that contains two protein hits to a single TE sequence – this could be what they use as tARGis? If so, they should be aware that the presence of a hit to a TE within a colinear block does not necessarily mean that it is transposable nor that other mechanisms of transfer cannot be involved. Doing this sort of bioinformatic analysis means being open about these limitations, but the authors write as if they have confidently identified 'transposable ARG islands' – I don't think they can claim this from their analysis.

Response: this is an important comment. We realize that the new term was problematic and in the revised version we use the common terms in the field. The title was also modified.

3. The work on plasmid types unfortunately suggests a lack of engagement with existing work on plasmid types. Plasmids come in many different types which are quite distinct. The small/large distinction is only one broad distinction, and plasmids cluster very tightly into different clusters – see previous work clustering plasmid groups (e.g. doi 10.1038/s41467-020-17278-2 or

10.1038/s41467-020-16282-w). This is where to start to look at gene overlap, not overlapping genes between small and large. Plasmid gene turnover is very fast: most plasmids that have 95% identical relaxases exhibit less than 50% similarity in their gene content (PMID: 35639760) so gene overlap between any two random plasmids will be extremely low. The authors talk about an 'objective approach' to classify plasmids as small or large: I think this is a bit of a strawman (isn't a size cutoff fine if we're talking about size?) I agree with the authors that in general large plasmids carry ARGs and small plasmids don't – but this is well-known. Also, conjugative plasmids clearly have a minimum size – they need the conjugative transfer apparatus, which sets a minimum size. The initial requirement for conjugation is the Mating pair formation (Mpf) machinery. For example, look at the F plasmid NC_002483.1 in proksee.ca and one can see that the tra genes required stretch around the plasmid from ~60kb to ~99kb (I think the minimum size of conjugative plasmid is about 20kb). The authors seem to pin a lot on their small vs. large genes rop and repB but these are known about – as they then go on to acknowledge. For example, repB (aka mobA) is part of mobility-typing, one of the oldest methods for grouping plasmids, which the authors are using by using MOBSuite. I found the inferences weren't supported by the data: for example, the authors find an absence of gene sharing between small and large plasmids but they don't give comparable figures for gene sharing within small and large plasmids, so there is no way of contextualising whether this is more or less than one would expect given their sizes etc.

Response: we thank the referee for this excellent comment. Following this suggestion, we performed a new analysis using the recently suggested plasmid taxonomic units (PTUs). We chose to retain the analysis of gene sharing between the small and large plasmids (now as supplementary text) as it reveals marker genes for small/large plasmid types. Importantly, we also show that the PTUs are size-specific (as far as we can tell, this property was not checked in the original PTUs publication). The RepB protein (WP_004182030.1) we used to infer the phylogeny here is not MobA.

Minor comments:

1. The dataset appears extremely similar to that analysed already by the authors in reference 16 (<https://royalsocietypublishing.org/doi/10.1098/rstb.2020.0467>) with only very slight discrepancies in numbers (in general, marginally fewer genomes in this later study e.g. (16) has 3,286 Escherichia plasmids, this study has 3,098; I'm not sure why). It's fine to reanalyse datasets in a new way which is what the authors are doing here, but I think it's good practice to be open and honest where analysis of the same (or highly similar) dataset has been previously presented by the same authors.

Response: please note that the similarity is only in the original isolate genomes that we being used (i.e., downloaded from NCBI). The most computation intensive part is the reconstruction of protein families, which was not included in the earlier publication. That being said, some of the data properties and the methodologies are indeed shared with our earlier publication and we are now make it clear in the methods section.

2. L118 'ampC is encoded universally in Escherichia chromosomes' I highlight this example because I was at first glance not sure what the authors meant by this. ampC is a gene, AmpC is the protein product. It seems trivial but actually if what is meant by 'ampC' is not strictly defined, any interpretation of the results is difficult. I could not check any supplementary data of AMR gene presence in the genomes (sorry if I missed it, but I don't think this is provided). I had a look in their previous paper using ~the same data, and I can see from Table 2 that ampC was seen in 1119/1169 Escherichia chromosomes and 1 plasmid. But now in this paper, assuming I read Table S2 correctly the authors find ampC or one of the related-but-different genes that make an AmpC beta-lactamase (CMY-2, DHA-1) on 84 Escherichia plasmids. So there's a discrepancy already that confuses me and suggests that the authors are talking not about ampC the gene but 'AmpC beta-lactamases' in general (AmpC is the protein name, so maybe AmpC protein family), This is not currently clear in the text.

There are two plausible readings of the authors' intended meaning:

- (1) Every Escherichia chromosome carries ampC
- (2) Every time ampC is seen, it is on an Escherichia chromosome

The authors can't possibly mean (2) because they have detected 'ampC' on plasmids here. But some readers reading less carefully might think that is the case, because that is what CARD will tell them for ampC: if one looks on CARD's own prevalence data then only ~12% of E. coli have it, because CARD prevalence uses a perfect matching criterion (see <https://card.mcmaster.ca/>)

ontology/41454). The only non-zero entries in the CARD prevalence table are for Escherichia and Shigella (i.e. Escherichia), excepting one plasmid for Klebsiella oxytoca. So this is consistent with (2) with a strict definition of what 'ampC' is. In fact, I suspect the authors detected CMY-2 on plasmids, which they have clustered with ampC because the AmpC product is a similar protein – but these are distinct genes, which is why CARD separates them.

Furthermore, even (1) may not be strictly true. If one blasts or clusters proteins using default settings, then at first glance not every E. coli has ampC e.g. on panX, 482/500 of genomes in the dataset used have the ampC protein: https://pangenome.org/Escherichia_coli When I double-checked this by blasting against these genomes, I also find 482 have a copy with >95% identity. Of course, the remaining 18 may well have a more diverged copy of ampC and probably do have an AmpC beta-lactamase, but this suggests 'near-universal' would be better. (I haven't reproduced the authors' method for clustering proteins, but I would be surprised if they identify ampC on every Escherichia chromosome in this dataset given that previously they found only 1119/1169 had it – if they used more permissive protein clustering, that may be why they grouped together different ARGs into the same protein family) As a side note, I suspect the authors are probably aware, but they do not state that chromosomal ampC is not really a useful clinical resistance gene without further mutations. For example, from a review by Tamma et al. 2019 PMC6763639: 'although ampC is chromosomally encoded in E. coli, it lacks the necessary inducible mechanisms for expressing this β -lactamase at a high-enough level to be clinically significant in the absence of rare promotor and/or attenuator mutations'. It's a strange example to pick to suggest 'barriers to gene transfer'.

Response: we thank the referee for this very observant comment. Indeed, we clustered AmpC (mainly found in chromosomes) and CMY-2 (in plasmids) into a single protein family, but we did our analysis on chromosome and plasmid separately. In the previous version we misannotated the plasmid part of protein family CMY-2 as AmpC. As we explain above, indeed our protein sequence clustering approach may group different epidemiological variants together. We therefore aimed to refrain from making conclusions in that resolution in the current version. The section referred here (ampC) has been deleted from the manuscript. Importantly, as mentioned above, the data in this manuscript is somewhat different from our earlier work. In the previous manuscript we identified homologs using a 'star analysis' with sequence similarity searches using protein families of Escherichia as queries. Here we clustered the protein sequences in the three genera into clusters of homologous proteins, hence the result may be somewhat different (and we think also more robust).

3. The 2x2 table for Escherichia for the ampC/CTX-M-15 example discussed looks like 'CTX-M-15 etc.'

'ampC etc.'	absent	present
absent	699	286
present	81	3

This adds up to just n=1,069 plasmids - yet the authors say that their dataset includes 3,098 plasmids from Escherichia (L536). I don't understand this discrepancy and read the methods a few times without finding the explanation (sorry if I missed it).

Response: in that dependency test, only plasmids that encode ARGs are included (hence the discrepancy from the total plasmids in the dataset). The idea is to test whether the co-occurrence of two ARGs is significantly different from their general prevalence in plasmids. Hence plasmids without antibiotic resistance genes are not counted.

Line-by-line comments:

Abstract L30 Why does a distinct evolutionary history implicate TEs as the 'main drivers' of AMR gene transfer?

Response: we modified the abstract. The conclusion of TEs as the main drivers is biased on the finding of 80% of the ARGs in resistance islands (in plasmids).

Introduction: check synonyms and if unnecessary, consider just using the same word. e.g. auxiliary/accessory, microbial/prokaryotic.

Response: thank you for this comment. The introduction has been modified, also in view of this comment.

L35 suggest delete 'between microbial populations' in this sentence

Response: agreed.

L37 I wouldn't use genome for a plasmid sequence (in normal usage, a genome is chromosome+plasmids; but a phage genome is allowed – not sure why!)

Response: here we disagree. We consider this the right term to describe the plasmid nucleotide sequence – i.e., genome.

L45 'antibiotics [sic] resistance' should be 'antibiotic resistance' (and elsewhere)

Response: modified throughout the manuscript.

L49 Define what a transposable element is

Response: the introduction was rewritten and includes a definition of the terms used throughout.

L51 'induce novel mutations' – unclear to me what is meant here

Response: this section was rewritten. At any rate, we meant that the integration of TEs generates a mutation (i.e., an insertion).

L58 up to 30 ARGs – I couldn't find this in the paper cited.

Response: this is shown in figure 3D (we show there also one plasmid with >30 ARGs).

L99 'frequently observed' also in chromosomes – how frequently? Using 'frequent' for different things in quick succession (L100) – suggest give the numbers

Response: thank you for this good advice. This section was modified (with that specific part excluded).

L117 'drug class of cephalosporin and penicillin' this is slightly unclear, because I think the authors are talking about extended-spectrum beta-lactams for blaCTX-M-15 not just 'penicillin'.

L120 'Four additional ARG pairs' of what sort?

L123 'antagonistic at the isolate level' – I don't understand.

L154 'significantly more transposases'. The results make perfect sense, although did the authors control for size of plasmid (using length in bp or number of genes) in these comparisons? It seems to me that a slight discrepancy in length would bias the observation of 8 vs. 6 for ABR vs. non-ABR.

L187 Fig. S3 – I'd appreciate more explanation in legend. Antibiotic inactivation vs. other categories is shown, but the x-axis order differs in each of them and some are missing (zero entries excluded?)

Response: this section was excluded.

L190 compact resistance islands in plasmids – I liked this part of the analysis.

Response: thank you! We kept it in the revised version.

L251 Doesn't increasing frequency of IS26 over time in terms of plasmids just suggest that we are sequencing more bacterial genomes, rather than any trend or inference about IS26 itself?

Response: it is, of course, related to the number of sequenced genomes and what we want to show is the relative frequency to IS26-v1. At any rate, in the new analysis (around Fig. 2) we are being transparent about the effect of data accumulation.

L255 Be aware that many non-mobilizable plasmids are probably mobilizable, as per PMID:36442505

Response: good point. We now cite that reference.

Couldn't the authors actually test their claims about temporal trends with some statistics? I don't personally believe I see any trend in transposon types in Fig 2c,d

Response: we now include a new analysis on temporal trends in the new Fig. 2.

L238 there is previous bioinformatic work on co-residence of small/large plasmids (that the authors cite), but I don't understand why this has relevance – because gene flow is possible due to co-residence, but doesn't occur? To really address this, I think one would have to look at gene flow between plasmid taxonomic units categorised by whether they have ever co-occurred. Which would be an interesting analysis, but different to what has been done here. The evolution and turnover of plasmid mobility is an interesting question – the authors cite Coluzzi 2022 in their discussion (doi: 10.1093/molbev/msac115) but don't seem to have integrated its insights here.

Response: thank you for this comment. Switching to analyzing PTUs led to considerable changes in that section but we still discuss, to some extent, the possibility of gene flow between resident plasmids. As for the transitions in plasmids mobility – we think that this is indeed what we see here. We did not include here an extended section on that since it is pretty tricky to infer the ancestral state of resistance islands in these transitions (just to be transparent: we are working on that topic and we have a manuscript on that in the pipeline).

L303 The authors present their work as an improvement on something by different authors (ref 23). I should declare that I wrote a letter in response to that study, which I thought had major issues – I highlighted only one in my letter, the problem of length. Another group of authors independently wrote a letter highlighting other problems (doi: 10.1073/pnas.2104685118); the same criticisms about integrating previous biological knowledge on ARG movement apply here. I liked the analysis that showed that the frequency of tARGs was independent of plasmid mobility. This is a neat thing to do.

Response: thank you for this comment. We think that with the addition of PTUs to the analysis, our work provides a truly better depiction of plasmid types that are vehicles of resistance islands. In the revised manuscript we cited also the manuscripts that raised issues with Che et al. 2021 PNAS.

L426 I don't understand L426 onwards 'the evolution of plasmid mobility' bit. What do the authors mean?

Response: here we refer exactly to the topic raised above – namely – the loss of plasmid mobility. This section has been revised.

L460 The periodic extinction model was confusing to me

L467 'A constant process of transposase diversification' – I don't understand. The authors write diversification but then say that there is purifying selection (which I agree with). Can they explain this more clearly please?

L488 I don't understand what the authors mean by finding a small ancestor of a large plasmid.

L516 do the reasons remain elusive? The evolutionary histories of TEs vs. plasmids are different because they are different elements. They are not 'incompatible' (L528). Isn't this like asking why a gene can have different evolutionary history to other genes?

Response: this section was deleted from the manuscript.

L553-555 I was surprised by the large contradiction between CARD and the authors' own protein family classification. The contradictions being only 3% in plasmids seems to fit my mental model of ~3% of gene content being plasmid-borne in E. coli, suggesting this is reflective of some sort of

misassignment of general genes. But I don't know...though excluding them to be conservative is sensible, it suggests that the protein families are far too permissive.

Response: considering the identification of chromosomal ARGs, we agree with the assessment by the referee, since many of those are 'core' bacterial genes with specific variants that supply resistance. Considering plasmid-encoded genes, we would argue that contrary – that CARD is being too restrictive. In other words, homologs of resistance genes in plasmids (with our threshold of sequence similarity) are unlikely to perform any other function than antibiotic resistance.

L555 why 114,457 here where it was 114,464 elsewhere? What does 'excluding members not identified as ARGs by CARD' mean in detail?

Response: We kept 114,464 ARGs identified by CARD, after protein family construction, 7 ARGs are not clustered into any protein families, in the further analysis only the 114,457 ARGs in protein families are included. Several protein families in our analysis contain ARGs identified by CARD, the members in the same families but not identified as ARGs are excluded from the further analysis. See line 87-91.

Miscellaneous

Phylogenetic trees – what do the scale bars correspond to? This isn't stated. A scale bar of 1 as seen in some of the trees suggests extreme differences. How confident are the authors in their alignments and trees? Do the branches have bootstrap support? Did they test for recombination?

Response: The scale bar corresponds to amino acid substitutions per site (now added in the legend). Indeed, the scale bar of the RepB phylogeny suggests a high sequence divergence between different clades. Nonetheless, we think that the phylogeny supplies a good approximation for the evolutionary relationship between the analyzed plasmids. The distribution of the PTUs in the tree nodes supports our view. That being said, we now added a graphic illustration of the bootstrap support of the Rop and RepB trees.

Fig. 4 'Conjunctive [sic]' should be conjugative in legend and in Fig. 5 also.

Response: that was a typo. Thanks!

Reviewer #3 (Remarks to the Author):

The authors analyzed the distribution pattern of ARGs and other genes on plasmids and chromosomes, then tried to identify rules in the contents of collinear syntenic blocks containing ARGs distributed on different types of plasmids found in *Escherichia*, *Salmonella*, and *Klebsiella*. The authors found a trend that CSB-carrying ARGs were not shared between small plasmids and large plasmids.

Most readers expect to observe particular IS groups, such as IS26, and Tn3-related elements, as drivers of ARG transmission among replicons in bacterial populations; however, the observation frequency for each mobile DNA, and their 'favorite replicons' has not yet been presented in a quantitative manner in previous pieces of literature based on a large-scale genomic data. Therefore, I think that this type of research is valuable and benefits the scientific community.

Response: we thank the referee for the kind words. Indeed, we think that analyzing the contribution of MGEs to plasmid ARG content in a quantitative manner is novel. The manuscript has been considerably revised with some sections entirely excluded. Please see the complete list of modifications in first response to R1.

However, I had difficulty in finding the research question. I assume that the title "Transposable elements drive the evolution of multiple drug resistance plasmids" is because 100% of CSB equaled known transposons registered in the database (Table 1). This means that known mechanisms fully explain how ARG clusters accumulate in KES replicons. This conflicts with the statement introduced at abstract line 22 in the abstract: "However, the genetic mechanisms involved in the accumulation of diverse resistance genes in plasmid genomes remain elusive". Genetic

mechanisms other than TEs/site-specific recombination, that can transfer ARGs between replicons are only illegitimate recombination and homologous recombination. The authors did not examine these two mechanisms. Moreover, the authors did not conduct either gene gain/loss event estimation on the tree (relevant to the statement in line 429) or TE insertion location search at the base position level to address how many independent CSB acquisitions occurred in the plasmid evolutionary history. I think that the authors' research is not designed to address if TEs or other alternative factors drive the evolution of plasmids (=gene content change of plasmids in this manuscript).

Response: We thank the referee for this criticism. Indeed, our study is not designed to distinguish TEs/ site-specific recombination or illegitimate recombination and homologous recombination in the evolution of plasmid-encoded ARGs. Rather, as described above, we approach the contribution of these mechanisms to plasmid evolution in a quantitative manner in different plasmid types. In the revised manuscript we modified the title and abstract accordingly (as well as considerable sections of the manuscript).

I think that this research needs to address more specific research questions, such as “Which protein families are the major players in clustering and transmitting coARGs among plasmids in KES”, “Whether specific TE protein families favor specific replicon types”, etc. It was not clear whether the authors ranked the strength of the association of protein families (MCL cluster) of all types of “DNA strand nicking/exchange enzymes (such as tyrosine recombinase of integron)” other than representative transposases with ARGs, before jumping to specific transposases at line 207 (Fig. 2B). Observation frequency of each MCL cluster is not presented as supplementary data (please provide with annotation). MCL clusters other than the five focused transposase may also be interesting.

Response: We thank the referee for this excellent suggestion. We now added an analysis to specifically address these two questions. Specifically, we identified protein families that correspond to site-specific recombinases (SSRs; see Fig. 1A) and that enables us to rank the contribution of those SSRs according to their frequency in the data. We furthermore provide information on the identified families in supplementary table S2.

More specific comments:

(1) Assigning “DNA strand nicking/exchange enzyme types” to protein families (MCL clusters) may help interpret the data trend of CSBs in the paragraph starting at line 142. Prokaryotic DNA strand nicking/exchange includes (i) DDE transposase (ii) HUH endonuclease (replication initiator, relaxase, IS transposase), (iii) DEDD transposase (ruvC type), (iv) tyrosine recombinase (integron integrase, phage/ICE integrase, integrases from other types of integrative elements), and (v) serine recombinase (resolvase, invertase, and some phage integrases). Another class (vi) is Cas homologs; however, Cas may be rare in KES.

Response: Again, thanks again for this excellent suggestion. We have done exactly that.

(2) Integron is frequently nested in Tn3 family transposons. I think that the authors' co-occurrence analysis cannot distinguish whether integron nested in a transposon brought ARGs into the transposon already located on a plasmid or transposition of the transposon already carrying integron with ARGs introduced ARGs into plasmids. This issue should be stated somewhere. The discovery rate of the tyrosine recombinase gene in CSB may help the discussion.

Response: We thank the referee for this comment. Indeed, using our pipeline we cannot infer nested SSRs (or MGEs). We therefore revised the manuscript considerably to avoid suggesting the presence of trends that we cannot really infer. See Line 139-142.

(3) The types of major TEs discovered: IS6100, DDE; IS91/ISCR, HUH; Tn3, DDE; IS26/IS26*, DDE; IS110, DEDD. Except for IS110 with an unknown transposition mechanism, all move by a replicative mode (copy-paste, also referred to as copy-in). Replicative transposons seem to be major players driving the CSB transfer among large plasmids. Again, please show the ranks of all protein families as supplementary data.

Response: We add now an analysis on all the recombinases protein families. In supplementary table S2 we supply ranks of protein families encoding site-specific recombinases in resistance island on plasmids.

(4) Line 508. Important new finding of location bias of specific transposase genes: I suggest that authors show the location bias of other types of transposase genes as contrasting examples. Is this characteristic of specific replicative transposon groups? How about *tnsB* of Tn7 (cut-paste) and *tnpA* of IS200 (peal-paste)? Addressing these questions would reinforce the new finding of the association between specific TE and replicon types.

Response: We thank the referee for this comment. Most of the SSR families (107 out of 116) are characterized by a biased distribution towards plasmids (i.e., they are plasmid-specific). The detailed results are supplied in supplementary table S3. Specifically to your question, the majority of *tnpA* of IS200 are found on chromosome. *tnsB* of Tn7 is found in a family with 41 proteins on chromosomes and 18 proteins on plasmids. Tn7 rarely coincide with coARGs on plasmid, found only one plasmid carry *tnsB* of Tn7 in a coARG TE-CSB. *TnsE* which is required for insertion of Tn7 into plasmids, found 35 proteins on chromosomes and two proteins on plasmids. Hence Tn7 rarely contributes to the plasmid ARG evolution in our dataset. Since the two transposons are mainly found in chromosomes, we do not report this in the current version of our manuscript.

(5) IS110 family targets 38 bp terminal inverted repeats of Tn21 subgroup of Tn3 family (<https://pubmed.ncbi.nlm.nih.gov/14563872/>). IS110 family members may not be the driver of ARGs transfer, but just a hitchhiker. Frequent observation of IS110 family transposase genes in CSB might be associated with co-occurring Tn3 family transposons hitchhiked by the ISs. This possibility should be stated. I suggest the authors closely look at the insertion location of the IS110 family elements.

Response: Thank you for this suggestion. We added an analysis of hitchhiking ISs in the revised version.

(6) Line 518. There is at least one paper wherein CRISPR spacers are deduced to be derived from IS elements. <https://pubmed.ncbi.nlm.nih.gov/23661565/>

Response: Thank you! We have now revised significantly the discussion.

(7) Line 519-521. These statements are incorrect. *TnpB* (cas homolog) of IS200 family is likely used by the ISs to introduce strand breaks at the target location or donor location, but not a part of host's immune system. IS200 family cannot function as acquired immunity as they do not have "spacer"-equivalent.

Response: We thank the comment from the reviewer

(8) *Rop* and *repB*: Please show NCBI Refseq protein ID of the representative plasmids in the main text. Headers in the fasta file submitted as Supplementary material are already converted numbers by software. They are not useful for readers to interpret the results.

Response: we now add the non-redundant RefSeq ID in the main text. We provided the non-redundant RefSeq ID in a separate supplementary file "info.csv".

REVIEWER COMMENTS

Reviewer #1 (Remarks to the Author):

This revised manuscript has changed significantly since the first version, as the authors have removed some parts and changed some of the analysis. The focus is now on “resistance islands” (REI) rather than “TARGIs”, analysis of site-specific recombinases (SRR) has been added and a published scheme that defines “plasmid taxonomic units” (PTU) has been used to classify plasmids. The combination of new parts with some of the previous analyses means that overall the new manuscript lacks coherence. It is also still difficult to follow the reasons behind parts of the analysis: exactly what was done, what is being proposed and why, particularly around the definition of REI used and how these were identified (see more detailed comments below). While some of the analysis/results might potentially be useful, most sections relate the results back to REI, which is problematic.

There are still inaccuracies in relation to interpretation of the existing literature and results are still not really considered in the light of what is known around the topic. For example, the new title “Plasmid antibiotic resistance gene content is shaped by the interaction with mobile genetic elements” again does not state anything new.

General points

1) ‘Resistance islands’

The main problem that I have with this version of the manuscript is that it’s not clear to me what is being defined REI for the purposes of the analysis, how these REI were identified or how an ARG was identified as being in an REI or not. Line 282 mentions ‘coinciding ARGs and SSRs’, then the presence of REI, but if this is the definition used then this needs to be given much more explicitly and much earlier in the manuscript.

In Fig. 1, the diagram labelled ‘4’ shows a ‘resistance island’ which appears to contain several ARGs flanked by an orange box at each end, apparently an SSR gene (from the diagram labelled ‘2’ above). Lines 505-6 state “CBSs that overlapped with identified MGEs with at least two complete protein-coding genes were retained for further analysis” but it is still not clear what is meant here, what has the complete protein encoding genes, if they must both encode specific types of proteins, e.g. SRR genes and /or ARG? My query about a similar statement in the previous version (Lines 580-9) was not really addressed in the response.

Also, the term ‘island’ is more usually used in describing genetic elements that each insert at a particular site in the chromosome. Ref 11, cited in several places here in reference to ‘resistance islands’ (including with quotes e.g. Line 104) does not appear to use this term, only ‘genomic island’, ‘genetic island’ and ‘pathogenicity island’. Distinct regions containing multiple ARG and IS, Tn etc found inserted in plasmid backbones are more usually referred to as MRR, MDR regions, CRL etc and the ends of these regions are normally defined by clear boundaries with the plasmid backbone, often corresponding to the ends of IS or transposons, and signatures such as direct repeats may provide evidence of the original insertion. As with obtaining evidence that a ‘TARGI’ has moved, identifying the true boundaries of MRR in large datasets / without manual checking may be difficult.

The authors response states that the focus is now on the quantification of clustered ARGs in plasmid genomes and Lines 25 & 155 state that 80% of the ARGs in MDR plasmid are in REI. However, without a clear explanation of what constitutes an REI and some concrete examples of ARG that are considered to

be in an REI island or not, it's difficult to work out whether this result is accurate/meaningful. Also, how is it proposed that the other 20% of ARG ended up on plasmids? And is this 80% of the total number of ARG of all types found in 'REI' (I think this is the case) or does it mean that 80% of ARG types are found only inside REI and the other 20% are only found outside?

2) Analysis of site-specific recombinases (SRR)

The analysis now includes identification of site-specific recombinases (SRR). It seems that SSR is used to encompass true site-specific recombinases but also transposases - if so, this should be better explained. According to <https://doi.org/10.1111/j.1574-6976.1997.tb00349.x> "Transposition is the process by which genetic elements move between different locations of the genome, whereas site-specific recombination is a reaction in which DNA strands are broken and exchanged at precise positions of two target DNA loci to achieve determined biological function." Table S2 is entitled "Protein families encoding transposases and site-specific recombinases (SSRs)" making this distinction, while Table S3, which has the same basic data, is entitled "Biased presence of SSRs in plasmid and chromosome replicons" and does not. This seems inconsistent throughout the text.

Also, the relationship between parts of the analysis looking for regions matching known mobile elements in CBS to identify REI vs. identifying and using information on SRR (Lines 480-6 in Methods) is not really clear. From Lines 90-104 it seems to be more about MGE, but from Fig. 1 and maybe Lines 109-110, SRR seem to have been included as part of the definition of REI?

3) PTU analysis

The distribution of REI in relation to plasmid taxonomic units (PTUs) has now been added, with Fig. 5 modified to reflect this, as stated in the response, but some parts of this analysis don't seem to really consider what has been done by the original authors of the PTU scheme.

Lines 229 – it would be helpful to explain briefly how plasmids are classified into PTU i.e., based on average nucleotide identity (ANI).

Lines 338-9 – definition of PTU relies on the whole plasmid while only the product of a single plasmid gene, RepB, is analysed here, so it is not clear that support is needed or if the analysis here really adds this.

Line 363 – having shared genes is why these plasmids are grouped in PTU, which considers the whole plasmid/backbone. PTUs are related to Inc groups which depend on similarities in replication machinery.

RESPONSE To REVIEWER 2 PREVIOUS COMMENT: Importantly, we also show that the PTUs are size-specific (as far as we can tell, this property was not checked in the original PTUs publication).

NEW COMMENT: Ref 26, shows sizes of the prototype plasmid for each PTU in Fig. 4. The authors of this paper are considered among the leading experts in plasmid biology and have previously published on plasmid size distributions. Presumably they do not explicitly discuss size specificity of PTU, as the differences between small and large plasmids are very well known in the field.

PREVIOUS COMMENT: Line 605 – the pMLST information doesn't seem to be reported anywhere?

RESPONSE: we used both PlasmidFinder and pMLST to perform plasmid typing, in the previous version these results were reported in the RepB tree (Fig. 5). Now we use this info mainly to match PTUs to other known classifications (shown, e.g., in Figure 3).

NEW COMMENT: Now Lines 521-5 – the only mention of incompatibility groups that I could find outside this Methods section is on Line 302, while I couldn't find pMLST anywhere else, including Fig. 3 or Fig. 5.

Specific points

1) Introduction

Lines 49-76 have several inaccuracies:

Lines 51-2 – the mechanism for IS and Tn, and the one most discussed in Ref 7, is transposition, which is different from site-specific recombination that covers e.g., insertion of gene cassettes into integrons (see comments above).

Line 53 - as explained in previous comments, the original definition of an IS is that it carries only the gene(s) needed for its own transposition, so the ‘cargo’ is not inside the IS but is usually mobilised by adjacent IS, either individually (e.g., IS26, ISCR) or in pairs in conventional compound transposons, although none of the specific elements mentioned here are known to form these.

Line 54 – it is now known that IS26 does not create conventional compound transposons– IS26 flanked structures have recently been named pseudocompound transposons – see papers by RM Hall et al.

Lines 54-6 – *sul1* is found in the 3'-conserved segments of class 1 integrons (and the “1” of *sul1* should also be in italics). A different *sul* gene, *sul3*, is found in class 1 integrons with a different structure beyond the gene cassettes, but *sul2*, which is different again, does not have the same type of relationship with class 1 integrons.

Line 59 – this could be worded more accurately - IS26-mediated translocatable units (TU) have been shown to insert most efficiently adjacent to an existing copy of IS26 to form a pseudocompound transposon – see comments above and recent papers by RM Hall and colleagues.

Lines 62, 76 – what is included under ‘mobile genetic elements’ here? This term is problematic as it is also used to encompass plasmid themselves and site-specific recombinases have been included in the new analysis.

Line 70-3 – Ref. 24 refers to a very specific interaction between Salmonella Genomic Islands (SGI), which are found on the chromosome, not plasmids, and *IncA/IncC* type plasmids that are known to mobilise them. As far as I know this effect does not apply more generally to MRR on plasmids - if it does then additional/different references would need to be cited to support this. Similarly, the only ‘island’ that Ref. 25 appears to discuss is SGI, in addition to interactions between phages and plasmids.

2) Results

Line 91 - going by descriptions in the response, and the 30% identity mentioned on line 472, proteins that would be clustered in the same family could have different origins and mobilisation histories (see R2 comments on original manuscript). These are more than minor differences seen in ‘epidemiological variants’ (usually only one to a few amino acid differences), so this is still problematic.

Lines 100-1 – “matched the sequence” is still not clear. See previous comment. Do you mean that known mobile elements (or sequences matching part of them, or SRR genes?) are detected within these CBS? Fig. 1A doesn’t really seem to relate to this point.

Lines 112-3 – how is this known? A suitable reference should be cited.

Lines 113-5 – this is also not easy to follow.

Line 119 – to be truly ‘plasmid specific’ wouldn’t there have to be no examples outside plasmids?

Line 122-5 – IS26 and IS26-v1 encode more or less the same transposase and all IS6100 transposases are more or less identical. Similarly, there is little variation in the integrase of class 1 integrons, called *Int1* (this name could be used here) – having very closely related *int1* genes is the basis for the designation “class 1”. In contrast, Tn3, IS91 and IS110 here refers to the families which will include many different Tn/IS with significantly different transposases See also comments on previous version.

Lines 127-31 – does this mean that a REI need not contain a complete transposon?

Lines 131-6 – hitchhiking is normally used to describe when a non-mobile type of element (e.g., a typical class 1 integron that a defective tni transposition region) is mobilised by another element (in this case, a Tn21 family transposon). Because IS4321/IS5075 insert in the 38 bp TIR of Tn21 family transposons that are recognised by the transposase it is likely that they interfere with transposition, rather than ‘hitchhiking’ on these transposons - see previous comments.

Also, Lines 133-4 are incorrect - the only examples of the IS110 family that appear to be referred to in this manuscript are IS4321 and IS5075, which do target Tn21 subfamily 38 bp TIR, but this applies only a subset of the IS1111 subfamily of the IS110 family. Other members of the IS110/IS1111 family target different locations e.g., the attC sites of gene cassettes.

Lines 167-9 – “two SRR variants of IS26” is problematic. It would be simpler/clearer to describe them as the transposase (or tnp) genes of IS26 and IS26-v1. *sul1* is found in the 3'-conserved segment that is common to a large proportion of class 1 integrons. The resulting close proximity to a great variety of gene cassettes likely at least partly explains the large number of connections that *sul1* has. Why is “*sul2/sul3*” here? - these are different genes with different origins known to be found in different contexts and they are not indicated in Fig. 2A.

Lines 180-2 – meaning is unclear.

Lines 182 – Fig. 2C seems to show only 7 ARG added after 2011, not 8. *bla*VEB-1 was first described in 1999 and maybe complete plasmids carrying this gene were not sequenced until after 2011 and there has been no retrospective sequencing. The same may be true for other rarer genes or those that have not been well studied yet.

Line 206 – again, these protein families have many variants that are much more divergent than those encoded by IS26 and IS26 v1 and the analysis mentions only a couple of examples of each.

Lines 226-7 – is this really true? e.g., see Ref 5. It is well known that large and small plasmids have different characteristics, replication mechanisms and therefore distinct rep gene types etc - see previous comments.

Lines 232-5 - how is it suggested that these ARG get on these plasmids? See previous comments about loss of mobile elements from small plasmids following insertion by MGE.

Lines 262-3 – wouldn't it be better to leave these cointegrates out of this analysis. Do they have other ('large plasmid type') replicon(s)?

Line 365 – are all of these Tn1331::IS26? Several plasmids with complete Tn1331 have been sequenced. If the only difference is the insertion and the backbones are the same, then are they really different plasmids?

Line 269 – this could be worded better – the single insertion is in a single plasmid, which then spreads?

Line 274 – are these the cointegrates?

Lines 285-7 - without the accession nos. it is not easy to work out which plasmid associated with Ref. 38 is meant here - CP045195 has *bla*OXA-181 and *qnrS1* but is described as IncX3 in Ref. 38 and CP045194 doesn't appear to have ARG.

Line 291 – PTU-FS are presumably *Salmonella* virulence plasmids (pSLT type) that are known to rarely carry ARG, so this is not surprising. Also Lines 328-31, 439-444.

Line 347 – what is meant by “shared properties”? This is too vague.

Lines 351-2 – what is the evidence for the “evolution of host specificity”?

Lines 360-1 – see comments on previous version. It is well known that different plasmid backbones can have related MRR.

Lines 379-86 – In numbers are really only cassette array numbers. In36 has aadA2 according to INTEGRALL, not aadA3. The genes are normally written dfrA16-aadA2 to indicate the cassette array, then qacEdelta1 (the E is upper case, the delta symbol should be used), then sul1. The same genes are then listed on Lines 383-4, which is presumably an error?

Lines 385 – this is not new – a relatively small set of MGE is known to move ARG in these species.

3) Discussion

Lines 397-8 – again, this is already well known in the field (see Lines 56-8).

Line 400 – see comment above on the term “hitchhiking” - Ref. 31 does not appear to use this term. A better example of “functional dependencies” would be class 1 integrons/Tn402-type transposons being res site hunters that target Tn21-family Tn.

Line 401 – Ref. 45 seems to be more about IS26-mediated deletions during evolution than insertions and doesn’t appear to mention “essential genes”.

Line 410 – tetracyclines are also old antibiotics.

Line 414 – what kind of combinations?

Line 424 – what is meant by “PTU-specific” here? That only some PTU typically have REI?

Lines 426-9 – it’s not really clear what is meant by “horizontal transfer” or “vertical inheritance” here, normally used to refer to plasmids and bacterial cells, not to ARGs moving between plasmids- see previous comments on LGT.

Lines 436-8 – this already known – see previous comment - small plasmids may lose TE after ARG have been inserted – see e.g., <https://doi.org/10.1016/j.plasmid.2011.10.001> in reference to small mobilizable IncQ-type plasmids

Line 445 – Salmonella have a different lifestyle from E. coli and Klebsiella.

4) Methods

Lines 455-7 – the plasmids are found in those genomes?

Line 472 – this seems a very low threshold for protein families.

Line 467 – what is meant by “chromosomal families” here? Some families presumably include ARG found both intrinsically on the chromosome but also mobilised on plasmids e.g., blaSHV, fosA, oxqAB. What does this 3% of contradictions include?

Lines 456-7 - should ‘genomes’ be ‘chromosomes’ here?

Lines 465 & 474 – is it correct that these numbers are different? If so, why?

Lines 487-93 – it’s not clear why this is still included, given comments on previous version.

Lines 495-8 – wording doesn’t make sense and “insertions”

Lines 505 – this is still not well explained and was not addressed in response to previous comments. Why two protein coding genes? Are these in the MGE or the overlap? Many IS only encode a single transposase.

Line 507 “sequences”

Lines 509 – this section is entitled “ARG island analysis” but doesn’t mention REI.

5) Other scientific points/accuracy etc

Throughout: conjugation and mobilisation abilities are only predicted, not experimentally tested, and should be stated as such.

There are many places where “gene” is used when it should be “protein” and vice versa e.g.: Line 93 –

the genes encoding the proteins co-occur, not the proteins. Lines 120-21 – it is the genes that may co-occur on plasmids, not the proteins; Lines 143, 282, 296 - SRR genes; Line 194 – IS26v1 does not have a single amino acid change, its Tnp protein does; Line 196 – IS26 variants do not comprise a protein family; Table S4 proteins are not “found in” plasmids or chromosomes. Table S5 footnotes “number of ABR plasmids encoding antibiotic resistance genes from both protein families”.

Line 39, 66, 76, 233 etc – I agree with previous Reviewer 2 that using the term “plasmid genome” is best avoided. In most cases just “plasmid” or “plasmids” can be used.

Lines 43, 65, 92-3 etc - genes encode proteins, they are not encoded. They can be described as being carried by plasmids, found on plasmid etc. Similarly for RE, MRR etc .

Line 69 - “elements” is not needed – the name is “IS”.

Lines 102, 111, 116 - “encode for” is still used– see previous comments.

Lines 260-1 – while most plasmids in the Col4401 clade have a purple box to indicate PTU-E4, only a few in the ColRNAI clade have a yellow box to indicate PTU-E71, so “correspond to” seems inappropriate here.

6) Figures

Figure 1

Part A, 4 – it is unclear what this is supposed to show. It looks it is showing that the ‘resistance island’ is the same as the ‘transposable element’? What is the meaning of ‘transposable element’ here? This term is problematic, as explained in previous comments. Having “SSR” genes present at both ends does not necessarily mean that a region is transposable, it depends on the type of SRR and how they function. In Part B, what are the units of distance?

Figure 2

A – the size of the nodes is proportional to the number of times that ARG / SRR gene was seen in the data? B-F – these could specify that the date is the isolation date on the axes. D – the legend should state what grey parts of the bars represent? In the legend, what is meant by ‘sequencing artefacts’? Errors?

Figure 3

Legend – the meaning of “in proportion of plasmids in each PTU” needs rewording. What is the cut off for the small and large plasmid categories – 19 kb as on line 231? “bacterial host genus”? The wording of the second-last sentence does not make sense. Also, “The names of the PTU encoding..”

Figure 5 – does this add to original PTU analysis in Ref 26.

7) Supplementary

The analysis of plasmid size distribution has been moved to the “Supplementary text” with former Fig. 3 now as Fig. ST2 and Fig. ST1 showing a density plot for plasmid size, giving a bi-modal distribution, as reported in several previous papers by different authors. A recent preprint (<https://doi.org/10.1101/2023.09.10.557055>) suggests problems with using a log scale for this type of analysis although it does show a bi-modal distribution for plasmids in *Enterobacteriaceae* (which encompasses the species analysed here) on a linear scale, giving a different antinode [~30 kb, although the preprint does state that “the shift in the antinode too small to affect general results (yet, this remains to be verified)”. The senior author of the manuscript under review is acknowledged in this preprint so should be aware of it.

Table S1

Seems to be an unchanged version of Table 1 in the original version that does not quite fit in with the new focus of the manuscript.

Tables S2 and S3 – SRR and transposases

It seems like these tables could be combined. Are SSR that were excluded from the analysis (Lines 484-6) listed here? If so, they should be indicated.

Table S3 - The Methods don't seem to explain how the bias was calculated?

Table S4 doesn't have a title and might be better as Table S1, suitably introduced first in the text. NA needs explaining. "Two plasmid duplications" in comments is unclear

Table S5K, E and S – inclusion of these tables doesn't seem to take account of previous comments about problems with grouping ARG into quite broad families based on mechanism. Some names used are for specific genes (e.g., CTX-M-15, dfrA12) while other are not (e.g., aadA, which includes many distinct named variants), so it's not clear whether particular genes/proteins or families are meant. The name and content of each "family" really needed to be explained somewhere. Also, the names used are a mix of gene and protein names - this should be consistent (gene names should not start with an uppercase letter, the gene name is mcr-1.1, MCR-1.1 is the protein name, mph(A) is the correct format).

It's not clear why "Escherichia coli soxR with mutation conferring antibiotic resistance" is included in Table S5E - information in CARD suggests this is a mutation in a chromosomal gene, not an acquired ARG. What does "inf" indicate?

8) Examples of minor formatting, wording etc problems

Line 41 - "antibiotic resistant"

Line 63 - "Interactions between plasmids...have direct".

Line 66 - suggest "Small plasmids typically..." for simplicity.

Line 121 - should be "ARG protein families".

Line 136 - "transposes" needs fixing.

Line 192 - "An example", "encoded by".

Line 199 - "insight into".

Line 208 - "hints as for temporal aspects" is unclear.

Line 218 - "has a higher abundance".

Line 221 - "additional sequencing" or "additional sequence data".

Line 222 - "existing" not "exiting"?

Line 276 - "a few".

Line 316 - "hints as for the association" does not make sense.

Line 321 - "or Klebsiella"?

Line 348 - "related to" and this sentence does not make sense.

Line 378 - "with each other".

Line 388 - wording does not make sense.

Line 391 - "the responses to such".

Line 402 - "comprising" is not the right word here, as plasmids have other components that make them plasmids. "carrying" would be better.

Line 404-5- is "these" referring to ARG or REI?

Line 405 - meaning the number of available plasmid sequences have increased? This could simply be stated.

Lines 410-11 - "sulphonamides", "colistin" - no upper case.

Line 414 - "resistance island building blocks".

Line 453 - "as in our recent" does not make sense. Word(s) missing?

Line 459 - "homologs of"

Line 478 - "members not identified as ARGs by CARD were excluded" might be clearer/easier to read.

Lines 482-4 - "were obtained", "were not included".

Line 485 - problems with parentheses and "recombinases".

Line 500 - "mobile genetic element databases".

Line 513 - "the distances between"

Line 528 - "transposases"

Tables S3 and S5 "resistance" and "adjusted" in column headings/ footnotes needs fixing.

9) New comments on responses to previous comments

PREVIOUS COMMENT: Methods Line 531 - how was "plasmid carrying" defined/identified?

RESPONSE: in each of the complete genome assemblies there is at least one plasmid.

NEW COMMENT: Now Lines 451-3 - How was this identified? e.g. from the presence of circular sequences of appropriate length, from the name 'plasmid' in the description?

RESPONSE: CARD identified only one ampC in a plasmid from E. coli (accession: NZ_CP048309.1) in our dataset.

NEW COMMENT: The description of the sequence under NZ_CP048309.1 calls it a plasmid, but the annotations and a BLASTn search suggest that it is a fragment of chromosome. This could be a wider problem in the sequences analysed.

PREVIOUS COMMENT: Line 229 - catI is not the correct name for this gene - it is probably catA1, again known to be associated with Tn21 in sequences from back as far as the 1950s (see doi: 6 10.1128/MMBR.63.3.507-522.1999).

RESPONSE: here we adopted the annotation from CARD, which formerly used catI. We note that the suggested reference also uses catI as a synonym.

NEW COMMENT: CARD does not always use the most appropriate ARG names. catI is an outdated name and the paper mentioned here includes it in parentheses to provide a link back. catB is used in Table S5 for a distinct cat family and catA would be more consistent with this.

PREVIOUS COMMENT: Lines 530-45 - much of the wording here is identical/near identical to the start of the Methods section in Ref. 16, which has authors in common. While such reuse of text in Methods may not be a problem, and can in fact be useful, the previous paper should at least be cited here to show that this approach/dataset is not specific to this manuscript. Are the differences in numbers due to inclusion of an extra 78 complete genomes that were not from RefSeq in Ref. 16? Also see Reviewer 2's comments.

RESPONSE: point well taken. We now cite our previous manuscript. The source of differences in numbers is indeed as the referee described.

NEW COMMENT: While this manuscript (now Ref. 32) is cited on Lines 453 in Methods, the previous analysis and any overlap (or lack of) still needs to be referred to more explicitly.

PREVIOUS COMMENT: Line 534 - it is not clear what is meant by "Samples lacking host information were

examined in detail". What were they examined for and how was this done?

RESPONSE: this means that we looked up the background of the strains manually in BioSample, GenBank database and literatures, we looked for any info indicating that the sample has been engineered or evolved in laboratory. We added the word 'manually'.

NEW COMMENT: More information should be given in the manuscript itself. For example, did you also try and obtain sample dates from publication, if this information was missing from database entries.?

REVIEWER 2 PREVIOUS COMMENT: L49 Define what a transposable element is.

RESPONSE: the introduction was rewritten and includes a definition of the terms used throughout.

NEW COMMENT: No clear definition of transposable element appears to be given in the revised manuscript.

PREVIOUS COMMENT: Lines 615 – the supplementary information does not provide enough detail to follow some parts of the manuscript.

RESPONSE: the supplementary material was modified and shortened considerably.

NEW COMMENT: Shortening the supplementary material did not deal with this, as more information is needed, not less, to judge whether the results of the analysis are accurate/meaningful.

Figs. 4 and S8

PREVIOUS COMMENT Fig. S8: the title says that this figure is for Escherichia and Salmonella only, Klebsiella included in the key and there seem to be the odd pink block in the innermost circle.

RESPONSE: We deleted Escherichia and Salmonella from the title. This part of Rop Phylogeny in the figure shows mainly the Rop proteins in Escherichia and Salmonella, indeed it includes a few Rop proteins in Klebsiella. We split the Rop phylogeny according to the sequence similarity, but not strictly according to the host genus.

NEW COMMENT: Now Figs. 4 and S2 and Lines 255-60. The distinction between the two trees is not well explained in text. Presumably the 'mostly in Klebsiella' and 'mostly in E. coli/Salmonella' groups were clearly separated on a tree that included all 903 Rop sequences, so you have shown these as two different trees?

PREVIOUS COMMENT: The references cited are not always the most appropriate and the information contained in many of those cited doesn't seem to have been taken into account when designing the analysis or interpreting results. The reference list also needs to be formatted correctly (journal name abbreviations, IS numbers should be in italics, use of Title Case Like This Needs Correcting) and some are missing page numbers or equivalents (e.g., Refs. 7, 16, 19, 32).

RESPONSE: thank you for this comment. In the revised version we sought to better explain the limitation of our approach comparing to the type of analyses that are described in those references (which we would assume is mostly done manually and in much detail). The references have been checked and corrected.

NEW COMMENT: There are still problems with citations, where the information in the reference does not always appear to be accurately reflected in what is written in the text. Reference formatting also still has a few problems, e.g., Refs 31, 39, 44, 61 –Title Case needs fixing. Ref 34, mef(B) format is incorrect, Ref 35, blaNDM format is incorrect. Supplementary references also need checking.

PREVIOUS COMMENT Line 383 – are these plasmids likely mobilizable?

RESPONSE: Yes, they are mobilizable and we mark that plasmid property in the Figure 4.

NEW COMMENT: Now Lines 271-3 - this referred to plasmids in the “Klebsiella ColRNAI clade’ (=PTU-E71, Line 261) found in Escherichia. In Fig. 4, all PTU-E71 plasmids appear to be shown as non-mobilizable.

Reviewer #2 (Remarks to the Author):

The authors have resubmitted their article and made changes in response to the comments from myself and the other reviewers. I particularly appreciate that the authors decided to delete several sections where perhaps insurmountable concerns were raised. Not easy to do – thanks to the authors for taking the concerns raised seriously.

The main take-home is the final of the three results sections, that ARG movement is not a free-for-all of horizontal gene transfer in the world of plasmids (as those who work on it are aware) but rather something highly structured. Though the broad aspects of this are known, as Reviewer 3 said previously the quantification of these known (mobile) resistance islands and their ‘favourite plasmids’ in a large-scale way seems like a good thing to have done. I liked the new analysis in terms of PTUs (I hope not just because I suggested something along those lines) and I found the analysis of the strong linkage of resistance islands and PTUs interesting.

I still have concerns about the way the authors choose to present their work and a few other aspects.

Usual peer review disclaimer: written in haste, apologies if I misunderstood or missed things. Comments are not intended as arbitrary orders to be followed to get published, they are attempts at honest criticism/appraisal.

Title

I don’t think it reflects the paper’s conclusion, and it reads as extremely vague. We don’t need more vague papers about AMR that use lots of genomic data! This gets at what the fundamental research question is – which is still slightly unclear to me.

Possible alternatives (based on what I liked about the paper, so may not be the best options):

‘Antibiotic resistance islands are strongly associated with plasmid lineages

‘Antibiotic resistance islands in plasmids are largely restricted to specific plasmid lineages’

General comments

1. Certain resistance islands are often specific to particular plasmid lineages. I would expect linkage of ARGs between plasmid lineages to be more due to smaller MGEs carrying often just one ARG, such as the composite ISApI1 transposon that moved *mcr-1.1* or the Tn125 *bla*NDM transposon. I didn’t really get a discussion of this point from the authors – that is, what the links between PTUs are where they do exist. Even if they are rare, this is how the novel ARGs are added into existing resistance islands, right? So it seems like it might be worth addressing.

2. I also got a bit concerned after thinking about it that identical plasmids might be somehow biasing the

signal of resistance island association with specific PTUs, but this is the usual problem with using available genomic data. Maybe it was addressed, but probably worth mentioning somewhere as a limitation (I may have missed)

Specific comments

L28 Confusing sentence: suggest revising to 'Resistance islands are almost always carried by large plasmids, which are not always conjugative.'

L33 This is very general and slightly unclear. I would rewrite. 'The architecture of ARGs in plasmids is attributed to the workings of smaller MGEs that operate mostly within existing plasmid lineages.'

L43 Do they have to be 'diverse' ARGs or is just >1 sufficient?

L46 This presentation is misleading. Some of this burden in terms of pathogen-drug combinations is not due to MDR plasmids - for example, MDR TB because M. tb has no plasmids.

L50 some would call a plasmid a MGE - ok to use this definition, but I would make it clear that you mean smaller MGEs.

L55 This makes it sound like the integron is inherently a mobilizing thing - but it isn't. 'Mobile integron' means an integron associated with a transposon or a plasmid - see Partridge 2009

<https://doi.org/10.1111/j.1574-6976.2009.00175.x>

L61 I disagree. I don't think it remains 'elusive' (Reviewer 3 also pointed this contradiction out in the previous round). In any case, I don't understand what exactly the authors mean by something as general as 'the contribution of MGEs to the evolution of MDR plasmids'. It's no problem to say that the contribution of this study is to confirm that ARGs on KES plasmids very much organise into resistance islands; known MGEs almost completely explain these resistance islands; and (the more interesting part) that these operate very much within plasmid lineages. I strongly suggest deleting the claim that it's 'elusive'

L63 again, I would say 'smaller MGEs'

L69 Thanks for this mention. However, this makes it sound like the authors of these letters disagreed with the original study's claim that IS elements associated with ARGs are enriched in conjugative plasmids - but I don't! This is an established fact in the field. What I (and the authors of the other response) disagreed with was the methodology of that study and its claims to novelty.

L71 'may have a deleterious effect' - perhaps a fussy point, but I would say 'can' because it sometimes does, it sometimes doesn't. 'may' to me makes it sound like it's a general principle, but surely not, because in biology most things that are seen at all can sometimes have a cost, sometimes a benefit.

L72 'consequences for the plasmid interaction with the host' - Yes...but then again, the fact that it happens so much suggests that it is selected for overall in many cases

L74 I still don't understand what 'contribution to evolution' really means.

L76 co-occurring? (coinciding is strange for this context in English, I think)

L103 I agree with this statement. I like the analysis demonstrating this, but, as has been also pointed out by the other reviewers, this is well-established - to quote e.g. Partridge 2011: 'Available evidence indicates that in Gram-negative bacteria, particularly the Enterobacteriaceae, the resistance genes and associated mobile elements carried on plasmids are often found clustered together in large multiresistance regions (MRR).'

L104 'pieces of resistance islands' - I was expecting this to be a quote from Baquero 2004 - but the phrase doesn't occur anywhere in that paper

L134 This IS110 and Tn21 example is a nice one, and I'm glad the authors took seriously the comments

about integrating existing knowledge. Is the coincidence with all Tn3 transposons or just driven by Tn21 subfamily?

L154 Did you quantify anything about those ARGs that aren't within resistance islands? What are they?

L185 Out of interest, what is mcr-1.1 connected to in the network? It emerged on a composite ISAp1 transposon as a 'single ARG' so I'm intrigued that it seems to have two connections in the network

L202 ISs are extremely ancient - the first observations are surely due to changes in prevalence rather than truly de novo emergence, so I wouldn't say 'origin of IS26' - it's just in agreement with these first observations in genomic data

L208 'as for temporal' – some sort of typo. Revise: at temporal

L220 'diverse and changes only little over time' - I like this way of putting it. Could even change 'and' to 'but' for emphasis.

L222 'exiting [sic]' should be 'exciting'. On this point (maybe for discussion – see L405-406) isn't a possible driver of the reduced novelty over time that people now more commonly sequence any old isolate, whereas before they only sequenced something if it had a very interesting resistance phenotype. So, sequencing used to be far more biased towards novelty than it is now, when it is used (among other things) for routine genomic epidemiology

L241 I missed the explanation for why some plasmids aren't assigned a PTU? Is this a problem with COPLA assignment? Or co-integrates (surely not all of these?). Anyway, the pattern of separation is clear, but it surprised me so maybe worth saying in legend.

L265 on Tn1331, perhaps of interest that the first observation of Tn1331 in *K. pneumoniae* pJHCMW1 in 1987 (PMID: 2830842)

L275 I didn't see anywhere the average size of the resistance islands you identify – would be nice to add / make clearer – but presumably something of the order of 10kb? This is bigger than quite a lot of small plasmids. Not saying you are wrong in this point, just emphasises that the large size of resistance islands is a barrier to them being added to smaller plasmids. Physical considerations likely play a role: for example, work vesicle-mediated gene transfer shows that packaging and transfer times are similar for plasmid sizes up to 15 kb (PMID: 30670543). So for resistance islands it really makes sense that they can't just be added into a small plasmid, whereas for individual genes it seems (to me) harder to explain why a small plasmid shouldn't happen to gain ~1kb.

L331-333 I liked this particular example and conclusion.

L396 'show that the' - I would say 'Our results provide a quantitative summary confirming previous observations' (or similar). It's not a new observation; doesn't mean it can't be a good analysis.

L397 'implicating the interaction between MGEs and plasmids as the major driving force in the evolution of MDR plasmids' – to me this is clearer than what is meant by the vaguer 'contribution to evolution' earlier. Maybe I'm misunderstanding but I find it hard to think about an *interaction* as itself a major driving force - I might say 'mechanism' instead? This is discussion so it's up to you really, but maybe this could also appear in the introduction too, because it (to me) explains what you are arguing for.

L402 'Plasmids are...in themselves, a hotspot'. On this point about resistance islands as hotspots: Rocha and Bikard (doi: 10.1371/journal.pbio.3001514) have proposed a model for defence island formation in bacterial genomes - MGE turnover at hotspots for integration could result in similar resistance island accumulation (is my belief - it makes sense). I should say I made this point in a discussion of a recent paper of mine (doi: 10.1101/2023.08.07.551646, in press Microbial Genomics) so I'm not insisting you add it, just preferring it because I find it interesting.

L405 as above, see point about ARG combinations growing slower. Maybe if one normalises by

sequencing they don't get slower.

L408 I don't think the 'temporal integration' is as interesting as all that. In particular, I wouldn't give these as 'main examples'. *sul1* is well-known as a very common gene nearly ubiquitous in clinical class 1 integrons, which has previously been put into an explicit model of their temporal evolution. See the model of their evolution in Gillings 2008 (doi: 10.1128/JB.00152-08). It's nice that it comes out as the most connected, but this is very much because of this known fact. Then for *mcr-1.1*, it's a very recent gene only detected in 2016, as you point out. It is not in any way connected to the introduction of colistin. To me using these as 'main examples' harms the paper rather than helping it.

L410 'that where [sic]' should be 'that were'

L411 Not sure what your source for 1970s is. I would say that polymyxins (of which colistin is one) was discovered from 1947 onwards (discovered multiple times independently by people screening *Bacillus* sp., in UK, US, Japan) and then launched as a drug in mid-1960s in US, but actually rarely used and still rarely used outside of MDR infections. The crucial fact for the emergence of *mcr-1.1* is the use of colistin for growth promotion in animal feed, which is what is believed to have driven the emergence of *mcr-1.1* from *Moraxella* onto plasmids in KES.

L438 'plasmid genetics' - I wouldn't say 'plasmid genetics' - the basic biology of plasmids ?

L444 Interesting hypothesis about PTU-FS. They are *Salmonella* plasmids - I don't know whether this takes into account the presence of ARGs being different in *Salmonella* compared to KE? (I may have missed)

Figures

Figure 1 I wonder if one restricted the chromosomal analysis to only those ARGs that are also seen on plasmids, would one see any difference between the ARGs and control?

Figure 3 I like this way of showing it. It might be worth adding number of resistance islands per PTU and their median/range in sizes somehow. That's something that I didn't get from the paper on a readthrough that would have been nice to quantify.

Supplementary Text

I still don't entirely understand the rationale for needing markers for plasmid size, but OK. I understand it's nice to make a phylogeny of a single gene present in all large plasmids and show it matches PTUs. On PTUs being size-specific: Redondo-Salvo et al.'s original definition of PTUs is based on pairwise ANI with a 50% length threshold, so it seems to me this is in part built into their definition.

Author: (I have a policy of signing all peer reviews)

Conflict of interest: I discussed this article with Tal Dagan at a workshop we both attended in September 2023.

Reviewer #3 (Remarks to the Author):

I think that the authors adequately revised manuscript to respond to reviewers' comments/suggestions. Description and presentation of the first part focused on SSR-ARG association (Fig.1, Fig2, supplementary table) have been improved and provide sufficiently clear message. New results of REI - Plasmid-type association (Fig 3, Fig.5BC) provides useful information for broad audience. It might give inspiration of

new research questions.

I have only a few comments regarding minor points:

- The “SSR” term is normally used to refer to only serine-recombinase and tyrosine recombinase. The “SSR” term in this manuscript is used to cover both transposes (DEDD, DDE, HUH) and true site-specific recombinase for simplicity. This should be noted somewhere.
- Line 286. Beta in symbol.
- Fig 2. Legends. (A) .. the edges to
- Fig 2. Legends. (D) What are the numbers on the stacked bar? Protein family IDs?
- Line 402. Earlier work by Sota et al 2007 (doi: 10.1128/JB.01906-06) addressed this point by experiments. This reference should be cited.

Reply to referee comments on Nature Communications submission NCOMMS-23-16325A-Z “Plasmid antibiotic resistance gene content is shaped by the interaction with mobile genetic elements” by Yiqing Wang and Tal Dagan (former title).

We thank the referees for their comments and suggestions that were extremely helpful for us in the context of the current literature on mobile genetic elements, our use of terminology, and the general accuracy of our interpretation of the results. We believe that thanks to their questions and comments our manuscript is now more accessible for microbial geneticists. The main changes in the current revised version are:

- We added a supplementary note (SN1) to explain the relation between the colinear syntenic blocks (CSBs) and resistance island pieces (or agglomerates). That note includes also demonstrative examples of such CSBs/MGEs.
- We added a supplementary figure S1 showing the CSB length distribution.
- We added a supplementary figure S2 to show the CSBs including *mcr-1.1*
- We performed additional detailed analysis of the CSBs that yielded a new manuscript Figure 6 showing an example for CSB sharing among PTUs.
- We added new analysis of CSB diversity (Supplementary Figure S6).
- We added supplementary data including all CSBs and a video to explain how to read that data with CSBfinder (currently in our website, link in the manuscript).

The new analysis adds support to our results on the limited distribution of resistance islands in PTUs and consequently we modified the manuscript title and abstract. Our detailed responses to all comments and suggestions made by the referees are found below.

Note that line numbers in our responses relate to the tracked changes version.

In the ‘clean’ manuscript version the figure legends and figures were moved to the end of the manuscript.

REVIEWER COMMENTS

Reviewer #1 (Remarks to the Author):

This revised manuscript has changed significantly since the first version, as the authors have removed some parts and changed some of the analysis. The focus is now on “resistance islands” (REI) rather than “TARGIs”, analysis of site-specific recombinases (SRR) has been added and a published scheme that defines “plasmid taxonomic units” (PTU) has been used to classify plasmids. The combination of new parts with some of the previous analyses means that overall the new manuscript lacks coherence.

It is also still difficult to follow the reasons behind parts of the analysis: exactly what was done, what is being proposed and why, particularly around the definition of REI used and how these were identified (see more detailed comments below). While some of the analysis/results might potentially be useful, most sections relate the results back to REI, which is problematic.

There are still inaccuracies in relation to interpretation of the existing literature and results are still not really considered in the light of what is known around the topic. For example, the new title “Plasmid antibiotic resistance gene content is shaped by the interaction with mobile genetic elements” again does not state anything new.

Response: we thank the referee for investing the time and thoroughly commenting on our revised submission. The comments were extremely helpful for us in rendering our manuscript overall

more accurate and accessible to other scientists. We are also grateful for the many references to the literature that were extremely helpful for our interpretation of the results and their discussion.

Reading the comments made by the referee we feel that the essence of our analysis pipeline remained unclear. To accommodate comments that relate to the quality and integrity of our inferences, we added a new supplementary note (SN1) with concrete examples that demonstrate steps in the analysis pipeline. Additionally, we made the data from our analysis pipeline accessible (in our website), including the CSBs annotation. Since this approach is quite new and not all may be familiar with the data structure, we also uploaded a short video that explains how to view the CSB data using CSB Finder. The link to that is found in the 'Data availability' section. We hope that this resource will be of benefit for others in the field. Following this comment and also the suggestion of the 2nd referee we modified the title. Following this comment and others we streamlined our manuscript and emphasize what we think are novel aspects of plasmid ARG content evolution.

General points

1) 'Resistance islands'

The main problem that I have with this version of the manuscript is that it's not clear to me what is being defined REI for the purposes of the analysis, how these REI were identified or how an ARG was identified as being in an REI or not. Line 282 mentions 'coinciding ARGs and SSRs', then the presence of REI, but if this is the definition used then this needs to be given much more explicitly and much earlier in the manuscript.

In Fig. 1, the diagram labelled '4' shows a 'resistance island' which appears to contain several ARGs flanked by an orange box at each end, apparently an SSR gene (from the diagram labelled '2' above). Lines 505-6 state "CBSs that overlapped with identified MGEs with at least two complete protein-coding genes were retained for further analysis" but it is still not clear what is meant here, what has the complete protein encoding genes, if they must both encode specific types of proteins, e.g. SRR genes and /or ARG? My query about a similar statement in the previous version (Lines 580-9) was not really addressed in the response.

Also, the term 'island' is more usually used in describing genetic elements that each insert at a particular site in the chromosome. Ref 11, cited in several places here in reference to 'resistance islands' (including with quotes e.g. Line 104) does not appear to use this term, only 'genomic island', 'genetic island' and 'pathogenicity island'. Distinct regions containing multiple ARG and IS, Tn etc found inserted in plasmid backbones are more usually referred to as MRR, MDR regions, CRL etc and the ends of these regions are normally defined by clear boundaries with the plasmid backbone, often corresponding to the ends of IS or transposons, and signatures such as direct repeats may provide evidence of the original insertion. As with obtaining evidence that a 'TARGI' has moved, identifying the true boundaries of MRR in large datasets / without manual checking may be difficult.

Response: We agree with the referee that the annotation of resistance islands (or MRRs) is very tricky (at least in a large-scale analysis). This is the reason why our pipeline is designed to look for 'pieces of resistance islands' instead. In our inference process, we rely mostly on the conservation of gene sequence and order (i.e., synteny). The colinear syntenic blocks (CSBs) we identify do not always correspond to intact mobile genetic elements since those may have a nested structure or may be fragmented in the genome due to non-functionalization followed by degradation (or amelioration of sequence similarity to functional elements). Following this comment, we added a definition of 'resistance island pieces' at the end of the first paragraph in the results section. The description of our approach as such is largely inspired by Baquero's review that we cite in that paragraph. To better explain the steps in our analysis pipeline and their outcome, we included a new section of supplementary note (SN1) that shows demonstrative examples of 'pieces' (as well as agglomerates) that we find.

Considering the term we use, after careful consideration, we decided to retain the term 'island'. We first cite Miriagou et al. 2006 in the introduction for resistance island. This review article published in 2006 named resistance gene clusters in *Salmonella* chromosomes and plasmids as antimicrobial resistance islands. We then cite Baquero 2004 as inspiration for terming the outcome of our pipeline 'pieces' of resistance islands. Indeed, we could see that different scientists use slightly different terms (e.g., MRRs). In keeping with our view on the evolution of bacterial genome structure we personally prefer the term 'resistance islands', mainly because of the parallels to other genetic islands (not necessarily mobile ones), e.g., defence islands.

The authors response states that the focus is now on the quantification of clustered ARGs in plasmid genomes and Lines 25 & 155 state that 80% of the ARGs in MDR plasmid are in REI. However, without a clear explanation of what constitutes an REI and some concrete examples of ARG that are considered to be in an REI island or not, it's difficult to work out whether this result is accurate/meaningful. Also, how is it proposed that the other 20% of ARG ended up on plasmids? And is this 80% of the total number of ARG of all types found in 'REI' (I think this is the case) or does it mean that 80% of ARG types are found only inside REI and the other 20% are only found outside?

Response: Indeed, we find that 84% (note that we slightly modified this number) of the total number of plasmid-encoded ARGs are found in the pieces of resistance islands as we defined (note – these are all ARGs in plasmids – not specific types). We now supply concrete examples in the supplementary note (SN1). We also supply the number of instances of the resistance islands in supplementary table S6. As for the remaining 16% ARGs, we can think of several possible explanations. Due to the limitation of our data collection and statistical testing method, not all the potential ARGs in islands can have a significant cooccurrence with other ARGs in all three genera. Similarly, limitations of the data and method also apply to searching for CSBs. In that step about half of the coARG instances were excluded, likely because they appear in <4 strains. Thus, from looking at genomes during this study, we are in the opinion that the remaining ARGs are likely also found in resistance islands. Nonetheless, our choice of conservative threshold in the different pipeline steps leads to their exclusion. We added a comment on that in the discussion (lines 992-996).

2) Analysis of site-specific recombinases (SRR)

The analysis now includes identification of site-specific recombinases (SRR). It seems that SSR is used to encompass true site-specific recombinases but also transposases - if so, this should be better explained. According to <https://doi.org/10.1111/j.1574-6976.1997.tb00349.x> "Transposition is the process by which genetic elements move between different locations of the genome, whereas site-specific recombination is a reaction in which DNA strands are broken and exchanged at precise positions of two target DNA loci to achieve determined biological function." Table S2 is entitled "Protein families encoding transposases and site-specific recombinases (SSRs)" making this distinction, while Table S3, which has the same basic data, is entitled "Biased presence of SSRs in plasmid and chromosome replicons" and does not. This is seems inconsistent throughout the text.

Also, the relationship between parts of the analysis looking for regions matching known mobile elements in CBS to identify REI vs. identifying and using information on SRR (Lines 480-6 in Methods) is not really clear. From Lines 90-104 it seems to be more about MGE, but from Fig. 1 and maybe Lines 109-10, SRR seem to have been included as part of the definition of REI?

Response: We thank the referee for this comment. We are indeed grouping here (at least) two types of genes into SRRs. We now make it clear in the second paragraph of the results section. Additionally, previous Table S2 and S3 have been merged in the revised version. Please note

that the location of SRRs was not included in the definition of pieces of REI, see also the response above (and the demonstrative example in SN1).

3) PTU analysis

The distribution of REI in relation to plasmid taxonomic units (PTUs) has now been added, with Fig. 5 modified to reflect this, as stated in the response, but some parts of this analysis don't seem to really consider what has been done by the original authors of the PTU scheme. Lines 229 – it would be helpful to explain briefly how plasmids are classified into PTU i.e., based on average nucleotide identity (ANI).

Response: Here we respectfully disagree. First, the PTUs were defined based on additional plasmid properties (not only ANI). Second, we cite the relevant manuscript from Redondo-Salvo et al. 2020 where all details can be found. To help the reader, we now include an explanation of PTUs in the introduction. Please also note that our dataset is different from the first PTU classification manuscript – that is, we have a considerable number of plasmids that were not classified before.

Lines 338-9 – definition of PTU relies on the whole plasmid while only the product of a single plasmid gene, RepB, is analysed here, so it is not clear that support is needed or if the analysis here really adds this.

Response: Different PTUs share *repB* as a common gene, hence the presence of *repB* supplies a higher taxonomic resolution of those PTUs. The RepB phylogeny serves as an approximation for the evolutionary relation between the PTUs in our set.

Line 363 – having shared genes is why these plasmids are grouped in PTU, which considers the whole plasmid/backbone. PTUs are related to Inc groups which depend on similarities in replication machinery.

Response: Right, but the PTU classification is not looking directly at shared genes. We do (and we deem it a slightly better approach. See also our recent Hanke et al. 2023 biorxiv doi: 10.1101/2023.11.08.566193; Supplementary Fig. ST2.2).

RESPONSE To REVIEWER 2 PREVIOUS COMMENT: Importantly, we also show that the PTUs are size-specific (as far as we can tell, this property was not checked in the original PTUs publication).

NEW COMMENT: Ref 26, shows sizes of the prototype plasmid for each PTU in Fig. 4. The authors of this paper are considered among the leading experts in plasmid biology and have previously published on plasmid size distributions. Presumably they do not explicitly discuss size specificity of PTU, as the differences between small and large plasmids are very well known in the field.

Response: We agree with this statement of the referee and we clearly cite the relevant publications of those leaders in the field for their previous contributions and findings in multiple reviews and research articles (including, e.g., Smillie et al. 2010, Redondo-Salvo et al. 2020 and others). Additionally, we show in our supplementary note (SN2) that small and large plasmids rarely share genes – that is – they are evolutionarily unrelated (or distantly related). This is the basic observation that leads to the definition of marker genes for small and large PTUs. We also note that the data in our manuscript is much larger in comparison to Smillie et al. 2010 (check our Fig. 2E – the data today is ca. 6-fold larger compared to 2010). The data in the 2020 publications is quite large but has no specific focus like us on KES. In other words, we deem our observations

in that part as validation of previous knowledge (and this is also how we present them). The transparent definition of marker genes is required as a justification for us to use the same gene for the plasmid backbone phylogenies.

PREVIOUS COMMENT: Line 605 – the pMLST information doesn't seem to be reported anywhere?

RESPONSE: we used both PlasmidFinder and pMLST to perform plasmid typing, in the previous version these results were reported in the RepB tree (Fig. 5). Now we use this info mainly to match PTUs to other known classifications (shown, e.g., in Figure 3).

NEW COMMENT: Now Lines 521-5 – the only mention of incompatibility groups that I could find outside this Methods section is on Line 302, while I couldn't find pMLST anywhere else, including Fig. 3 or Fig. 5.

Response: We added details on the usage of pMLST in the methods section.

Specific points

1) Introduction

Lines 49-76 have several inaccuracies:

Lines 51-2 – the mechanism for IS and Tn, and the one most discussed in Ref 7, is transposition, which is different from site-specific recombination that covers e.g., insertion of gene cassettes into integrons (see comments above).

Response: Thank you. We added transposition in that sentence.

Line 53 - as explained in previous comments, the original definition of an IS is that it carries only the gene(s) needed for its own transposition, so the 'cargo' is not inside the IS but is usually mobilised by adjacent IS, either individually (e.g., IS26, ISCR) or in pairs in conventional compound transposons, although none of the specific elements mentioned here are known to form these.

Response: Thank you. We added now the cargo (or neighbouring) genes in that sentence.

Line 54 – it is now known that IS26 does not create conventional compound transposons– IS26 flanked structures have recently been named pseudocompound transposons – see papers by RM Hall et al.

Response: Many thanks for the reference. We now use the term and cite that manuscript accordingly.

Lines 54-6 – *sul1* is found in the 3'-conserved segments of class 1 integrons (and the "1" of *sul1* should also be in italics). A different *sul* gene, *sul3*, is found in class 1 integrons with a different structure beyond the gene cassettes, but *sul2*, which is different again, does not have the same type of relationship with class 1 integrons.

Response: Thank you for pointing it out, we now mention only *sul1*.

Line 59 – this could be worded more accurately - IS26-mediated translocatable units (TU) have been shown to insert most efficiently adjacent to an existing copy of IS26 to form a pseudocompound transposon – see comments above and recent papers by RM Hall and colleagues.

Response: thanks, we modified the text accordingly.

Lines 62, 76 – what is included under ‘mobile genetic elements’ here? This term is problematic as it is also used to encompass plasmid themselves and site-specific recombinases have been included in the new analysis.

Response: Right. We now qualify that term in the beginning of that paragraph.

Line 70-3 – Ref. 24 refers to a very specific interaction between Salmonella Genomic Islands (SGI), which are found on the chromosome, not plasmids, and IncA/IncC type plasmids that are known to mobilise them. As far as I know this effect does not apply more generally to MRR on plasmids - if it does then additional/different references would need to be cited to support this. Similarly, the only ‘island’ that Ref. 25 appears to discuss is SGI, in addition to interactions between phages and plasmids.

Response: Following this comment, and also the general comment above on the manuscript coherence, we replaced this section with a description of plasmid taxonomic units. We think that this section in the introduction serves well to connect our question on ARG content and the evolution of plasmid taxonomic units.

2) Results

Line 91 - going by descriptions in the response, and the 30% identity mentioned on line 472, proteins that would be clustered in the same family could have different origins and mobilisation histories (see R2 comments on original manuscript). These are more than minor differences seen in ‘epidemiological variants’ (usually only one to a few amino acid differences), so this is still problematic.

Response: We agree that this way of clustering gene families may have some limitations, yet this is a property of the data that we cannot change. At the same time, we think that this resolution of ‘homologous gene families’ (or ‘COGs’; Tatusov et al. 1997 doi: 10.1126/science.278.5338.631) is suitable for phylogenomic reconstructions of plasmids in the KES genera and for answering our research questions on the evolution of plasmid gene content. Please note that the gene families data is integrated in our pipeline with information on CSBs and transposable elements. We added a section in the supplementary note (SN1) to demonstrate how different epidemiological variants are distinguished at the level of CSBs.

Lines 100-1 – “matched the sequence” is still not clear. See previous comment. Do you mean that known mobile elements (or sequences matching part of them, or SRR genes?) are detected within these CBS? Fig. 1A doesn’t really seem to relate to this point.

Response: The details of this analysis are described in the methods section where we explain the thresholds used for this analysis. In order to explain our pipeline in more detail we now included a demonstrative example of the pipeline outcomes in the new supplementary note (SN1) section.

Lines 112-3 – how is this known? A suitable reference should be cited.

Response: our protein clustering method is aiming to find orthologous gene families. Frequent duplication will split paralogs into different gene families. We observed those paralogous gene families encoding transposases and SSRs in our dataset and analysis. This is a property of our data of which we are well aware. We modified that sentence such that it doesn’t sound like a

general statement (hence a reference is not required here, we think).

Lines 113-5 – this is also not easy to follow.

Response: Our clustering analysis results IS26 and IS26-v1 in two different gene families, despite of the extremely high sequence similarity. The difference between them is a single amino-acid in their protein sequence. This slight variation coupled with the frequent coexistences of IS26 and IS26-v1 on the same replicons (plasmid or chromosomes) contributes to the formation of two distinct families during the clustering process. The best reciprocal blast hit of IS26 are mostly other IS26 instances on different replicons as orthologs, IS26-v1 is a paralog. The clustering procedure subsequently forms two families for them separately. We slightly modified this sentence to make it clearer and also refer to the original manuscript describing the general principle at the basis of this approach (which is quite standard today in comparative genomics).

Line 119 – to be truly ‘plasmid specific’ wouldn’t there have to be no examples outside plasmids?

Response: Correct. We cannot rule out such exceptions. We deleted this and kept only the comment about the biased distribution.

Line 122-5 – IS26 and IS26-v1 encode more or less the same transposase and all IS6100 transposases are more or less identical. Similarly, there is little variation in the integrase of class 1 integrons, called IntI1 (this name could be used here) – having very closely related intl genes is the basis for the designation “class 1”. In contrast, Tn3, IS91 and IS110 here refers to the families which will include many different Tn/IS with significantly different transposases See also comments on previous version.

Response: We agree with the reviewer that our gene families have different ranges of sequence similarity among their member sequences. This is the nature of our method and dataset, to make this property transparent, we supplied phylogenetic trees as supplementary figure S3 for each of the gene families mentioned here.

Lines 127-31 – does this mean that a REI need not contain a complete transposon?

Response: Yes, exactly. All of our analysis is based on the pieces of island as we defined before. We hope that the new supplementary note (SN1) is helpful in explaining our pipeline.

Lines 131-6 – hitchhiking is normally used to describe when a non-mobile type of element (e.g., a typical class 1 integron that a defective tni transposition region) is mobilised by another element (in this case, a Tn21 family transposon). Because IS4321/IS5075 insert in the 38 bp TIR of Tn21 family transposons that are recognised by the transposase it is likely that they interfere with transposition, rather than ‘hitchhiking’ on these transposons - see previous comments.

Also, Lines 133-4 are incorrect - the only examples of the IS110 family that appear to be referred to in this manuscript are IS4321 and IS5075, which do target Tn21 subfamily 38 bp TIR, but this applies only a subset of the IS1111 subfamily of the IS110 family. Other members of the IS110/IS1111 family target different locations e.g., the attC sites of gene cassettes.

Response: We modified the text to be more specific. Many thanks for pointing this out (line 387).

Lines 167-9 – “two SRR variants of IS26” is problematic. It would be simpler/clearer to describe them as the transposase (or *tnp*) genes of IS26 and IS26-v1. *sul1* is found in the 3'-conserved segment that is common to a large proportion of class 1 integrons. The resulting close proximity to a great variety of gene cassettes likely at least partly explains the large number of connections that *sul1* has. Why is “*sul2/sul3*” here? - these are different genes with different origins known to be found in different contexts and they are not indicated in Fig. 2A.

Response: Thanks for pointing this out. We modified the term “variants”. Sorry for the misleading location of that label. The gene family *sul2/sul3* is represented by another node in the central network, we now use *qacEΔ1* as a second example of ARG node found in the central location (Figure 2A; line 524).

Lines 180-2 – meaning is unclear.

Response: we modified that sentence. It means that the trend we see is relevant to ca. 40% of the network (lines 565-567).

Lines 182 – Fig. 2C seems to show only 7 ARG added after 2011, not 8. *blaVEB-1* was first described in 1999 and maybe complete plasmids carrying this gene were not sequenced until after 2011 and there has been no retrospective sequencing. The same may be true for other rarer genes or those that have not been well studied yet.

Response: We definitely agree with this comment. Our analysis is limited by the publicly available genome sequences and may be subject to bias due to delayed sampling and sequencing, this delay could result in an underrepresentation of certain ARGs. In Fig. 2C we marked 7 genes added after 2011 out, but we have another minor *sul1* gene family with only 2 instances on plasmids, which was added once in 2014. Those two genes were clustered into another gene family due to the requirement of a reciprocal best blast hit among family members. This is an example for ‘noise’ in our large-scale analysis. Please note that none of our conclusions is based on such gene families. Since we omitted that gene family in the figure, we modified that ‘eight’ to ‘seven’ (line 567). Many thanks for that comment.

Line 206 – again, these protein families have many variants that are much more divergent than those encoded by IS26 and IS26 v1 and the analysis mentions only a couple of examples of each.

Response: Indeed. This note refers only to those families showed in figure S1. This is a general description of the rapid evolution of novel variants, even if that are more than two main variants.

Lines 226-7 – is this really true? e.g., see Ref 5. It is well known that large and small plasmids have different characteristics, replication mechanisms and therefore distinct rep gene types etc - see previous comments.

Response: We agree with this comment, in principle, yet, this is our intro for going into PTUs, which were published in 2020/2021. We modified this sentence to avoid misunderstanding (lines 678-680)

Lines 232-5 - how is it suggested that these ARG get on these plasmids? See previous comments about loss of mobile elements from small plasmids following insertion by MGE.

Response: This section does not mention loss of MGE from small plasmids. We later on explain

the distribution of ARGs in these plasmids. Note that if we have only a single ARG in plasmids, these are not included in our results as 'resistance island pieces' since the very first criterion is the presence of at least two co-occurring ARGs. This part of the pipeline is in accordance with our focus on regions that contain multiple ARGs. To make that clear we modified the sentence to include 'multiple drug resistance' (line 784).

Lines 262-3 – wouldn't it be better to leave these cointegrates out of this analysis. Do they have other ('large plasmid type') replicon(s)?

Response: Indeed, we add details on that in comments of the supplementary table S1, they do have large type replicons. We prefer to keep this set as is – also as it highlights the kinds of plasmids that may be grouped into the same phylogeny of plasmid backbone genes.

Line 265 – are all of these Tn1331deta:IS26? Several plasmids with complete Tn1331 have been sequenced. If the only difference is the insertion and the backbones are the same, then are they really different plasmids?

Response: Yes, the plasmids we describe here all contain Tn1331deta:IS26. The small plasmids are highly similar (and sometime identical), but found in strains with different genetic backgrounds, which means the small plasmids are quite conserved (see also comment by R2 and our response on that matter).

Line 269 – this could be worded better – the single insertion is in a single plasmid, which then spreads?

Response: Right. We modified this sentence (lines 776-777).

Line 274 – are these the cointegrates?

Response: Yes, see table S1. We write later on that they likely originated from fusion events.

Lines 285-7 - without the accession nos. it is not easy to work out which plasmid associated with Ref. 38 is meant here - CP045195 has blaOXA-181 and qnrS1 but is described as IncX3 in Ref. 38 and CP045194 doesn't appear to have ARG.

Response: The Ref. 38 we cited here is for plasmid pWSD411_4 (NZ_CP045677.1), which is one of the three non-mobilizable E15 MDR plasmids (two others are NZ_CP061398.1 and NZ_CP021962.1). Plasmid pWSD411_4 doesn't have ARG to β -lactams, but the rest of two plasmids have. We modified the text accordingly (line 794).

Line 291 – PTU-FS are presumably Salmonella virulence plasmids (pSLT type) that are known to rarely carry ARG, so this is not surprising. Also Lines 328-31, 439-444.

Response: Indeed. We discuss these observations in the context plasmid PTUs that include MDR plasmids. The PTU-FS plasmids correspond only partially to the Salmonella virulence plasmids (see also our Hanke et al. 2023 mentioned above). Following this comment, we added information on that in the discussion (lines 1113-1114).

Line 347 – what is meant by "shared properties"? This is too vague.

Response: we modified the text, here we mean shared plasmid sequence similarity (line 888),

Lines 351-2 – what is the evidence for the “evolution of host specificity”?

Response: here we refer to the distribution of the three types of plasmids in the strains of different host. This is explained two sentences above (lines 890-891).

Lines 360-1 – see comments on previous version. It is well known that different plasmid backbones can have related MRR.

Response: Indeed. We modified the sentence and added one reference (line 908).

Lines 379-86 – In numbers are really only cassette array numbers. In36 has aadA2 according to INTEGRALL, not aadA3. The genes are normally written *dfrA16-aadA2* to indicate the cassette array, then *qacEdelta1* (the E is upper case, the delta symbol should be used), then *sul1*. The same genes are then listed on Lines 383-4, which is presumably an error?

Response: Thank you for this comment. We double checked our data, fixed the wrong ARG names. The two plasmid groups FE and FK have differences in their shared CSB content, we give here just the most prominent CSBs for example. The two example CSBs are quite similar but have different gene cassettes *dfrA17-aadA5* in FE and *dfrA12-aadA2* in FK. We modified that section and report the ARGs included in the CSBs (Figure 6.).

Lines 385 – this is not new – a relatively small set of MGE is known to move ARG in these species.

Response: Indeed, MGEs are known to transfer ARGs between species, here we refer to the possibility of transfer between PTUs. Following this comment and a comment by R2 we modified that section to include additional results on the possibility of transfer between PTUs. See the new Fig. 6 and the relevant results section (lines 940-982) and discussion section (lines 1060-1071).

3) Discussion

Lines 397-8 – again, this is already well known in the field (see Lines 56-8).

Response: Indeed, but as far as we know this phenomenon has not been quantified (as we do in our study).

Line 400 – see comment above on the term “hitchhiking” - Ref. 31 does not appear to use this term. A better example of “functional dependencies” would be class 1 integrons/Tn402-type transposons being *res* site hunters that target Tn21-family Tn.

Response: we thank reviewer for this suggestion and we adopted the suggested example in our discussion (line 1000).

Line 401 – Ref. 45 seems to be more about IS26-mediated deletions during evolution than insertions and doesn't appear to mention “essential genes”.

Response: Well, it's a combination of both aspects. This study shows how a IS26-mediated deletion may ‘knockout’ an essential plasmid locus.

Line 410 – tetracyclines are also old antibiotics.

Response: Indeed. We modified this section, also following the comment from R#2 (starting in line 1009).

Line 414 – what kind of combinations?

Response: we deleted this sentence.

Line 424 – what is meant by “PTU-specific” here? That only some PTU typically have REI?

Response: Exactly, we show that REIs are found in specific PTUs, but not simply large conjugative plasmids. We also show that different PTUs also contain different REIs.

Lines 426-9 – it’s not really clear what is meant by “horizontal transfer” or “vertical inheritance” here, normally used to refer to plasmids and bacterial cells, not to ARGs moving between plasmids- see previous comments on LGT.

Response: yes, this is exactly what we mean. We modified the sentence to accommodate this comment (line 1068-1071).

Lines 436-8 – this already known – see previous comment - small plasmids may lose TE after ARG have been inserted – see e.g., <https://doi.org/10.1016/j.plasmid.2011.10.001> in reference to small mobilizable IncQ-type plasmids

Response: Thank you for this comment. Losing a TE is not exactly evidence for barriers for plasmid size increase in plasmid evolution. The loss can be related also to fitness effect of the plasmid on the host. The suggested reference is very helpful. We now cite it in the context of ARGs in small plasmids in the last discussion paragraph (lines 1081-1108).

Line 445 – Salmonella have a different lifestyle from E. coli and Klebsiella.

Response: right. This is what we mean here.

4) Methods

Lines 455-7 – the plasmids are found in those genomes?

Response: yes. We modified to strains to make it clearer.

Line 472 – this seems a very low threshold for protein families.

Response: in our experience the results using such a threshold are quite trustworthy. Please note that this sequence similarity threshold is calculated for global (rather than local) pairwise alignments. We consider this threshold (30% identical amino acids in a global alignment) sufficient for clustering of orthologs and paralogs (but not for highly similar epidemiological variants).

Line 476 – what is meant by “chromosomal families” here? Some families presumably include ARG found both intrinsically on the chromosome but also mobilised on plasmids e.g., blaSHV, fosA, oxqAB. What does this 3% of contradictions include?

Response: here we mean that 97% of the contradictions between ‘same gene family’ and ‘same CARD annotation’ are observed in chromosomal genes and 3% are observed in plasmid genes. It basically means that our gene family clustering approach is more trustworthy for plasmid genes

(because chromosomal ARGs are often variants of chromosomal core genes). We modified the sentence to be more precise (lines 1183-1184).

Lines 456-7 - should 'genomes' be 'chromosomes' here?

Response: Yes. We use complete genomes here, so the number of genomes is the number of chromosomes, all KES strains have only one chromosome.

Lines 465 & 474 – is it correct that these numbers are different? If so, why?

Response: Yes, this is correct. Not all ARGs are clustered into families (as we also explain in our response to R#2).

Lines 487-93 – it's not clear why this is still included, given comments on previous version.

Response: Identifying significantly cooccurring ARGs (coARGs) is still part of our pipeline.

Lines 495-8 – wording doesn't make sense and "insertions"

Response: Using the term 'insertions' we follow the terms of the original paper that described the method and its parameters (one of which is called 'insertions'). This parameter sets the number of genes that are allowed to differ between syntenic blocks in the search for CSBs (see also the original publication of CSB-Finder). We allowed for zero insertions – i.e., we are looking for identical gene clusters.

Lines 505 – this is still not well explained and was not addressed in response to previous comments. Why two protein coding genes? Are these in the MGE or the overlap? Many IS only encode a single transposase.

Response: please note that we are looking here at CSBs with at least two ARGs. This is now explained with an example in the supplementary note (SN1).

Line 507 "sequences"

Response: thanks. We modified accordingly.

Lines 509 – this section is entitled "ARG island analysis" but doesn't mention REI.

Response: This analysis corresponds to the results presented in Fig. 1 where no REI are yet discussed.

5) Other scientific points/accuracy etc

Throughout: conjugation and mobilisation abilities are only predicted, not experimentally tested, and should be stated as such.

Response: Correct. We added 'predicted' in the figure legends and the methods section.

There are many places where "gene" is used when it should be "protein" and vice versa e.g.:
Line 93 – the genes encoding the proteins co-occur, not the proteins. Lines 120-21 – it is the genes that may co-occur on plasmids, not the proteins; Lines 143, 282, 296 - SRR genes;

Response: Thank you for this comment. We have now streamlined the text throughout the manuscript.

Line 194 – IS26v1 does not have a single amino acid change, its Tnp protein does;

Response: that’s exactly what we meant with the citation one sentence before; we add here now a transposase (line 617).

Line 196 – IS26 variants do not comprise a protein family;

Response: we modified this sentence (line 619).

Table S4 proteins are not “found in” plasmids or chromosomes.

Response: we modified the table title (now S2).

Table S5 footnotes “number of ABR plasmids encoding antibiotic resistance genes from both protein families”.

Response: we modified that footnote (Table S5K, S5E, S5S).

Line 39, 66, 76, 233 etc – I agree with previous Reviewer 2 that using the term “plasmid genome” is best avoided. In most cases just “plasmid” or “plasmids” can be used.

Response: while we understand this view, we prefer to use the term genome. We tried to avoid it where it wasn’t absolutely necessary but we also think of plasmids as DNA molecules (e.g., when discussing plasmid size) hence we find it easier to use genome in several cases.

Lines 43, 65, 92-3 etc - genes encode proteins, they are not encoded. They can be described as being carried by plasmids, found on plasmid etc. Similarly for RE, MRR etc .

Response: thank you for that comment. We modified the text throughout the manuscript.

Line 69 - “elements” is not needed – the name is “IS”.

Response: thank you. We modified this line.

Lines 102, 111, 116 - “encode for” is still used– see previous comments.

Response: We modified the text throughout the manuscript.

Lines 260-1 – while most plasmids in the Col4401 clade have a purple box to indicate PTU-E4, only a few in the ColRNAI clade have a yellow box to indicate PTU-E71, so “correspond to” seems inappropriate here.

Response: Indeed, we now fixed the mistake and modified the sentence (line 745).

6) Figures

Figure 1

Part A, 4 – it is unclear what this is supposed to show. It looks it is showing that the ‘resistance island’ is the same as the ‘transposable element’? What is the meaning of ‘transposable

element' here? This term is problematic, as explained in previous comments. Having “SSR” genes present at both ends does not necessarily mean that a region is transposable, it depends on the type of SRR and how they function. In Part B, what are the units of distance?

Response: this term corresponds our definition and the text. We now point the reader to the methods for additional clarifications in that Figure's legend.

Figure 2

A – the size of the nodes is proportional to the number of times that ARG / SRR gene was seen in the data? B-F – these could specify that the date is the isolation date on the axes. D – the legend should state what grey parts of the bars represent? In the legend, what is meant by 'sequencing artefacts'? Errors?

Response: Thanks for the comments. We explain now the date in the legend and state the grey parts in the stacked bar plot. Yes, we meant errors. We modified to sequencing errors.

Figure 3

Legend – the meaning of “in proportion of plasmids in each PTU” needs rewording. What is the cut off for the small and large plasmid categories – 19 kb as on line 231? “bacterial host genus”? The wording of the second-last sentence does not make sense. Also, “The names of the PTU encoding..”

Response: we modified that figure legend.

Figure 5 – does this add to original PTU analysis in Ref 26.

Response: definitely. That study didn't look at ARGs. Also, that study does not include an inference of PTU relatedness.

7) Supplementary

The analysis of plasmid size distribution has been moved to the “Supplementary text” with former Fig. 3 now as Fig. ST2 and Fig. SN1 showing a density plot for plasmid size, giving a bi-modal distribution, as reported in several previous papers by different authors. A recent preprint (<https://doi.org/10.1101/2023.09.10.557055>) suggests problems with using a log scale for this type of analysis although it does show a bi-modal distribution for plasmids in *Enteobacteriaeae* (which encompasses the species analysed here) on a linear scale, giving a different antinode [~30 kb, although the preprint does state that “the shift in the antimode too small to affect general results (yet, this remains to be verified)”. The senior author of the manuscript under review is acknowledged in this preprint so should be aware of it.

Response: thanks for this comment. Yes of course we are aware of that publication. We discussed that opinion with the authors of the other publication (I Dewan and H Uecker). We think that they have a point when it comes to a transparent presentation of numerical data but in the case of plasmid size, the presentation using a log scale reveals a biologically relevant pattern in the data (namely, the two main size classes of KES plasmids).

Table S1. Seems to be an unchanged version of Table 1 in the original version that does not quite fit in with the new focus of the manuscript.

Tables S2 and S3 – SRR and transposases

It seems like these tables could be combined. Are SSR that were excluded from the analysis

(Lines 484-6) listed here? If so, they should be indicated.

Table S3 - The Methods don't seem to explain how the bias was calculated?

Table S4 doesn't have a title and might be better as Table S1, suitably introduced first in the text. NA needs explaining. "Two plasmid duplications" in comments is unclear

Response: Thank you for these comments. In the revised manuscript we slightly changed the supplementary tables order. Table S1 is now the contigs list. Table S3 (formerly S1) presents the results of the analysis pipeline. We added a comment that the chromosomal data is shown for reasons of data integrity. Tables S2 and S3 are now combined as table S2, the excluded SRRs are not listed in table S2. The biased distribution was tested using Fisher's exact test, now added in the methods section. Inf is positive infinity – we have notes at the bottom of each table where this info is listed (also 'NA'). We now refer the reader to those from the table title.

Table S5K, E and S – inclusion of these tables doesn't seem to take account of previous comments about problems with grouping ARG into quite broad families based on mechanism. Some names used are for specific genes (e.g., CTX-M-15, dfrA12) while other are not (e.g., aadA, which includes many distinct named variants), so it's not clear whether particular genes/proteins or families are meant. The name and content of each "family" really needed to be explained somewhere. Also, the names used are a mix of gene and protein names - this should be consistent (gene names should not start with an uppercase letter, the gene name is mcr-1.1, MCR-1.1 is the protein name, mph(A) is the correct format).

It's not clear why "Escherichia coli soxR with mutation conferring antibiotic resistance" is included in Table S5E - information in CARD suggests this is a mutation in a chromosomal gene, not an acquired ARG. What does "inf" indicate?

Response: our approach works within the framework of gene families, those are genes that are evolutionary related via the process of species divergence, gene duplication and gene transfer (that is, orthologs, paralogs and xenologs). The clustering of genes into gene families is based on sequence similarity (not the mechanism as pointed out by the referee). This is a fundamental definition (and analysis approach) in the field of molecular evolution and we acknowledge the fact that this resolution of the data is not comparable to looking at epidemiologically relevant variants in the manuscript. Nonetheless, we would like to emphasize that the integration of that data with the CSBs appears as extremely useful in distinguishing between epidemiological variants that have a distinct genomic neighbourhood, i.e., specific CSBs. We highlight that in the new supplementary note (SN1).

We fixed the all ARG names in the main article, thanks for pointing out. As for the table: we report all the annotations of the table in order to be transparent. All the ARGs we reported here were identified using CARD, we also reported the name of that ARO used by CARD accordingly, we added 'ARO' and 'ARO name' in the Table S2. soxR with mutation in Table S5 is one example of chromosomal gene found in plasmid (it is a very rare case, also shown with the numbers in the first four columns in Table S5), this gene was not found in any plasmid resistance islands. The meaning of 'inf' (infinite) is listed at the table footnotes.

8) Examples of minor formatting, wording etc problems

Line 41 - "antibiotic resistant"

Line 63 – "Interactions between plasmids...have direct".

Line 66 – suggest "Small plasmids typically..." for simplicity.

Line 121 - should be "ARG protein families".

Line 136 - "transposes" needs fixing.

Line 192 – "An example", "encoded by".

Line 199 – "insight into".

Line 208 - "hints as for temporal aspects" is unclear.
Line 218 - "has a higher abundance".
Line 221 - "additional sequencing" or "additional sequence data".
Line 222 - "existing" not "exiting"?
Line 276 - "a few".
Line 316 - "hints as for the association" does not make sense.
Line 321 - "or Klebsiella"?
Line 348 - "related to" and this sentence does not make sense.
Line 378 - "with each other".
Line 388 - wording does not make sense.
Line 391 - "the responses to such".
Line 402 - "comprising" is not the right word here, as plasmids have other components that make them plasmids. "carrying" would be better.
Line 404-5- is "these" referring to ARG or REI?
Line 405 - meaning the number of available plasmid sequences have increased? This could simply be stated.
Lines 410-11 - "sulphonamides", "colistin" - no upper case.
Line 414 - "resistance island building blocks".
Line 453 - "as in our recent" does not make sense. Word(s) missing?
Line 459 - "homologs of"
Line 478 - "members not identified as ARGs by CARD were excluded" might be clearer/easier to read.
Lines 482-4 - "were obtained", "were not included".
Line 485 - problems with parentheses and "recombinases".
Line 500 - "mobile genetic element databases".
Line 513 - "the distances between"
Line 528 - "transposases"
Tables S3 and S5 "reisistance" and "adjusted" in column headings/ footnotes needs fixing.

Response: We thank the reviewer for these comments. The typos were fixed and awkward sentences were pruned.

9) New comments on responses to previous comments

PREVIOUS COMMENT: Methods Line 531 - how was "plasmid carrying" defined/identified?

RESPONSE: in each of the complete genome assemblies there is at least one plasmid.

NEW COMMENT: Now Lines 451-3 - How was this identified? e.g. from the presence of circular sequences of appropriate length, from the name 'plasmid' in the description?

Response: yes, from the description of the sequence record. We did identify some 'plasmid' sequences in KES strains that appeared to share too much genes with the chromosome. These were marked by us as likely assembly errors, and often we have seen that such 'plasmids' disappeared from updated versions of the genome assembly (an automatic process done by NCBI).

RESPONSE: CARD identified only one ampC in a plasmid from E. coli (accession: NZ_CP048309.1) in our dataset.

NEW COMMENT: The description of the sequence under NZ_CP048309.1 calls it a plasmid, but the annotations and a BLASTn search suggest that it is a fragment of chromosome. This could be a wider problem in the sequences analysed.

Response: we agree with this comment (see above). Obviously, we missed this one, many

thanks for highlighting this contig. Following this comment, we flagged this contig in our data (Supplementary Table S1). This occurrence is not mentioned in the current version of the manuscript.

PREVIOUS COMMENT: Line 229 – *catI* is not the correct name for this gene – it is probably *catA1*, again known to be associated with Tn21 in sequences from back as far as the 1950s (see doi: 6 10.1128/MMBR.63.3.507-522.1999).

RESPONSE: here we adopted the annotation from CARD, which formerly used *catI*. We note that the suggested reference also uses *catI* as a synonym.

NEW COMMENT: CARD does not always use the most appropriate ARG names. *catI* is an outdated name and the paper mentioned here includes it in parentheses to provide a link back. *catB* is used in Table S5 for a distinct *cat* family and *catA* would be more consistent with this.

Response: thank you. CARD also updated the name to *catA1*. We modified the gene name in Table S5.

PREVIOUS COMMENT: Lines 530-45 – much of the wording here is identical/near identical to the start of the Methods section in Ref. 16, which has authors in common. While such reuse of text in Methods may not be a problem, and can in fact be useful, the previous paper should at least be cited here to show that this approach/dataset is not specific to this manuscript. Are the differences in numbers due to inclusion of an extra 78 complete genomes that were not from RefSeq in Ref. 16? Also see Reviewer 2's comments.

RESPONSE: point well taken. We now cite our previous manuscript. The source of differences in numbers is indeed as the referee described.

NEW COMMENT: While this manuscript (now Ref. 32) is cited on Lines 453 in Methods, the previous analysis and any overlap (or lack of) still needs to be referred to more explicitly.

Response: there is no overlap in the analyses between the present manuscript and our previous publication. To make is clear, we added that 'the same dataset was used in...' (lines 1124).

PREVIOUS COMMENT: Line 534 – it is not clear what is meant by "Samples lacking host information were examined in detail". What were they examined for and how was this done?

RESPONSE: this means that we looked up the background of the strains manually in BioSample, GenBank database and literatures, we looked for any info indicating that the sample has been engineered or evolved in laboratory. We added the word 'manually'.

NEW COMMENT: More information should be given in the manuscript itself. For example, did you also try and obtain sample dates from publication, if this information was missing from database entries?

Response: we added here an explanation that we examined in the detail the strain metadata. We did not try to search for isolation date for those isolates with no date in the metadata.

REVIEWER 2 PREVIOUS COMMENT: L49 Define what a transposable element is.

RESPONSE: the introduction was rewritten and includes a definition of the terms used throughout.

NEW COMMENT: No clear definition of transposable element appears to be given in the revised manuscript.

Response: we added a citation of the databases we used for this annotation in the first mentioning of this term (line 321).

PREVIOUS COMMENT: Lines 615 – the supplementary information does not provide enough detail to follow some parts of the manuscript.

RESPONSE: the supplementary material was modified and shortened considerably.

NEW COMMENT: Shortening the supplementary material did not deal with this, as more information is needed, not less, to judge whether the results of the analysis are accurate/meaningful.

Response: sorry for that misunderstanding. We now supply a new supplementary note, new (or modified) supplementary tables and we also provide the full dataset used in the analysis. We also supply the CSBs data and provide an explanatory video that shows how to read this data using CSB Finder (which has to be installed).

Figs. 4 and S8

PREVIOUS COMMENT Fig. S8: the title says that this figure is for Escherichia and Salmonella only, Klebsiella included in the key and there seem to be the odd pink block in the innermost circle.

RESPONSE: We deleted Escherichia and Salmonella from the title. This part of Rop Phylogeny in the figure shows mainly the Rop proteins in Escherichia and Salmonella, indeed it includes a few Rop proteins in Klebsiella. We split the Rop phylogeny according to the sequence similarity, but not strictly according to the host genus.

NEW COMMENT: Now Figs. 4 and S2 and Lines 255-60. The distinction between the two trees is not well explained in text. Presumably the ‘mostly in Klebsiella’ and ‘mostly in E. coli/Salmonella’ groups were clearly separated on a tree that included all 903 Rop sequences, so you have shown these as two different trees?

Response: Exactly. These are two parts of the Rop phylogeny we added the genera in the title of the supplementary figure legend (line 507 in supplementary material file).

PREVIOUS COMMENT: The references cited are not always the most appropriate and the information contained in many of those cited doesn't seem to have been taken into account when designing the analysis or interpreting results. The reference list also needs to be formatted correctly (journal name abbreviations, IS numbers should be in italics, use of Title Case Like This Needs Correcting) and some are missing page numbers or equivalents (e.g., Refs. 7, 16, 19, 32).

RESPONSE: thank you for this comment. In the revised version we sought to better explain the limitation of our approach comparing to the type of analyses that are described in those references (which we would assume is mostly done manually and in much detail). The references have been checked and corrected.

NEW COMMENT: There are still problems with citations, where the information in the reference does not always appear to be accurately reflected in what is written in the text. Reference formatting also still has a few problems, e.g., Refs 31, 39, 44, 61 –Title Case needs fixing. Ref 34, mef(B) format is incorrect, Ref 35, blaNDM format is incorrect. Supplementary references also need checking.

Response: many thanks for this comment. We modified the faulty references and double checked the references in the supplementary file.

PREVIOUS COMMENT Line 383 – are these plasmids likely mobilizable?

RESPONSE: Yes, they are mobilizable and we mark that plasmid property in the Figure 4.

NEW COMMENT: Now Lines 271-3 - this referred to plasmids in the ‘Klebsiella ColRNAI clade’ (=PTU-E71, Line 261) found in Escherichia. In Fig. 4, all PTU-E71 plasmids appear to be shown

as non-mobilizable.

Response: Thanks for pointing this out, in the last reply we were referring to the three ColRNAI plasmids. There was a mistake in the parentheses of Line 261, now we fixed this sentence (line 745).

Reviewer #2 (Remarks to the Author):

The authors have resubmitted their article and made changes in response to the comments from myself and the other reviewers. I particularly appreciate that the authors decided to delete several sections where perhaps insurmountable concerns were raised. Not easy to do – thanks to the authors for taking the concerns raised seriously.

The main take-home is the final of the three results sections, that ARG movement is not a free-for-all of horizontal gene transfer in the world of plasmids (as those who work on it are aware) but rather something highly structured. Though the broad aspects of this are known, as Reviewer 3 said previously the quantification of these known (mobile) resistance islands and their ‘favourite plasmids’ in a large-scale way seems like a good thing to have done. I liked the new analysis in terms of PTUs (I hope not just because I suggested something along those lines) and I found the analysis of the strong linkage of resistance islands and PTUs interesting.

I still have concerns about the way the authors choose to present their work and a few other aspects.

Usual peer review disclaimer: written in haste, apologies if I misunderstood or missed things. Comments are not intended as arbitrary orders to be followed to get published, they are attempts at honest criticism/appraisal.

Response: many thanks for the comments and suggestions. All of these were extremely helpful in improving our manuscript.

Title

I don't think it reflects the paper's conclusion, and it reads as extremely vague. We don't need more vague papers about AMR that use lots of genomic data! This gets at what the fundamental research question is – which is still slightly unclear to me.

Possible alternatives (based on what I liked about the paper, so may not be the best options):

‘Antibiotic resistance islands are strongly associated with plasmid lineages

‘Antibiotic resistance islands in plasmids are largely restricted to specific plasmid lineages’

Response: Thank you for the suggestions, we changed the title in line with the second suggestion and modified the introduction to include an explanation on the plasmid lineages (i.e., PTUs). We also modified the abstract to clarify the research question. Instead of focussing on quantifying the contribution of MGEs (which both referees were very critical about), we now emphasize the second half of our study where we compare the ARG content in resistance islands between PTUs.

General comments

1. Certain resistance islands are often specific to particular plasmid lineages. I would expect linkage of ARGs between plasmid lineages to be more due to smaller MGEs carrying often just one ARG, such as the composite ISApI1 transposon that moved mcr-1.1 or the Tn125 blaNDM transposon. I didn't really get a discussion of this point from the authors – that is, what the links between PTUs are where they do exist. Even if they are rare, this is how the novel ARGs are

added into existing resistance islands, right? So it seems like it might be worth addressing.

Response: Thank you for this comment, following which we extended our analysis in this section. We added Fig. 6 that shows a demonstrative example of shared (nested) CSBs among the five PTUs. Furthermore, we extended on those results by quantifying the PTU diversity of CBSs using the Gini-Simpson index. The results are reported in lines 940-982 and the discussion of those results is in lines 1060-1071.

2. I also got a bit concerned after thinking about it that identical plasmids might be somehow biasing the signal of resistance island association with specific PTUs, but this is the usual problem with using available genomic data. Maybe it was addressed, but probably worth mentioning somewhere as a limitation (I may have missed)

Response: In our dataset redundancy of genomic data of the same strain has been carefully avoided, especially strains from laboratories. However, it's crucial to note that identical plasmids were identified in strains sampled in different background, especially small plasmids. Small plasmids are expected to evolve very slow due to segregation drift (see our Garoña et al. 2023 PLoS Genet doi: 10.1371/journal.pgen.1010829), hence identical plasmids may be conserved over long evolutionary time scales. Considering that, identical small plasmids shouldn't be simply excluded if they were sampled in different host backgrounds. We double checked the redundancy in our dataset and can report that out of the 2,021 MDR plasmids, a substantial 84% exhibit unique non-redundant gene content. Our data includes only few extreme examples of identical plasmids. There were 11 identical MDR plasmids found in *Klebsiella* strains obtained during an outbreak from various sources within hospital settings. Similarly, 8 identical MDR plasmids were identified in *Salmonella* strains sampled over different years. Considering these small frequencies, the redundancy in our data shouldn't bias the association of CSBs with specific PTUs too much. Following this comment, we added information on redundant plasmids in the Methods section.

Specific comments

L28 Confusing sentence: suggest revising to 'Resistance islands are almost always carried by large plasmids, which are not always conjugative.'

L33 This is very general and slightly unclear. I would rewrite. 'The architecture of ARGs in plasmids is attributed to the workings of smaller MGEs that operate mostly within existing plasmid lineages.'

Response: these are indeed a simpler formulation. We adopted these suggestions. Thanks (see modifications in the abstract).

L43 Do they have to be 'diverse' ARGs or is just >1 sufficient?

Response: Correct. We modified to multiple (line 78).

L46 This presentation is misleading. Some of this burden in terms of pathogen-drug combinations is not due to MDR plasmids - for example, MDR TB because M. tb has no plasmids.

Response: Correct. We now specifically describe the contribution of *Escherichia* and *Klebsiella* (lines 81-83).

L50 some would call a plasmid a MGE - ok to use this definition, but I would make it clear that you mean smaller MGEs.

Response: Correct. We don't think that the size is the distinguishing feature as some tiny plasmids are smaller than large transposable elements. We added a note to make it clear that the MGEs termed exclude plasmids (line 89).

L55 This makes it sound like the integron is inherently a mobilizing thing - but it isn't. 'Mobile integron' means an integron associated with a transposon or a plasmid - see Partridge 2009 <https://doi.org/10.1111/j.1574-6976.2009.00175.x>

Response: Thank you for this correction. We modified that sentence (and still cite Partridge 2011) (line 91).

L61 I disagree. I don't think it remains 'elusive' (Reviewer 3 also pointed this contradiction out in the previous round). In any case, I don't understand what exactly the authors mean by something as general as 'the contribution of MGEs to the evolution of MDR plasmids'. It's no problem to say that the contribution of this study is to confirm that ARGs on KES plasmids very much organise into resistance islands; known MGEs almost completely explain these resistance islands; and (the more interesting part) that these operate very much within plasmid lineages. I strongly suggest deleting the claim that it's 'elusive'

Response: sorry for that, it seems that our revision on that point was not sufficient. We now modified the sentence to make it clear that we refer to a quantification of the overall MGE contribution to plasmid evolution.

L63 again, I would say 'smaller MGEs'

Response: since we now qualified our MGE definition in the previous paragraph, we think that adding here 'smaller' is not necessary. We note that there are some plasmids that are smaller than some MGEs.

L69 Thanks for this mention. However, this makes it sound like the authors of these letters disagreed with the original study's claim that IS elements associated with ARGs are enriched in conjugative plasmids - but I don't! This is an established fact in the field. What I (and the authors of the other response) disagreed with was the methodology of that study and its claims to novelty.

Response: Sorry for that, we did not mean to give a wrong impression here. In the revised manuscript we still cite the two comments but in the same place since all authors seem to agree on the high frequency of MGEs in conjugative plasmids (line 104).

L71 'may have a deleterious effect' – perhaps a fussy point, but I would say 'can' because it sometimes does, it sometimes doesn't. 'may' to me makes it sounds like it's a general principle, but surely not, because in biology most things that are seen at all can sometimes have a cost, sometimes a benefit.

L72 'consequences for the plasmid interaction with the host' - Yes...but then again, the fact that it happens so much suggests that it is selected for overall in many cases

Response: this paragraph was replaced, also following comments from R1, with an introduction of the plasmid taxonomic units.

L74 I still don't understand what 'contribution to evolution' really means.

Response: we modified this to 'contribution to antibiotic resistance gene content'. We hope that this sentence is clearer now.

L76 co-occurring? (coinciding is strange for this context in English, I think)

Response: thinking about this issue again, we tend to agree. Consequently, we modified to cooccurrence throughout the manuscript.

L103 I agree with this statement. I like the analysis demonstrating this, but, as has been also pointed out by the other reviewers, this is well-established - to quote e.g. Partridge 2011: 'Available evidence indicates that in Gram-negative bacteria, particularly the Enterobacteriaceae, the resistance genes and associated mobile elements carried on plasmids are often found clustered together in large multiresistance regions (MRR).'

L104 'pieces of resistance islands' - I was expecting this to be a quote from Baquero 2004 - but the phrase doesn't occur anywhere in that paper

Response: Re L103: indeed, and we also explain and refer to that previous knowledge that in the introduction. The novelty in this section of our study is the quantification of the ARGs in such regions in *KES* plasmids. Re L104: we realize that this concept of 'pieces' or rather the CSBs requires additional explanations. Consequently, we now added a section of supplementary note (SN1) that demonstrates one demonstrative example of how our approach works. Additionally, we modified this paragraph and also added a sentence towards the explanation of CSBs (which we think was not well communicated before). It is important to note that the CSBs correspond to clusters of genes whose order is conserved in multiple plasmids. The CSBs we identify do not necessarily correspond to intact MGEs, which may be found in plasmids in partial form, e.g., due to genome rearrangements and MGE degradation processes (see the example in the new Supplementary Note SN1). The conserved gene clusters we identify here are thus better described as pieces of genetic elements – as previously nicely described (in somewhat abstract manner) by Baquero in that reference. Inspired by this view (hence the reference) we termed these CSBs 'pieces of resistance islands.' Taken together, our analysis shows that most cooccurring ARGs in plasmids are organized in conserved syntenic blocks (lines 324-328).

L134 This IS110 and Tn21 example is a nice one, and I'm glad the authors took seriously the comments about integrating existing knowledge. Is the coincidence with all Tn3 transposons or just driven by Tn21 subfamily?

Response: here we performed the test using gene family encoding Tn3 family transposases. As we plotted in the supplementary Fig. S3 the Tn3 family includes mainly transposases in Tn21 subfamily and Tn5403.

L154 Did you quantify anything about those ARGs that aren't within resistance islands? What are they?

Response: This comment was raised by R1 as well – the ca 20% that are not in the resistance island pieces are likely excluded due to sampling density. The 20% remaining ARGs are excluded gradually in our pipeline and most are excluded with the CSB step (see Supplementary Table S3), where our threshold requires that the syntenic block has to be conserved in >4 different strains (i.e., plasmids). This threshold is quite conservative. We now explain this in the first discussion paragraph (lines 992-996).

L185 Out of interest, what is mcr-1.1 connected to in the network? It emerged on a composite ISAp11 transposon as a 'single ARG' so I'm intrigued that it seems to have two connections in the network

Response: Thank you for this question. mcr-1.1 was connected to the network by a CSB including ISAp11 transposase and also blaCTX-M-14 and FosA3. This CSB was found in plasmids of four different strains in *Escherichia* and *Salmonella*. mcr-1.1 was found in a Tn6330-like structure without a 5' transposase. We added this information in the results (lines 571-575) and in a new supplementary figure S2. This example led us to double check the CSB length distribution, which we now present as supplementary figure S1. We also report the length of CSBs in the first results section. Indeed, some of our CSBs are rather long and likely correspond to likewise examples of highly conserved CSBs in closely related plasmids.

L202 ISs are extremely ancient - the first observations are surely due to changes in prevalence rather than truly de novo emergence, so I wouldn't say 'origin of IS26' - it's just in agreement with these first observations in genomic data

Response: Indeed. We modified to 'earlier observations' (line 626).

L208 'as for temporal' – some sort of typo. Revise: at temporal

Response: fixed. Thanks (line 631)

L220 'diverse and changes only little over time' - I like this way of putting it. Could even change 'and' to 'but' for emphasis.

Response: we adopted the suggestion. Thanks (line 643).

L222 'exiting [sic]' should be 'exciting'. On this point (maybe for discussion – see L405-406) isn't a possible driver of the reduced novelty over time that people now more commonly sequence any old isolate, whereas before they only sequenced something if it had a very interesting resistance phenotype. So, sequencing used to be far more biased towards novelty than it is now, when it is used (among other things) for routine genomic epidemiology

Response: we fixed that typo ('existing'). As for the reduced novelty in genome data: this is a likely hypothesis. We do not feel that our data enables us to speculate on that. It is possible that the pattern we see depends on the frequency of sequencing bacterial isolates (i.e., sampling density).

L241 I missed the explanation for why some plasmids aren't assigned a PTU? Is this a problem with COPLA assignment? Or co-integrates (surely not all of these?). Anyway, the pattern of separation is clear, but it surprised me so maybe worth saying in legend.

Response: the reference PTU catalogue don't cover these plasmids. We added a note on that in the legend.

L265 on Tn1331, perhaps of interest that the first observation of Tn1331 in *K. pneumoniae* pJHCMW1 in 1987 (PMID: 2830842)

Response: Thanks for that! We added the reference (line 749).

L275 I didn't see anywhere the average size of the resistance islands you identify – would be nice to add / make clearer – but presumably something of the order of 10kb? This is bigger than quite a lot of small plasmids. Not saying you are wrong in this point, just emphasises that the large size of resistance islands is a barrier to them being added to smaller plasmids. Physical considerations likely play a role: for example, work vesicle-mediated gene transfer shows that packaging and transfer times are similar for plasmid sizes up to 15 kb (PMID: 30670543). So for resistance islands it really makes sense that they can't just be added into a small plasmid, whereas for individual genes it seems (to me) harder to explain why a small plasmid shouldn't happen to gain ~1kb.

Response: Unfortunately, we cannot reliably supply the information on the size of resistance islands. We are looking at colinear syntenic blocks (CSBs) that can be nested and we do not have the exact boundaries. This is why we use the terminology of 'pieces of resistance islands'. To accommodate this comment, we added a new supplementary figure showing the size of CSBs (Supplementary Figure S1). This information is also reported in the first results section. We hope that the new explanation in the first results section and the extended supplementary note render this aspect clear now. That being said, we agree with the referee that small plasmids may evolve under some constraints for a small size, but we think that these constraints are related to the replication type rather than selection induced by vesicles-mediated transfer (which we think is not a very important mechanism for plasmid transfer; e.g., see our recent Nies et al. 2023 DOI: 10.1111/1758-2229.13203). Following a helpful comment from R1 we added an example for constraints on plasmid size due to the replication mechanism in IncQ plasmids (last discussion paragraph).

L331-333 I liked this particular example and conclusion.

Response: Thank you!

L396 'show that the' - I would say 'Our results provide a quantitative summary confirming previous observations' (or similar). It's not a new observation; doesn't mean it can't be a good analysis.

L397 'implicating the interaction between MGEs and plasmids as the major driving force in the evolution of MDR plasmids' – to me this is clearer than what is meant by the vaguer 'contribution to evolution' earlier. Maybe I'm misunderstanding but I find it hard to think about an *interaction* as itself a major driving force - I might say 'mechanism' instead? This is discussion so it's up to you really, but maybe this could also appear in the introduction too, because it (to me) explains what you are arguing for.

Response: Thanks. We modified this text to: 'Our results supply a quantitative assessment according to which the majority of antibiotic resistance genes in plasmids are located in resistance islands. In agreement with previous studies, our results implicate the interaction between mobile genetic elements and plasmids as the major driving force in the evolution of MDR plasmids.' We hope that this is acceptable (lines 990-998).

L402 'Plasmids are...in themselves, a hotspot'. On this point about resistance islands as hotspots: Rocha and Bikard (doi: 10.1371/journal.pbio.3001514) have proposed a model for defence island formation in bacterial genomes - MGE turnover at hotspots for integration could result in similar resistance island accumulation (is my belief - it makes sense). I should say I made this point in a discussion of a recent paper of mine (doi: 10.1101/2023.08.07.551646, in press Microbial Genomics) so I'm not insisting you add it, just preferring it because I find it

interesting.

Response: thanks for the references. Both are highly relevant and we now refer to those in the discussion.

L405 as above, see point about ARG combinations growing slower. Maybe if one normalises by sequencing they don't get slower.

Response: Indeed, yet we prefer to show the data as is.

L408 I don't think the 'temporal integration' is as interesting as all that. In particular, I wouldn't give these as 'main examples'. *sul1* is well-known as a very common gene nearly ubiquitous in clinical class 1 integrons, which has previously been put into an explicit model of their temporal evolution. See the model of their evolution in Gillings 2008 (doi: 10.1128/JB.00152-08). It's nice that it comes out as the most connected, but this is very much because of this known fact. Then for *mcr-1.1*, it's a very recent gene only detected in 2016, as you point out. It is not in any way connected to the introduction of colistin. To me using these as 'main examples' harms the paper rather than helping it.

Response: Right. We modified this sentence to point *sul1* and *mcr-1.1* as a validation of our approach. We also added there a short discussion of *tet(X4)* as an example for a recently reported ARG (starts in line 1046).

L410 'that where [sic]' should be 'that were'

Response: thanks. Fixed.

L411 Not sure what your source for 1970s is. I would say that polymyxins (of which colistin is one) was discovered from 1947 onwards (discovered multiple times independently by people screening *Bacillus* sp., in UK, US, Japan) and then launched as a drug in mid-1960s in US, but actually rarely used and still rarely used outside of MDR infections. The crucial fact for the emergence of *mcr-1.1* is the use of colistin for growth promotion in animal feed, which is what is believed to have driven the emergence of *mcr-1.1* from *Moraxella* onto plasmids in KES.

Response: Thank you for that. We modified this discussion paragraph to include the relevant references – we cite Liu 2016 (as before) and added also a comment by Shen et al. 2016 (lines 1013-1014).

L438 'plasmid genetics' - I wouldn't say 'plasmid genetics' - the basic biology of plasmids ?

Response: we modified this to plasmid replicon type.

L444 Interesting hypothesis about PTU-FS. They are *Salmonella* plasmids - I don't know whether this takes into account the presence of ARGs being different in *Salmonella* compared to KE? (I may have missed)

Response: This is a good point. We checked in our data and indeed *Salmonella* shares fewer ARGs with *Escherichia* and *Klebsiella* compared to the two latter genera. Please note that the PTU-FS plasmids very often do not encode any ARGs and this observation led to our hypothesis. Other *Salmonella* plasmids do harbour ARG and MGE combinations. We note that some of the *Salmonella* plasmids here are 'domesticated' (e.g., pSLT-like plasmids). Following a comment

from R1 we expanded on that in the last discussion paragraph.

Figures

Figure 1 I wonder if one restricted the chromosomal analysis to only those ARGs that are also seen on plasmids, would one see any difference between the ARGs and control?

Response: This is – hypothetically – a good point. We have some evidence for plasmid CSBs in chromosomes (i.e., MGEs) but did not study that in depth. There is a general bias in ARG presence in plasmids and chromosomes (e.g., as we report in Wang et al. 2021 doi: 10.1098/rstb.2020.0467) that would make such a comparison quite tricky due to very limited sample size of chromosomal instances.

Figure 3 I like this way of showing it. It might be worth adding number of resistance islands per PTU and their median/range in sizes somehow. That's something that I didn't get from the paper on a readthrough that would have been nice to quantify.

Response: As we replied above, with the current approach we cannot calculate the number or size of the resistance islands. We can show only the ARG/SRR content.

Supplementary Text

I still don't entirely understand the rationale for needing markers for plasmid size, but OK. I understand it's nice to make a phylogeny of a single gene present in all large plasmids and show it matches PTUs.

Response: Exactly. Assuming that PTUs will not solve the challenge of general plasmid classification entirely, we deem the topic of plasmid marker genes important, also in the context of metagenomic analyses (see e.g., Fogarty et al. 2023 doi: 10.1101/2023.03.25.534219). Since plasmid size is often used in the field as a characteristic for various comparisons (e.g., as in Che et al. 2021), we think that it is important to show that specific plasmid size classes are distinct (i.e., not evolutionary related). Furthermore, those 'marker genes' can be useful for reconstructing evolutionary relations among PTUs (as we do here).

On PTUs being size-specific: Redondo-Salvo et al.'s original definition of PTUs is based on pairwise ANI with a 50% length threshold, so it seems to me this is in part built into their definition.

Response: Correct, but only indirectly. As far as we can tell they do not state clearly in their studies that PTUs are of similar size, which is an interesting observation (we think) in the context of plasmid evolution.

Author: (I have a policy of signing all peer reviews)

Conflict of interest: I discussed this article with Tal Dagan at a workshop we both attended in September 2023.

Response: Thank you!

Reviewer #3 (Remarks to the Author):

I think that the authors adequately revised manuscript to respond to reviewers' comments/suggestions. Description and presentation of the first part focused on SSR-ARG association (Fig.1, Fig2, supplementary table) have been improved and provide sufficiently clear

message. New results of REI - Plasmid-type association (Fig 3, Fig.5BC) provides useful information for broad audience. It might give inspiration of new research questions.

Response: Thank you!

I have only a few comments regarding minor points:

- The “SSR” term is normally used to refer to only serine-recombinase and tyrosine recombinase. The “SSR” term in this manuscript is used to cover both transposases (DEDD, DDE, HUH) and true site-specific recombinase for simplicity. This should be noted somewhere.

Response: we now add this note in the results.

- Line 286. Beta in symbol.

Response: thanks. Modified.

- Fig 2. Legends. (A) .. the edges to

- Fig 2. Legends. (D) What are the numbers on the stacked bar? Protein family IDs?

Response: we fixed the typo and modified the legend for clarity.

- Line 402. Earlier work by Sota et al 2007 (doi: 10.1128/JB.01906-06) addressed this point by experiments. This reference should be cited.

Response: Thank you for that! We now cite that study.